# ADAPTIVE ROBUST INTEGRATION OF INTERNAL DATA WITH EXTERNAL SUMMARIES UNDER DISTRIBUTIONAL SHIFT

## ABSTRACT

Integrating evidence from heterogeneous datasets is challenging when predictor spaces differ and data distributions shift. Large datasets such as biobanksabcdefghijrefer to as external dataabcdefghijoffer substantial sample sizes but often lack in-depth information due to cost constraints. In contrast, internal datasets from smaller analytic studies provide richer, individual-level detail. We propose a general Distributionally Robust Optimization (DRO) framework for integrating internal individual-level data with external summary-level data under distributional shift. Our method minimizes Cressie-Read divergence between a full model (fit to internal data with many predictors) and a reduced model (estimated from external data with fewer predictors), using a specialized nested-iteration algorithm. While effective under moderate shift, standard DRO can degrade when the distributional shift is severe. To mitigate this, we introduce an Empirical Bayes DRO (EB-DRO), which stabilizes estimates by adaptively shrinking toward internal-only solutions. We further develop an ensemble EB-DRO method that aggregates across multiple divergence families to improve robustness without selecting a single best family. Our proposed methods preserve privacy by operating on external summary statistics, support robust integration under shift, and enable valid inference when no shift is present. Simulations show that DRO improves over internal-only estimates under light shifts, EB-DRO adds stability under greater shifts, and ensemble EB-DRO achieves the most consistent robustness overall.

## 1 INTRODUCTION

The rapid expansion of large-scale data resources—spanning biobank-scale cohorts (e.g., UK Biobank, All of Us), healthcare systems with electronic health records, and digital traces from wearables and online platforms, is transforming both machine learning and applied sciences (Lapatas et al., 2015; Ashley, 2016; Russ et al., 2019). Large *external* datasets provide unprecedented breadth (sample size) but often lack detailed measurements due to cost constraints. Smaller *internal* studies, in contrast, collect richer covariates on a limited scale. The challenge is to integrate breadth with depth in a principled way, enhancing efficiency, robustness and generalizability of inference.

However, direct sharing of individual-level data is often blocked by privacy, regulatory, and logistical barriers (Rothstein, 2010; Kisselburgh and Beever, 2022), so large cohorts typically release only *summary statistics* (e.g., regression coefficients, odds ratios). These summaries, in practice, are often derived from *reduced covariate sets* and from populations that differ from the richly measured internal study, creating pervasive *distributional shift* between internal and external data sources. If ignored, such shifts can bias estimates and erode the benefits of integration (Zhai and Han, 2022; Taylor et al., 2023). To address this, we develop a *distributionally robust data integration* framework that combines individual-level data with external summary statistics, enabling efficient learning that is stable under diverse shift scenarios.

**Related work.** Existing methods typically mitigate distributional shift by targeting specific forms, such as covariate shift in which covariate distributions differ but the conditional model remain invariant, or conditional shift in which the conditional model itself differs across studies. For example, Han and Lawless (2019) employed a small supplementary sample to bridge marginal differences in covariates. When the supplementary sample is unavailable, Estes et al. (2018) proposed a matrix-weighted empirical Bayes estimator that guarantees prediction error no worse and often better than the internal-only estimate. Gu et al. (2023) extended this idea to settings with multiple external stud-

ies. In a similar spirit, Han et al. (2024) used a James-Stein-type shrinkage method to cast the shift as a linear constraint. Furthermore, Sheng et al. (2024) modeled the distributional shift via unknown parameters and developed shift-adjusted estimation procedures. Besides, selective-integration methods, such as Chen et al. (2021); Zhai and Han (2022), applied penalized shrinkage to many external sources, borrowing strength only when they align with the internal population and thus avoiding bias under shift. Sheng et al. (2022) proposed a two-step procedure: conduct a homogeneous population hypothesis testing to identify compatible studies and integrate information only from those retained. While effective in their settings, these approaches are not designed to adapt broadly across different shift mechanisms.

**Our approach.** We leverage distributionally robust optimization (DRO), which seeks models that perform well under worst-case perturbations of the distribution (Duchi and Namkoong, 2021). Although widely studied in statistics and machine learning, DRO has not, to our knowledge, been applied to data integration with external summaries under moment constraints. In related work, empirical likelihood (EL) has provided a popular framework for incorporating summary statistics by treating them as moment conditions (Qin, 2000; Chatterjee et al., 2016; Han and Lawless, 2019; Kundu et al., 2019). In particular, DRO's dual reweighting form generalizes EL, which preserves the empirical-likelihood structure while explicitly accommodating distributional shifts between studies. This makes DRO a natural and flexible foundation for the summary-based data integration. Building on the DRO framework, we introduce an empirical Bayes stabilization step that adaptively balances efficiency and robustness, and further develop an ensemble across divergence families. Taken together, these innovations yield an integrative method that delivers robust performance without requiring assumptions on the type or extent of distributional shift.

In this paper, we address the challenge of integrating internal individual-level data with external summary statistics when both distributions and covariate sets differ. Our main contributions are:

- We propose a DRO formulation for integrating internal individual-level data with external summaries under moment constraints, based on the Cressie-Read divergence with a dual representation and a scalable nested optimization algorithm.

- We develop an empirical Bayes stabilized variant (EB-DRO) that adaptively shrinks toward the internal-only estimator to safeguard against severe shift, and further introduce an ensemble EB-DRO that aggregates across the Cressie-Read family of divergences.

- Theoretically, we establish asymptotic normality of the DRO estimator, showing it is at least as efficient as the internal-only estimate without shift and converges to a pseudo-true value with a leading bias term under shift, and prove that EB-DRO enjoys an oracle risk guarantee.

- We demonstrate through simulations that DRO excels under mild shifts, EB-DRO stabilizes under stronger shifts, and the ensemble achieves the most consistent robustness across regimes.

## 2 METHODS

**Problem setup.** We consider two heterogeneous data sources:

- **Internal (individual-level) data:** $\{(Y_i, \mathbf{X}_i, \mathbf{Z}_i)\}_{i=1}^{N_{\text{in}}}$, where $\mathbf{X}_i \in \mathbb{R}^q$ are covariates measured in both studies and $\mathbf{Z}_i \in \mathbb{R}^{d-q}$ are additional covariates observed only internally.

- **External (summary-level) data:** a large study with only $\mathbf{X}$ observed, providing the maximum likelihood estimate (MLE) $\boldsymbol{\theta}^\star$ for the reduced model $f_{\boldsymbol{\theta}}(Y \mid \mathbf{X})$. Due to the massive sample size $N_{\text{ext}} \gtrsim 10^5$ or $10^6$, we treat $\boldsymbol{\theta}^\star$ as the true reduced-model parameter $\boldsymbol{\theta}$.

We model $Y$ via the following generalized linear models (GLMs):

$$Y \mid (\mathbf{X}, \mathbf{Z}) \sim \text{GLM}\{\eta = (\mathbf{X}^\top, \mathbf{Z}^\top)\boldsymbol{\beta}\}, \qquad Y \mid \mathbf{X} \sim \text{GLM}(\eta = \mathbf{X}^\top \boldsymbol{\theta}),$$

where $\boldsymbol{\beta} \in \mathbb{R}^d$ parameterizes the internal study and $\boldsymbol{\theta} \in \mathbb{R}^q$ is the reduced-model parameter for the external study. Notably, $\boldsymbol{\theta}$ is not simply the $\mathbf{X}$-subvector of $\boldsymbol{\beta}$, since in GLMs the estimated effect of a feature depends on other features included in the model.

**Distributional shift.** The two studies may involve different populations or follow different protocols, leading to *distributional shift*. Such shifts may stem from changes in the marginal covariate

distribution, the conditional distribution, or both. Our DRO formulation is agnostic to the shift type and provides robustness in all cases, though the conditional shift poses greater methodological challenges. Let $\boldsymbol{\beta}^\omega$ denote the external full-model parameter, which may differ from the internal $\boldsymbol{\beta}$; the reduced parameter $\boldsymbol{\theta}^\star$ is the projection of $\boldsymbol{\beta}^\omega$ onto the shared covariates $\mathbf{X}$. Our goal is to robustly estimate $\boldsymbol{\beta}$ by integrating internal data with the external summary $\boldsymbol{\theta}^\star$, leveraging the large external sample while accounting for possible shifts (see Fig. A1).

## 2.1 DISTRIBUTIONALLY ROBUST DATA INTEGRATION

### 2.1.1 FORMULATION AND DUAL REPRESENTATION

We estimate $\boldsymbol{\beta}$ via a distributionally robust optimization that augments the internal likelihood with a divergence penalty and a moment constraint. Let $p = (p_1, \ldots, p_{N_{\text{in}}})$ be non-negative weights defining a reweighted empirical distribution over the $N_{\text{in}}$ internal samples. Our objective is

$$\min_{\boldsymbol{\beta}} \left\{ -\frac{2}{N_{\text{in}}} \sum_{i=1}^{N_{\text{in}}} \log f_{\boldsymbol{\beta}}(Y_i \mid \mathbf{X}_i, \mathbf{Z}_i) + \min_{p_1, \ldots, p_{N_{\text{in}}}} \frac{1}{N_{\text{in}}} \sum_{i=1}^{N_{\text{in}}} \phi_k\left(\frac{p_i}{1/N_{\text{in}}}\right) \right\},$$

$$s.t. \quad \sum_{i=1}^{N_{\text{in}}} p_i = 1, \quad p_i \geq 0, \quad \sum_{i=1}^{N_{\text{in}}} p_i \boldsymbol{u}(\mathbf{X}_i, \mathbf{Z}_i; \boldsymbol{\beta}, \boldsymbol{\theta}^\star) = \mathbf{0}. \tag{1}$$

Here $f_{\boldsymbol{\beta}}$ is the GLM likelihood for observation $i$, and $\phi_k(\cdot)$ denotes a Cressie-Read $\phi$-divergence, which regularizes reweighting by measuring the deviation of $p$ from the uniform weights, with $k$ tuning its strength. The vector-valued moment function $\boldsymbol{u}$ encodes the external summary, defined as

$$\boldsymbol{u}(\mathbf{X}, \mathbf{Z}; \boldsymbol{\beta}, \boldsymbol{\theta}^\star) = \mathbb{E}_{f_{\boldsymbol{\beta}}(Y \mid \mathbf{X}, \mathbf{Z})}\{\boldsymbol{h}(Y, \mathbf{X}; \boldsymbol{\theta}^\star) \mid \mathbf{X}, \mathbf{Z}\} = \int \boldsymbol{h}(Y, \mathbf{X}; \boldsymbol{\theta}^\star) f_{\boldsymbol{\beta}}(Y \mid \mathbf{X}, \mathbf{Z}) dY,$$

where $\boldsymbol{h}(Y, \mathbf{X}; \boldsymbol{\theta}^\star) \in \mathbb{R}^q$ is the reduced-model estimating function used to obtain summary $\boldsymbol{\theta}^\star$ from external data. Although individual-level external data are unavailable, evaluating $\boldsymbol{h}(Y, \mathbf{X}; \boldsymbol{\theta}^\star)$ based on internal $Y$ and $\mathbf{X}$ enables moment matching for cross-source compatibility.

**Theorem 1** (Variational dual of DRO integration). *Let $\phi$ be a proper, closed, convex divergence with $\phi(1) = 0$ and convex conjugate $\phi^*$. We write $\boldsymbol{u}_i(\boldsymbol{\beta}) \equiv \boldsymbol{u}(\mathbf{X}_i, \mathbf{Z}_i; \boldsymbol{\beta}, \boldsymbol{\theta}^\star)$, then the estimator in Eq. 1 admits the saddle formulation:*

$$\min_{\boldsymbol{\beta}} \max_{\gamma \in \mathbb{R}, \boldsymbol{\lambda} \in \mathbb{R}^q} \left[ -\frac{2}{N_{\text{in}}} \sum_{i=1}^{N_{\text{in}}} \log f_{\boldsymbol{\beta}}(Y_i \mid \mathbf{X}_i, \mathbf{Z}_i) + \gamma - \frac{1}{N_{\text{in}}} \sum_{i=1}^{N_{\text{in}}} \phi^*\{\gamma + \boldsymbol{\lambda}^\top \boldsymbol{u}_i(\boldsymbol{\beta})\} \right].$$

Specializing to the Cressie-Read family, the inner maximization over $\gamma$ in Theorem 1 can be solved in closed form, yielding a compact expression depending only on $\boldsymbol{\lambda}$, stated below.

**Corollary 1** (Cressie-Read dual form). *For the Cressie-Read divergence $\phi_k(x) = \frac{2}{k(k+1)}(x^{-k} - 1)$ with conjugate $\phi_k^\star(y) = -\frac{2}{k}\left(-\frac{k+1}{2}y\right)^{k/(k+1)} + \frac{2}{k(k+1)}$, the dual DRO problem reduces to the following form (derivation in Appendix C):*

$$\min_{\boldsymbol{\beta}} \max_{\boldsymbol{\lambda} \in \mathbb{R}^q} \left\{ -\sum_{i=1}^{N_{\text{in}}} \log f_{\boldsymbol{\beta}}(Y_i \mid \mathbf{X}_i, \mathbf{Z}_i) + H_k(\boldsymbol{\lambda}, \boldsymbol{\beta}) \right\}, \tag{2}$$

*where*

$$H_k(\boldsymbol{\lambda}, \boldsymbol{\beta}) = \frac{N_{\text{in}}}{k(k+1)} \left[ \frac{1}{N_{\text{in}}} \sum_{i=1}^{N_{\text{in}}} \{1 + \boldsymbol{\lambda}^\top \boldsymbol{u}_i(\boldsymbol{\beta})\}^{\frac{k}{k+1}} \right]^{k+1}.$$

*We define the optimum value of Eq. 2 as the DRO estimator $\widehat{\boldsymbol{\beta}}_{\text{DRO},k}$.*

For a fixed $\boldsymbol{\beta}$, maximizing $H_k$ is equivalent to minimizing $M_k$, whose specific forms are reported in Table 1. Since the mapping $H_k \mapsto M_k$ is monotone, we have

$$\max_{\boldsymbol{\lambda} \in \mathbb{R}^q} H_k(\boldsymbol{\lambda}) \iff \min_{\boldsymbol{\lambda} \in \mathbb{R}^q} M_k(\boldsymbol{\lambda}).$$

Notably, the inner minimization in $M_k(\boldsymbol{\lambda})$ is convex for all $k$ in the Cressie-Read family. For $k < 0$, convexity follows from the map $t \mapsto t^{k/(k+1)}$, while for $k > 0$, the sign adjustment in the last row of Table 1 ensures convexity. In addition, Table 1 indicates that the proposed framework naturally incorporates both the empirical likelihood (EL) and the exponential tilting (ET) methods, thereby unifying a broad class of classical estimators. These properties are formalized in Lemmas A1-A2 of Appendix D: Lemma A1 establishes the EL and ET limits, and Lemma A2 proves the convexity of the inner dual.

Table 1: Inner objectives across the Cressie-Read family.

| Special Cases | $k$ | $H_k(\boldsymbol{\lambda})$ | $M_k(\boldsymbol{\lambda})$ |
|---|---|---|---|
| Empirical likelihood | $0$ | $\sum_i \log(1 + \boldsymbol{\lambda}^\top \boldsymbol{u}_i)$ | $-\sum_i \log(1 + \boldsymbol{\lambda}^\top \boldsymbol{u}_i)$ |
| Exponential tilting | $-1$ | $-N_{\mathrm{in}} \log\{\sum_i \exp(\boldsymbol{\lambda}^\top \boldsymbol{u}_i/2)/N_{\mathrm{in}}\}$ | $\sum_i \exp(\boldsymbol{\lambda}^\top \boldsymbol{u}_i/2)$ |
| GMM | $-2$ | $N_{\mathrm{in}}/2\{\sum_i (1 + \boldsymbol{\lambda}^\top \boldsymbol{u}_i)^2/N_{\mathrm{in}}\}^{-1}$ | $\sum_i (1 + \boldsymbol{\lambda}^\top \boldsymbol{u}_i)^2$ |
| Pearson/Neyman $\chi^2$ | $1$ | $N_{\mathrm{in}}/2\{\sum_i \sqrt{1 + \boldsymbol{\lambda}^\top \boldsymbol{u}_i}/N_{\mathrm{in}}\}^2$ | $-\sum_i \sqrt{1 + \boldsymbol{\lambda}^\top \boldsymbol{u}_i}$ |
| Freeman-Tukey | $-1/2$ | $-4N_{\mathrm{in}}\{\sum_i (1 + \boldsymbol{\lambda}^\top \boldsymbol{u}_i)^{-1}/N_{\mathrm{in}}\}^{1/2}$ | $\sum_i (1 + \boldsymbol{\lambda}^\top \boldsymbol{u}_i)^{-1}$ |
| General CR family | $k < 0$ | $N_{\mathrm{in}}/\{k(k+1)\}\{\sum_i (1 + \boldsymbol{\lambda}^\top \boldsymbol{u}_i)^{k/(k+1)}/N_{\mathrm{in}}\}^{k+1}$ | $\sum_i (1 + \boldsymbol{\lambda}^\top \boldsymbol{u}_i)^{k/(k+1)}$ |
| | $k > 0$ | (same as above) | $-\sum_i (1 + \boldsymbol{\lambda}^\top \boldsymbol{u}_i)^{k/(k+1)}$ |

Consequently, Eq. 1 provides a novel DRO framework for data integration with external summaries. It recovers classical estimators such as EL, ET, GMM, and $\chi^2$. More importantly, its scope is considerably broader: varying the Cressie-Read index $k$ can generate a continuum of penalty forms, each shaping the bias-variance-robustness trade-off in distinct ways. This flexibility is central to our proposed framework: while classical methods arise as fixed choices of $k$, our approach enables systematic exploration across a broad continuum of Cressie-Read families beyond the traditional cases.

### 2.1.2 ROBUSTNESS PROPERTIES AND INSIGHTS

We now turn to the theoretical properties of the proposed DRO estimator. In particular, we study its asymptotic behavior under varying levels of distributional shift.

**Theorem 2** (Asymptotic normality under no shift). *Assume the regularity conditions in Appendix E and the well-specified moments $\mathbb{E}\{\boldsymbol{u}_i(\boldsymbol{\beta}^\star)\} = \boldsymbol{0}$. Let $(\widehat{\boldsymbol{\beta}}_{\mathrm{DRO},k}, \widehat{\boldsymbol{\lambda}}_k)$ solve the first-order conditions of the DRO objective with Cressie-Read index $k$. We define $\ell_i(\boldsymbol{\beta}) := \log f_{\boldsymbol{\beta}}(Y_i \mid \boldsymbol{X}_i, \boldsymbol{Z}_i)$, then*

$$\sqrt{N_{\mathrm{in}}} \begin{pmatrix} \widehat{\boldsymbol{\beta}}_{\mathrm{DRO},k} - \boldsymbol{\beta}^\star \\ \widehat{\boldsymbol{\lambda}}_k \end{pmatrix} \xrightarrow{d} \mathcal{N}\left( \boldsymbol{0}, \begin{bmatrix} \mathbf{J} & -\mathbf{C}^\top \\ -\mathbf{C} & \boldsymbol{\Omega}_2 \end{bmatrix}^{-1} \begin{bmatrix} \mathbf{J} & \boldsymbol{0} \\ \boldsymbol{0} & \boldsymbol{\Omega}_1 \end{bmatrix} \begin{bmatrix} \mathbf{J} & -\mathbf{C}^\top \\ -\mathbf{C} & \boldsymbol{\Omega}_2 \end{bmatrix}^{-1} \right),$$

*where $\mathbf{J} := \mathbb{E}\{-\nabla^2_{\boldsymbol{\beta}\boldsymbol{\beta}}\ell_i(\boldsymbol{\beta}^\star)\}$, $\mathbf{C} := \mathbb{E}\{\nabla^2_{\boldsymbol{\lambda}\boldsymbol{\beta}}H_k(\boldsymbol{0}, \boldsymbol{\beta}^\star)\}$, $\boldsymbol{\Omega}_1 = \mathbb{E}\{\nabla_{\boldsymbol{\beta}}\boldsymbol{u}_i(\boldsymbol{\beta}^\star)\nabla_{\boldsymbol{\beta}}\boldsymbol{u}_i(\boldsymbol{\beta}^\star)^\top\}$ and $\boldsymbol{\Omega}_2 = -\mathbb{E}\{\nabla^2_{\boldsymbol{\lambda}\boldsymbol{\lambda}}H_k(\boldsymbol{0}, \boldsymbol{\beta}^\star)\}$. Profiling out $\boldsymbol{\lambda}$ yields*

$$\sqrt{N_{\mathrm{in}}}(\widehat{\boldsymbol{\beta}}_{\mathrm{DRO},k} - \boldsymbol{\beta}^\star) \xrightarrow{d} \mathcal{N}(\boldsymbol{0}, \ \mathbf{A}^{-1}\mathbf{V}\mathbf{A}^{-1}), \mathbf{A} := \mathbf{J} + \mathbf{C}^\top\boldsymbol{\Omega}_2^{-1}\mathbf{C}, \mathbf{V} := \mathbf{J} + \mathbf{C}^\top\boldsymbol{\Omega}_2^{-1}\boldsymbol{\Omega}_1\boldsymbol{\Omega}_2^{-1}\mathbf{C}.$$

*Denote $\mathbf{G} := \mathbb{E}\{\nabla_{\boldsymbol{\beta}}\boldsymbol{u}_i(\boldsymbol{\beta}^\star)\}$ and $\mathbf{S} := \mathbb{E}\{\boldsymbol{u}_i(\boldsymbol{\beta}^\star)\boldsymbol{u}_i(\boldsymbol{\beta}^\star)^\top\}$, for the Cressie-Read family, Lemmas A3-A4 show*

$$\mathbf{C} = \mathbf{G}/(k+1), \quad \boldsymbol{\Omega}_1 = \boldsymbol{\Omega}_2 = \mathbf{S}/(k+1)^2.$$

*Clearly, all $(k+1)$-factors cancel, so the first-order limit coincides with the efficient EL/ET/GMM variance under correct specification. Full Derivations are in Appendix E.*

*Remark* 1. Note that the asymptotic variance $(\mathbf{J}+\mathbf{G}^\top\mathbf{S}^{-1}\mathbf{G})^{-1}$ is no larger than $\mathbf{J}^{-1}$. Thus, relative to the "naive" estimator with variance $\mathbf{J}^{-1}$ that ignores the external summaries, the DRO estimator achieves strictly greater efficiency whenever $\mathbf{G} \neq \boldsymbol{0}$, under no distributional shift.

**Challenges under distributional shift.** Deriving the asymptotic distribution of the DRO estimator is considerably more involved under shift than in the no-shift case. The main difficulty is that the dual penalty $H_k(\boldsymbol{\lambda}, \boldsymbol{\beta})$ is not additively separable across samples: it is a nonlinear functional of the empirical mean of transformed scores. This prevents direct use of standard M-estimation theory. To proceed, Lemma A6 provides an influence-function expansion of $H_k$ that linearizes the non-additive structure and represents first-order fluctuations as sums of i.i.d. per-sample terms.

**Theorem 3** (Asymptotic normality under distributional shift). *Assume the regularity conditions in Appendix E. Let* $(\widehat{\boldsymbol{\beta}}_{\mathrm{DRO},k}, \widehat{\boldsymbol{\lambda}}_k)$ *solve the DRO estimating equations with Cressie-Read index* $k$, *and let* $\boldsymbol{\beta}^{\dagger}$ *denote the population target. Then*

$$\sqrt{N_{\mathrm{in}}}(\widehat{\boldsymbol{\beta}}_{\mathrm{DRO},k} - \boldsymbol{\beta}^{\dagger}) \xrightarrow{d} \mathcal{N}(\mathbf{0}, \mathbf{V}_{\beta}^{\dagger}),$$

*where* $\mathbf{V}_{\beta}^{\dagger}$ *is a* profile sandwich variance *of the form*

$$\mathbf{V}_{\beta}^{\dagger} = (\mathbf{A}_{\beta}^{\dagger})^{-1}\{\mathbf{K}^{\dagger} + (\mathbf{C}^{\dagger})^{\top}(\boldsymbol{\Omega}_2^{\dagger})^{-1}\boldsymbol{\Omega}_1^{\dagger}(\boldsymbol{\Omega}_2^{\dagger})^{-1}\mathbf{C}^{\dagger}$$
$$+ (\mathbf{C}^{\dagger})^{\top}(\boldsymbol{\Omega}_2^{\dagger})^{-1}\mathbf{K}_{sg}^{\dagger} + (\mathbf{K}_{sg}^{\dagger})^{\top}(\boldsymbol{\Omega}_2^{\dagger})^{-1}\mathbf{C}^{\dagger}\}(\mathbf{A}_{\beta}^{\dagger})^{-1}.$$

*Remark* 2. The variance expression reflects both the usual score variability and the additional contribution from the dual multipliers $\widehat{\boldsymbol{\lambda}}_k$, which capture the effect of distributional shift. All detailed definitions of the block matrices $\mathbf{A}_{\beta}^{\dagger}, \mathbf{K}^{\dagger}, \mathbf{C}^{\dagger}, \boldsymbol{\Omega}_1^{\dagger}, \boldsymbol{\Omega}_2^{\dagger}, \mathbf{K}_{sg}^{\dagger}$, together with the linearization of $H_k$ into summable influence functions, are provided in Lemma A6 and Theorem A6 of Appendix F.

The estimator $\widehat{\boldsymbol{\beta}}_{\mathrm{DRO},k}$ converges to a pseudo-true value $\boldsymbol{\beta}^{\dagger}$, which generally differs from $\boldsymbol{\beta}^{\star}$ under distributional shift. Lemma A7 in Appendix F gives that a local expansion around $\boldsymbol{\beta}^{\star}$:

$$\widehat{\boldsymbol{\beta}}_{\mathrm{DRO},k} - \boldsymbol{\beta}^{\star} = (\mathbf{J} - \mathbf{G}^{\top}\mathbf{S}^{-1}\mathbf{G})^{-1}\mathbf{G}^{\top}\mathbf{S}^{-1}\boldsymbol{\delta} + O_p(N_{\mathrm{in}}^{-1/2}) + o(\|\boldsymbol{\delta}\|),$$

with shift term $\boldsymbol{\delta} = \mathbb{E}\{\boldsymbol{u}_i(\boldsymbol{\beta}^{\star})\}$. Hence, under mild shifts ($\|\boldsymbol{\delta}\|$ small), the bias is negligible relative to sampling noise, while it dominates and persists asymptotically under large shifts. Importantly, the leading bias and variance are $k$-free, though $k$ affects higher-order terms and finite-sample behavior. Thus, DRO estimators are robust under small misspecification but deteriorate as shifts grow, motivating the stabilized EB-DRO method in Section 2.2.

### 2.1.3 ALGORITHMIC DETAILS

The DRO problem is solved by a nested optimization (Algorithm 1), with an outer update for $\boldsymbol{\beta}$ and an inner update for $\boldsymbol{\lambda}$. Both use Newton-Raphson steps scaled by the Hessian, with stability ensured via a backtracking line search: the update step is multiplied by a factor $\tau$, halved until the descent condition is satisfied. In the outer loop, the objective is $\ell_H(\boldsymbol{\beta}) = -\sum_i \log f_{\boldsymbol{\beta}}(Y_i \mid \mathbf{X}_i, \mathbf{Z}_i) + H_k\{\widehat{\boldsymbol{\lambda}}(\boldsymbol{\beta}), \boldsymbol{\beta}\}$, *implicit differentiation* is applied to incorporate the dependence of the inner optimizer $\widehat{\boldsymbol{\lambda}}(\boldsymbol{\beta})$ on $\boldsymbol{\beta}$. This approach extends the empirical likelihood algorithm of Han and Lawless (2019) to the full Cressie-Read family, providing a unified method for any divergence index $k$. Gradient/Hessian derivations and implicit differentiation details appear in Appendix H.

---

**Algorithm 1** Nested DRO solver (Cressie-Read index $k$) with Newton updates and backtracking

---

1: **Input:** initial $\boldsymbol{\beta}^{(0)}$; tolerances $(\varepsilon_{\mathrm{out}}, \varepsilon_{\mathrm{in}})$; max iters $(T_{\mathrm{out}}, T_{\mathrm{in}})$;
2: **for** $t = 0, 1, \ldots, T_{\mathrm{out}}$ **do**                                        # outer loop
3:     **Inner solve (given $\boldsymbol{\beta}^{(t)}$):** set $\boldsymbol{\lambda}^{(0)} = \mathbf{0}$
4:     **for** $s = 0, 1, \ldots, T_{\mathrm{in}}$ **do**                                    # inner loop
5:         **if** $\|\nabla_{\boldsymbol{\lambda}} M_k(\boldsymbol{\lambda}^{(s)}, \boldsymbol{\beta}^{(t)})\| \leq \varepsilon_{\mathrm{in}}$ **then break**
6:         **end if**
7:         Update $\boldsymbol{\lambda}^{(s+1)} \leftarrow \boldsymbol{\lambda}^{(s)} - \tau_{\mathrm{in}}\{\nabla_{\boldsymbol{\lambda}\boldsymbol{\lambda}}^2 M_k(\boldsymbol{\lambda}^{(s)}, \boldsymbol{\beta}^{(t)})\}^{-1}\nabla_{\boldsymbol{\lambda}} M_k(\boldsymbol{\lambda}^{(s)}, \boldsymbol{\beta}^{(t)})$ with backtracking ($\tau_{\mathrm{in}} \leftarrow \tau_{\mathrm{in}}/2$) until descent holds
8:     **end for**
9:     Set $\widehat{\boldsymbol{\lambda}}(\boldsymbol{\beta}^{(t)}) \leftarrow \boldsymbol{\lambda}^{(s)}$
10:    **if** $\|\nabla_{\boldsymbol{\beta}}\ell_H\{\boldsymbol{\beta}^{(t)}, \widehat{\boldsymbol{\lambda}}(\boldsymbol{\beta}^{(t)})\}\| \leq \varepsilon_{\mathrm{out}}$ **then break**
11:    **end if**
12:    Update $\boldsymbol{\beta}^{(t+1)} \leftarrow \boldsymbol{\beta}^{(t)} - \tau_{\mathrm{out}}[\nabla_{\boldsymbol{\beta}\boldsymbol{\beta}}^2\ell_H\{\boldsymbol{\beta}^{(t)}, \widehat{\boldsymbol{\lambda}}(\boldsymbol{\beta}^{(t)})\}]^{-1}\nabla_{\boldsymbol{\beta}}\ell_H\{\boldsymbol{\beta}^{(t)}, \widehat{\boldsymbol{\lambda}}(\boldsymbol{\beta}^{(t)})\}$ with backtracking ($\tau_{\mathrm{out}} \leftarrow \tau_{\mathrm{out}}/2$) until descent holds
13: **end for**
14: **Return:** $\widehat{\boldsymbol{\beta}}_{\mathrm{DRO},k} = \boldsymbol{\beta}^{(t)}, \widehat{\boldsymbol{\lambda}}_k = \boldsymbol{\lambda}^{(s)}$

---

## 2.2 EMPIRICAL BAYES-STABILIZED DRO (EB-DRO)

Although the DRO estimator $\widehat{\boldsymbol{\beta}}_{\mathrm{DRO},k}$ can substantially improve efficiency under mild distributional shift, its bias may grow quickly as the shift increases. Under severe shifts, $\widehat{\boldsymbol{\beta}}_{\mathrm{DRO},k}$ can even perform worse than the internal-only estimator $\widehat{\boldsymbol{\beta}}_I$, taken as the maximum likelihood estimator (MLE) from the GLM fit using the internal sample. To guard against this deterioration, we introduce an Empirical Bayes (EB) stabilization step: an adaptive shrinkage procedure that combines the DRO and naive estimators. This construction preserves DRO's efficiency gains under mild shifts while ensuring robustness under large shifts.

Let $\mathbf{Z} := \widehat{\boldsymbol{\beta}}_{\mathrm{DRO},k} - \widehat{\boldsymbol{\beta}}_I$, $\widehat{\mathbf{A}} := \mathbf{Z}\mathbf{Z}^\top$ and $\boldsymbol{\Sigma}_I := \mathrm{Var}(\widehat{\boldsymbol{\beta}}_I)$. The EB-DRO estimator is

$$\widehat{\boldsymbol{\beta}}_{\mathrm{EB},k} = (\mathbf{I} - \widehat{\mathbf{W}})\widehat{\boldsymbol{\beta}}_{\mathrm{DRO},k} + \widehat{\mathbf{W}}\widehat{\boldsymbol{\beta}}_I, \quad \widehat{\mathbf{W}} := \widehat{\mathbf{A}}(\widehat{\mathbf{A}} + \widehat{\boldsymbol{\Sigma}}_I)^{-1}.$$

This form arises naturally from a Gaussian hierarchical model in which $\widehat{\boldsymbol{\beta}}_{\mathrm{DRO},k}$ serves as the prior center (uncertainty $\widehat{\mathbf{A}}$) and $\widehat{\boldsymbol{\beta}}_I$ is modeled as a Gaussian observation with covariance $\widehat{\boldsymbol{\Sigma}}_I$, as detailed in Appendix I.1. Furthermore, the following proposition makes explicit the resulting shrinkage property, with proof given in Appendix I.2.

**Proposition 1.** *The EB-DRO estimator admits the equivalent representation*

$$\widehat{\boldsymbol{\beta}}_{\mathrm{EB},k} = \widehat{\boldsymbol{\beta}}_{\mathrm{DRO},k} + \alpha\mathbf{Z} = (1-\alpha)\widehat{\boldsymbol{\beta}}_{\mathrm{DRO},k} + \alpha\widehat{\boldsymbol{\beta}}_I, \ \alpha = \mathbf{Z}^\top\widehat{\boldsymbol{\Sigma}}_I^{-1}\mathbf{Z}/(1 + \mathbf{Z}^\top\widehat{\boldsymbol{\Sigma}}_I^{-1}\mathbf{Z}) \in (0,1). \quad (3)$$

This proposition shows that EB-DRO adaptively interpolates between the DRO and naive estimators depending on their Mahalanobis discrepancy. When $\widehat{\boldsymbol{\beta}}_{\mathrm{DRO},k}$ and $\widehat{\boldsymbol{\beta}}_I$ are close, $\alpha \approx 0$ and EB-DRO coincides with DRO, preserving efficiency. When they diverge, $\alpha \to 1$ and EB-DRO reverts to the naive estimator, ensuring robustness.

**Theorem 4** (Oracle risk guarantee and regret). *For $w \in [0,1]$, define*

$$\widehat{\boldsymbol{\beta}}(w) = (1-w)\widehat{\boldsymbol{\beta}}_{\mathrm{DRO},k} + w\widehat{\boldsymbol{\beta}}_I, \quad R(w) = \mathbb{E}\|\widehat{\boldsymbol{\beta}}(w) - \boldsymbol{\beta}^\star\|_2^2.$$

*Let $\Pi_{[0,1]}(t) = \min\{1, \max\{0, t\}\}$. The oracle weight*

$$w^\star = \Pi_{[0,1]}\{a - m/(a + c - 2m)\}, \quad a = \|\boldsymbol{b}_k\|_2^2 + \mathrm{tr}(\mathbf{V}_{\mathrm{DRO},k}), \ c = \mathrm{tr}(\boldsymbol{\Sigma}_I), \ m = \mathrm{tr}(\mathbf{C}_k),$$

*with $\boldsymbol{b}_k = \mathbb{E}(\widehat{\boldsymbol{\beta}}_{\mathrm{DRO},k}) - \boldsymbol{\beta}^\star$, $\mathbf{V}_{\mathrm{DRO},k} = \mathrm{Var}(\widehat{\boldsymbol{\beta}}_{\mathrm{DRO},k})$ and $\mathbf{C}_k = \mathrm{Cov}(\widehat{\boldsymbol{\beta}}_{\mathrm{DRO},k}, \widehat{\boldsymbol{\beta}}_I)$, minimizes $R(w)$ and satisfies*

$$R(w^\star) \leq \min\{R(0), R(1)\}.$$

*Hence the oracle EB-DRO is never worse than DRO or naive. Moreover, for any $w \in [0,1]$,*

$$R(w) = R(w^\star) + (a + c - 2m)(w - w^\star)^2,$$

*giving an exact quadratic regret identity.*

*Remark* 3. The EB-DRO estimator uses the data-adaptive weight $\alpha$ from Eq. 3. Although $\alpha \neq w^\star$ in general, Appendix I.3 shows that $\alpha \xrightarrow{p} \alpha_0$, where $\alpha_0 = \boldsymbol{b}_k^\top\boldsymbol{\Sigma}_I^{-1}\boldsymbol{b}_k/(1 + \boldsymbol{b}_k^\top\boldsymbol{\Sigma}_I^{-1}\boldsymbol{b}_k)$. In the extreme regimes of no shift or large shift, $\alpha_0 = w^\star$, so EB-DRO inherits the oracle guarantee asymptotically. Moreover, Theorem 4 implies that any plug-in weight (including $\alpha$) incurs at most a quadratic excess-risk term relative to $w^\star$, which vanishes asymptotically. This motivates our ensemble EB-DRO strategy: by aggregating across divergence indices $k$, we further dampen plug-in variability and stabilize performance across a wider range of shifts.

## 2.3 ENSEMBLE EB-DRO

Different divergence indices $k$ yield EB-DRO estimators $\{\widehat{\boldsymbol{\beta}}_{\mathrm{EB},k} : k \in \mathcal{K}\}$, each associated with a posterior covariance

$$\widehat{\boldsymbol{\Omega}}_k = (\widehat{\boldsymbol{\Sigma}}_I^{-1} + \widehat{\mathbf{A}}_k^+)^{-1},$$

where $\widehat{\mathbf{A}}_k^+$ denotes the Moore-Penrose pseudoinverse of $\widehat{\mathbf{A}}_k = (\widehat{\boldsymbol{\beta}}_{\mathrm{DRO},k} - \widehat{\boldsymbol{\beta}}_I)(\widehat{\boldsymbol{\beta}}_{\mathrm{DRO},k} - \widehat{\boldsymbol{\beta}}_I)^\top$.

To stabilize performance across various shifts, we aggregate the individual EB-DRO estimators into a precision-weighted ensemble:

$$\widehat{\boldsymbol{\beta}}_{\mathrm{Ens}} = \left\{ \sum\nolimits_{k \in \mathcal{K}} (\widehat{\boldsymbol{\Omega}}_k)^{-1} \right\}^{-1} \left\{ \sum\nolimits_{k \in \mathcal{K}} (\widehat{\boldsymbol{\Omega}}_k)^{-1} \widehat{\boldsymbol{\beta}}_{\mathrm{EB},k} \right\}, \tag{4}$$

This ensemble can be interpreted as the posterior mean of a higher-level Gaussian model in which $\{\widehat{\boldsymbol{\beta}}_{\mathrm{EB},k}\}_{k \in \mathcal{K}}$ serve as complementary noisy signals about the same target.

**Theorem 5** (Stability of ensemble EB-DRO). *Let $\boldsymbol{\Omega}_{k\ell} = \mathrm{Cov}(\widehat{\boldsymbol{\beta}}_{\mathrm{EB},k}, \widehat{\boldsymbol{\beta}}_{\mathrm{EB},\ell})$ and assume $\widehat{\boldsymbol{\Omega}}_k \xrightarrow{p}$ $\boldsymbol{\Omega}_{kk}$ for all $k$. Then $\mathbb{E}\|\widehat{\boldsymbol{\beta}}_{\mathrm{Ens}} - \boldsymbol{\beta}^\star\|_2^2 \leq \min_{j \in \mathcal{K}} \mathbb{E}\|\widehat{\boldsymbol{\beta}}_{\mathrm{EB},j} - \boldsymbol{\beta}^\star\|_2^2 + \Delta_{\mathrm{cross}} + o(1)$, where $\Delta_{\mathrm{cross}}$ depends only on the off-diagonal cross-covariances $\{\boldsymbol{\Omega}_{k\ell}\}_{k \neq \ell}$ (explicit form in Appendix J).*

Theorem 5 shows that the precision-weighted ensemble achieves mean squared error no larger than the best EB-DRO up to a correlation penalty $\Delta_{\mathrm{cross}}$. Individual EB-DROs can be unstable: plug-in variability in $\alpha$ (Eq. 3) may underweight the naive estimator, leading to inflated risk and sometimes performing worse than naive. The ensemble mitigates this by averaging across $k$, dampening variability and pulling risk back toward a safe baseline, thereby ensuring stability across shift regimes.

**Summary.** We introduced three estimators: (i) *DRO*, which leverages external information to improve efficiency under mild shift but may suffer bias under severe shift; (ii) *EB-DRO*, which adaptively shrinks between DRO and the internal estimator to stabilize performance; and (iii) the *ensemble EB-DRO*, a precision-weighted aggregation across divergences $k$ that dampens plug-in variability and ensures robustness across a wider range of shifts. We next investigate their empirical behavior through simulations.

## 3 SIMULATION EXPERIMENTS

### 3.1 SIMULATION SETUP

We simulate binary outcomes from a logistic GLM. Let $\mathbf{X} = (1, X_1, \ldots, X_5)^\top$ include an intercept and five covariates. The true coefficient vector is $\boldsymbol{\beta}^\star = (1, -1, -1, 1, -1, -1)^\top = (\beta_0^\star, \beta_1^\star, \ldots, \beta_5^\star)^\top \in \mathbb{R}^6$, where $\beta_0^\star$ is the intercept. Internal outcomes ($N_{\mathrm{in}} = 500$) follow

$$Y_{\mathrm{in}} \mid \mathbf{X} \sim \mathrm{Bernoulli}\{\mathrm{expit}(\mathbf{X}^\top \boldsymbol{\beta}^\star)\}, \quad \mathrm{expit}(t) = \frac{1}{1+e^{-t}}.$$

External outcomes ($N_{\mathrm{ext}} = 100{,}000$) are generated under a *shifted* coefficient $\boldsymbol{\beta}_{\mathrm{ext}}$, defined as a perturbation of $\boldsymbol{\beta}^\star$. The external summaries $\boldsymbol{\theta}$ are obtained by fitting a *reduced* GLM with covariates $(1, X_1, X_2, X_3)^\top \in \mathbb{R}^4$ to the external sample.

**Shift scenarios.** We consider three families of distributional shift:

(A) **Intercept shift:** Replace $\beta_0^\star$ with $\beta_0^\star + \Delta$, $\Delta \in \{0, 0.1, \ldots, 1.0\}$ (11 settings), keeping slopes fixed. *Intuition:* Change baseline prevalence without altering covariate effects.

(B) **Tail perturbation:** For a fraction $p \in \{0, 0.05, \ldots, 1.0\}$ (21 settings) of external units in the upper tail of $X_1$, reduce $(\beta_1^\star, \beta_2^\star, \beta_3^\star)$ by 1.5; the remainder use $\boldsymbol{\beta}^\star$. *Intuition:* Induces a localized, subgroup-specific shift concentrated in high-$X_1$ regions.

(C) **Angle (directional) shift:** Rotate true $\boldsymbol{\beta}^\star$ toward a fixed orthogonal direction $\boldsymbol{v}$ via Gram-Schmidt by an angle $\varphi \in \{0, 0.0776, \ldots, 0.6981\}$ (10 values, up to $\approx 40°$), i.e., $\boldsymbol{\beta}_{\mathrm{ext}}(\varphi) = \cos(\varphi)\boldsymbol{\beta}^\star + \sin(\varphi)\boldsymbol{v}$. *Intuition:* Induces a global misspecification that changes the direction of the signal while preserving its magnitude.

**Estimators and evaluation.** At each shift setting, we generate one external dataset ($N_{\mathrm{ext}} = 100{,}000$) for $\widehat{\boldsymbol{\theta}}$ and repeatedly draw internal datasets ($N_{\mathrm{in}} = 500$, $R = 1000$ replicates). On each replicate we compute: (i) the internal MLE $\widehat{\boldsymbol{\beta}}_I$ (naive); (ii) DRO estimators $\widehat{\boldsymbol{\beta}}_{\mathrm{DRO},k}$ across 19 divergence indices $k$; (iii) EB-DRO $\widehat{\boldsymbol{\beta}}_{\mathrm{EB},k}$ for the same set of $k$ values; and (iv) the ensemble $\widehat{\boldsymbol{\beta}}_{\mathrm{Ens}}$. Performance is measured by mean $\ell_2$ error $\mathbb{E}\|\widehat{\boldsymbol{\beta}} - \boldsymbol{\beta}^\star\|_2$, averaged over replicates.

### 3.2 SIMULATION RESULTS

We evaluate DRO and EB-DRO estimators across 19 divergence indices $k$ (values shown in the legends of Figs. 1-2). Specially, Fig. 1 shows that robustness varies strongly with $k$: some families

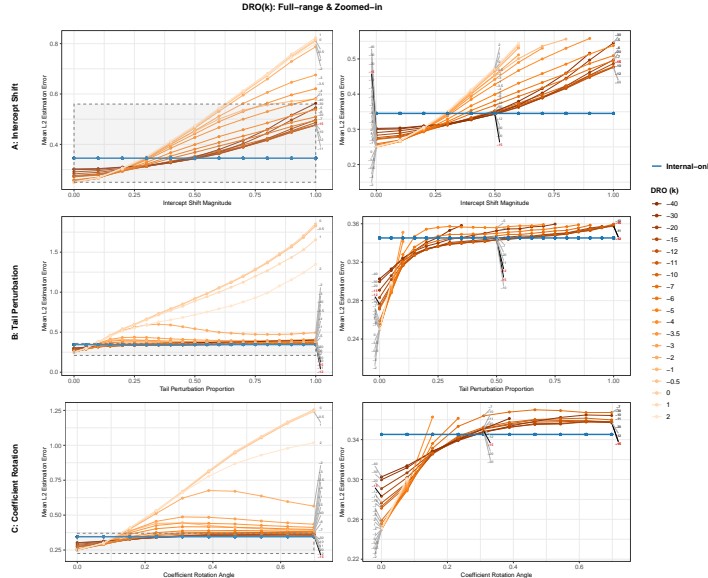

Figure 1: Performance of DRO across 19 divergence families. Left: full range of results. Right: zoomed-in view of the shaded region to highlight finer differences. Internal-only estimates are included for reference. *Rows:* (A) intercept shift, (B) tail perturbation, and (C) coefficient rotation.

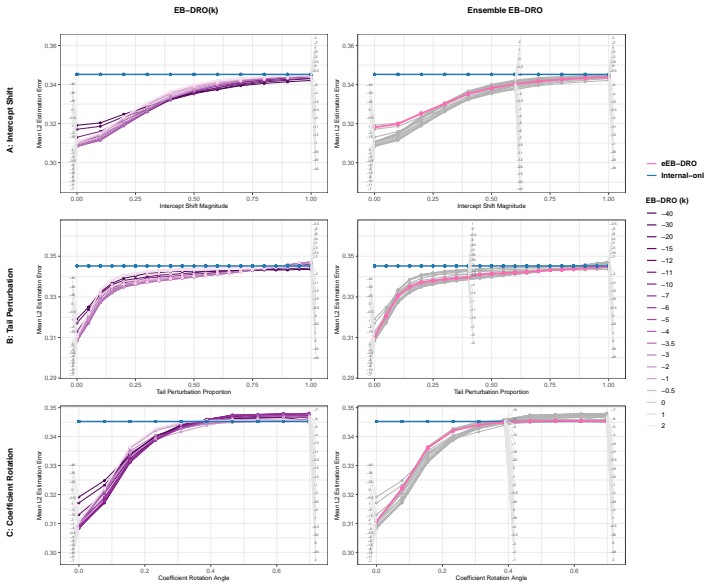

Figure 2: Performance of EB-DRO($k$) and ensemble EB-DRO. Left: EB-DRO($k$) for each family (colored). Right: per-family EB-DRO($k$) (gray) with ensemble EB-DRO (pink) overlaid for comparison. Internal-only estimates are also shown.

(e.g., $k = -15$) remain competitive with or better than the naive estimator even under moderately large shifts, while others deteriorate quickly. This illustrates the $k$-dependent trade-off between bias, variance, and robustness: certain divergences yield stable performance across shifts, whereas others overshoot once bias accumulates. The pattern is consistent with Theorem 2 and Theorem 3, which shows DRO improves efficiency under mild shifts but becomes biased under severe ones. Exploring the full continuum of Cressie-Read divergences therefore extends beyond classical fixed cases (EL, ET, GMM, $\chi^2$) to reveal a richer spectrum of robustness behaviors.

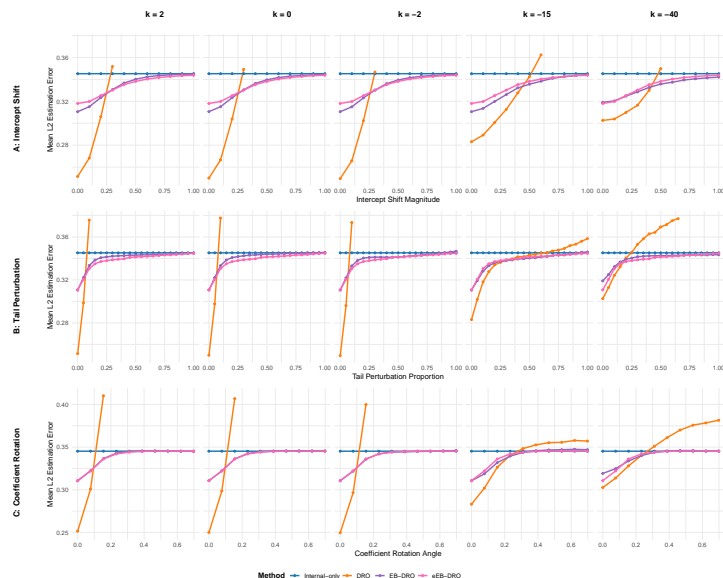

Figure 3: Performance of internal-only, DRO, EB-DRO($k$), and ensemble EB-DRO for selected families ($k = 2, 0, -2, -15, -40$). Results for all 19 families are given in Figs A2-A4.

Fig. 2 dispalys that EB-DRO markedly improves over DRO, remaining close to or better than the naive estimator across most settings. In extreme cases (e.g., angle shifts $> 0.4$ or tail perturbations $> 0.9$), some EB-DRO($k$) curves become slightly worse than naive due to noisy plug-in weights. Notably, the ensemble consistently pulls results back toward the naive baseline under severe shifts, highlighting its stability.

Fig. 3 directly compares internal-only, DRO, EB-DRO, and the ensemble. The pattern is clear: DRO can improve efficiency over naive under mild shifts but becomes unstable under severe ones; EB-DRO stabilizes performance by adaptively shrinking toward naive; and the ensemble consolidates these gains, avoiding DRO's blow-ups while preserving EB-DRO's efficiency under small shifts. These findings confirm the theoretical guarantees developed in Section 2.

## 4 DISCUSSION

We proposed a distributionally robust framework for integrating internal individual-level data with external summaries, introducing three estimators: (i) *DRO*, which improves efficiency under mild shifts but suffers bias under severe ones; (ii) *EB-DRO*, which stabilizes performance by adaptively shrinking between DRO and the internal estimator; and (iii) the *ensemble EB-DRO*, which aggregates across divergences via precision weighting to dampen plug-in variability.

This framework also unifies and extends classical estimators: the duality links DRO to the Cressie-Read family, recovering empirical likelihood, exponential tilting, GMM, and $\chi^2$ as special cases. EB-DRO parallels empirical Bayes shrinkage, offering protection against noisy summaries, and the ensemble inherits the strengths of well-performing divergences while avoiding catastrophic failures.

**Future directions.** Several avenues remain open. First, extending the theory to high-dimensional covariates would broaden applicability. Second, exploring adaptive weighting schemes beyond precision weighting could further enhance stability. Third, combining our framework with conformal or post-selection inference may yield valid uncertainty quantification under severe distributional shifts. Finally, applying the framework to large-scale biomedical studies, such as those involving epigenetic profiling, where detailed measurements are rarely available in large cohorts, will be crucial for demonstrating practical impact.

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

## APPENDIX A    SUPPLEMENTARY FIGURES

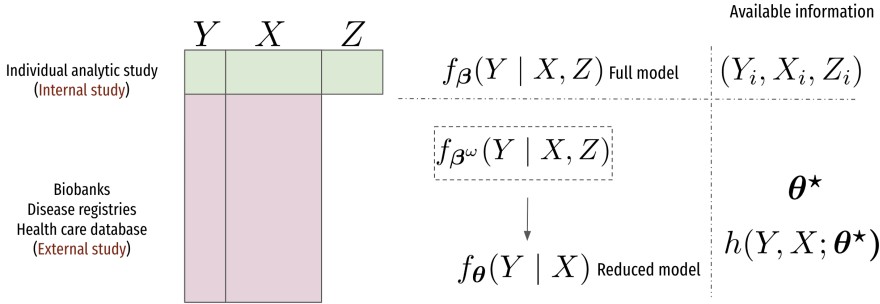

Figure A1: Problem Setup

## APPENDIX B    PROOF OF THEOREM 1 (VARIATIONAL DUAL FORMULATION)

*Proof.* For a given $\boldsymbol{\beta}$, the inner problem in Eq. 1 over reweighting $p$ is

$$\min_{p \geq 0 \in \mathbb{R}^N} \frac{1}{N} \sum_{i=1}^N \phi(Np_i)$$

$$s.t. \sum_{i=1}^N p_i = 1, \quad \sum_{i=1}^N p_i \boldsymbol{u}_i(\boldsymbol{\beta}) = \mathbf{0},$$

where $N := N_{\text{in}}$. Introduce Lagrange multipliers $\gamma \in \mathbb{R}$ (normalization) and $\boldsymbol{\lambda} \in \mathbb{R}^q$ (moments), and set $s_i := Np_i \geq 0$. The Lagrangian is

$$\mathcal{L}(s, \gamma, \boldsymbol{\lambda}) = \frac{1}{N} \sum_{i=1}^N \phi(s_i) + \frac{1}{N} \sum_{i=1}^N s_i \{\gamma + \boldsymbol{\lambda}^\top \boldsymbol{u}_i(\boldsymbol{\beta})\} - \gamma.$$

The dual function is the infimum over $s_i \geq 0$ (note separability in $i$):

$$g(\gamma, \boldsymbol{\lambda}) = \sum_{i=1}^N \inf_{s_i \geq 0} \frac{1}{N} \{\phi(s_i) + s_i t_i\} - \gamma, \quad t_i := \gamma + \boldsymbol{\lambda}^\top \boldsymbol{u}_i(\boldsymbol{\beta}).$$

By the definition of the convex conjugate, $\inf_{s \geq 0}\{\phi(s)+st\} = -\phi^*(-t)$ (the domain restriction $s \geq 0$ is standard for $\phi$-divergences and yields the same value for $t$ in the effective domain). Therefore,

$$g(\gamma, \boldsymbol{\lambda}) = -\frac{1}{N} \sum_{i=1}^N \phi^*(-t_i) - \gamma = -\frac{1}{N} \sum_{i=1}^N \phi^* \{-\gamma - \boldsymbol{\lambda}^\top \boldsymbol{u}_i(\boldsymbol{\beta})\} - \gamma.$$

Maximizing $g$ over $(\gamma, \boldsymbol{\lambda})$ is equivalent by the sign change $(\gamma, \boldsymbol{\lambda}) \mapsto (-\gamma, -\boldsymbol{\lambda})$ to

$$\max_{\gamma, \boldsymbol{\lambda}} \gamma - \frac{1}{N} \sum_{i=1}^N \phi^* \{\gamma + \boldsymbol{\lambda}^\top \boldsymbol{u}_i(\boldsymbol{\beta})\}.$$

By strong duality (Slater's condition holds whenever there exists a strictly feasible $p$ with $p_i > 0$, $\sum p_i = 1$, and $\sum p_i \boldsymbol{u}_i(\boldsymbol{\beta}) = \mathbf{0}$), the inner primal value equals this dual value. Since the negative log-likelihood term

$$-\frac{2}{N} \sum_{i=1}^N \log f_{\boldsymbol{\beta}}(Y_i \mid \mathbf{X}_i, \mathbf{Z}_i)$$

does not depend on $p$, it passes unchanged through the dualization. Taking the outer minimization over $\boldsymbol{\beta}$ completes the proof. □

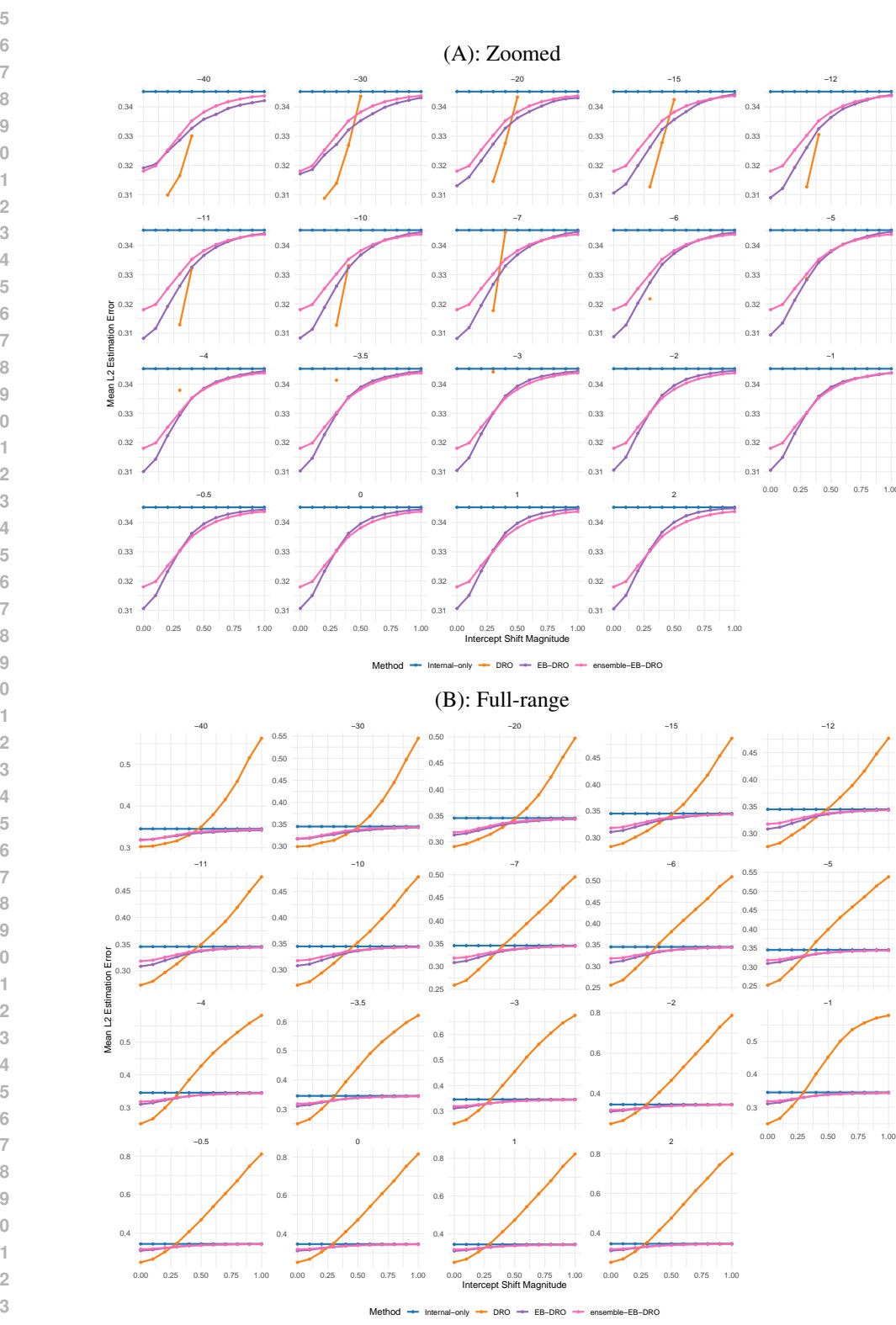

Figure A2: Theme A: Intercept shift magnitude

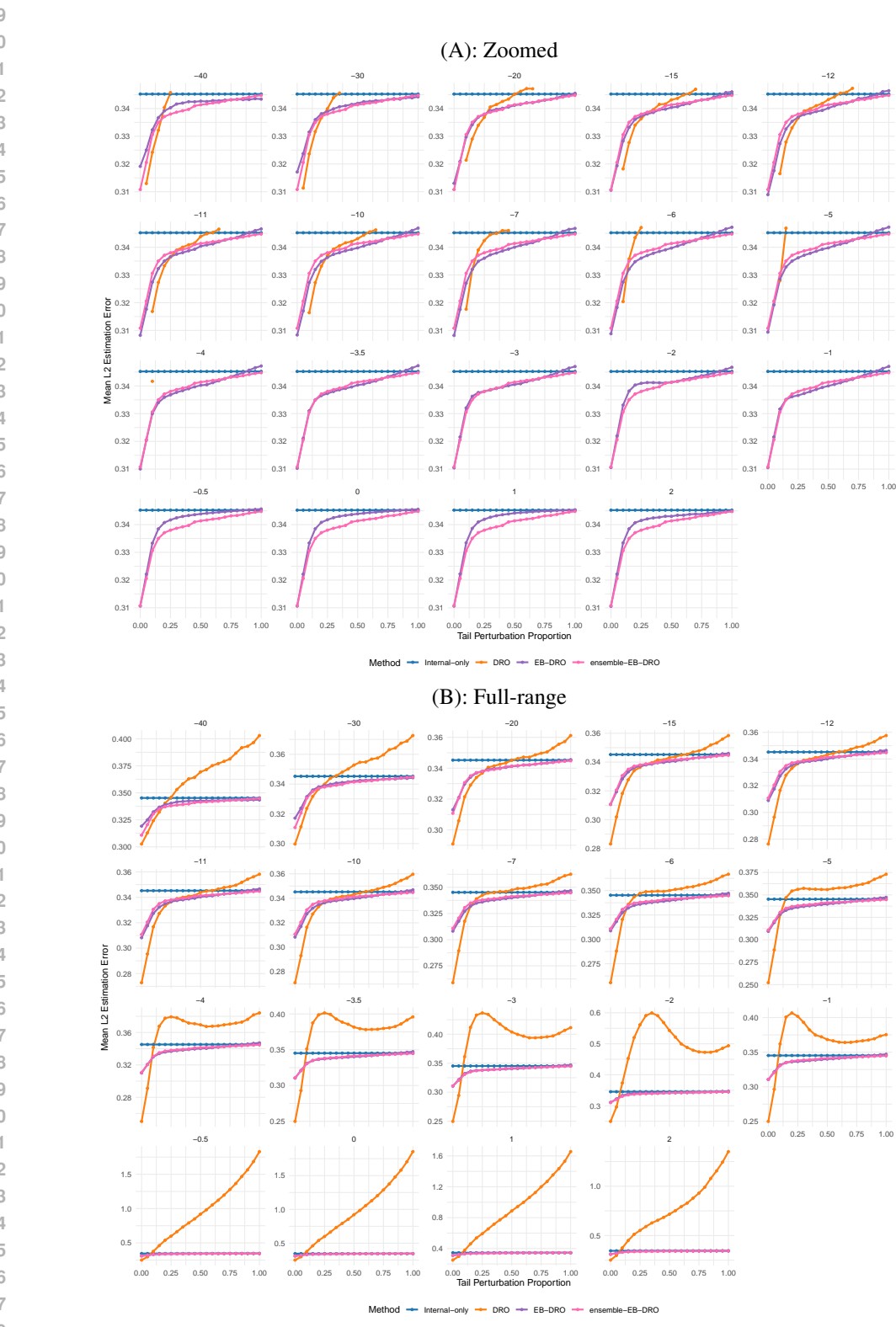

Figure A3: Theme B: Tail perturbation

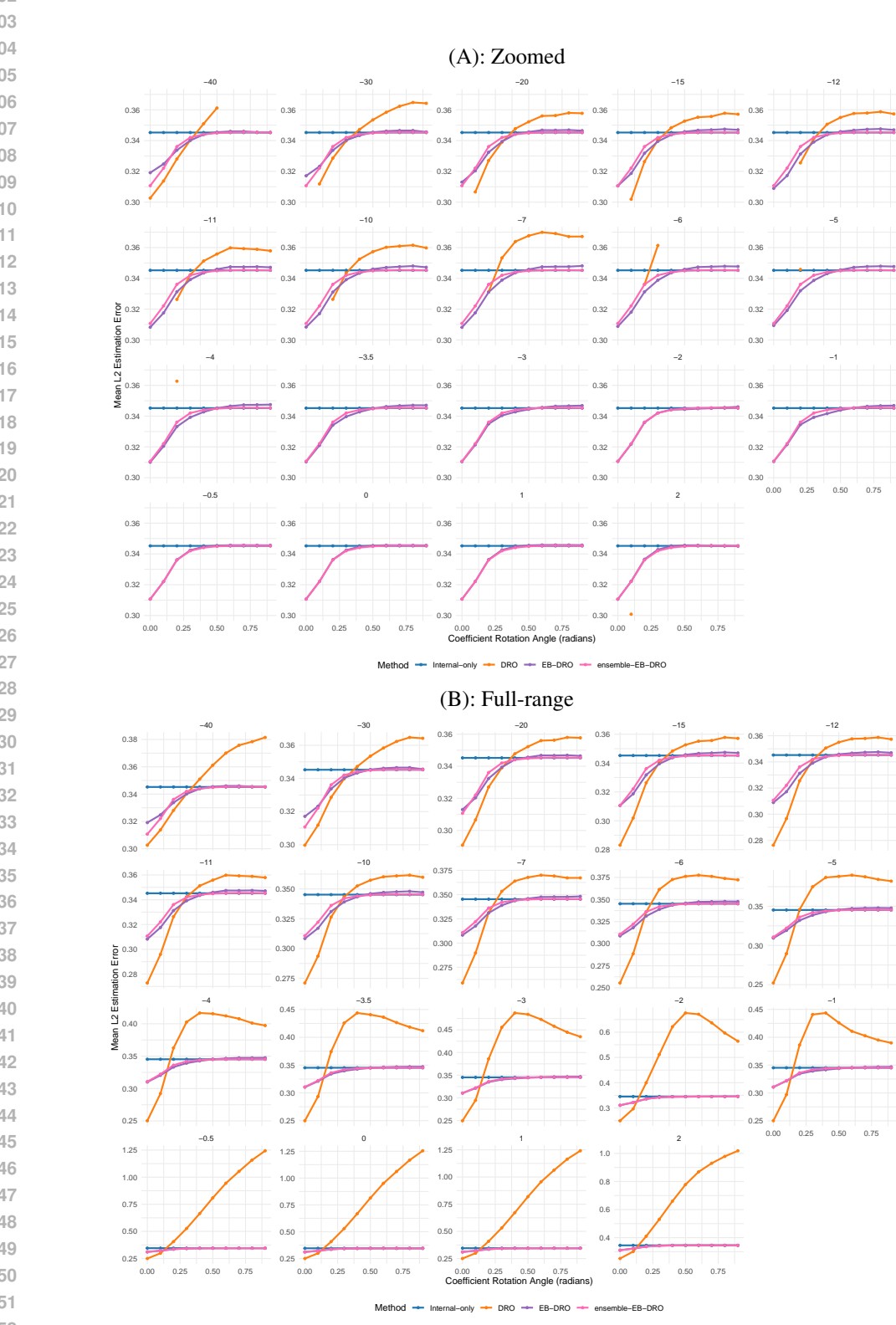

Figure A4: Theme C: Angle shift of the whole vector

## APPENDIX C   PROOF OF COROLLARY 1

*Proof.* By Theorem 1, for fixed $(\boldsymbol{\beta}, \boldsymbol{\lambda})$ the inner dual in $\gamma \in \mathbb{R}$ is

$$\max_{\gamma \in \mathbb{R}} \ \gamma - \frac{1}{N} \sum_{i=1}^{N} \phi_k^* \{\gamma + \boldsymbol{\lambda}^\top \boldsymbol{u}_i(\boldsymbol{\beta})\}, \quad N := N_{\text{in}}.$$

Set $t_i := 1 + \boldsymbol{\lambda}^\top \boldsymbol{u}_i(\boldsymbol{\beta})$ and assume $t_i > 0$ (the natural domain for CR). With the given $\phi_k^*$, the objective equals (constants in $\phi_k^*$ cancel when summed and absorbed)

$$g(\gamma) = \gamma + \frac{2}{kN} \sum_{i=1}^{N} [-\tfrac{k+1}{2} \{\gamma + (t_i - 1)\}]^{\frac{k}{k+1}}.$$

Introduce $\alpha := -\frac{k+1}{2}\gamma$ $(\alpha \geq 0 \Leftrightarrow \gamma \leq 0)$ so that $-\frac{k+1}{2}\{\gamma + (t_i - 1)\} = \alpha t_i$ and $\gamma = -\frac{2}{k+1}\alpha$. Then

$$g(\alpha) = -\frac{2}{k+1}\alpha + \frac{2}{kN} \sum_{i=1}^{N} (\alpha t_i)^{\frac{k}{k+1}} = -\frac{2}{k+1}\alpha + \frac{2}{k}\alpha^{\frac{k}{k+1}} \left( \frac{1}{N} \sum_{i=1}^{N} t_i^{\frac{k}{k+1}} \right).$$

Since $\frac{k}{k+1} \in (0,1)$ for $k > -1$, the map $\alpha \mapsto g(\alpha)$ is strictly concave on $\alpha \geq 0$, so the maximizer is characterized by the first-order condition:

$$g'(\alpha) = -\frac{2}{k+1} + \frac{2}{k} \cdot \frac{k}{k+1} \alpha^{-\frac{1}{k+1}} \left( \frac{1}{N} \sum_{i=1}^{N} t_i^{\frac{k}{k+1}} \right) = 0.$$

Solving gives

$$\alpha^* = \left( \frac{1}{N} \sum_{i=1}^{N} t_i^{\frac{k}{k+1}} \right)^{k+1} \implies \gamma^* = -\frac{2}{k+1} \left( \frac{1}{N} \sum_{i=1}^{N} t_i^{\frac{k}{k+1}} \right)^{k+1}.$$

Evaluating $g(\alpha^*)$ yields

$$g(\alpha^*) = \frac{2}{k} \left( \frac{1}{N} \sum_{i=1}^{N} t_i^{\frac{k}{k+1}} \right)^{k+1} - \frac{2}{k+1} \left( \frac{1}{N} \sum_{i=1}^{N} t_i^{\frac{k}{k+1}} \right)^{k+1} = \frac{2}{k(k+1)} \left( \frac{1}{N} \sum_{i=1}^{N} t_i^{\frac{k}{k+1}} \right)^{k+1}.$$

Multiplying by $N$ (recall the prefactor $1/N$ sits in front of the sum inside the dual) gives

$$H_k(\boldsymbol{\lambda}, \boldsymbol{\beta}) = \frac{N}{k(k+1)} \left\{ \frac{1}{N} \sum_{i=1}^{N} \left(1 + \boldsymbol{\lambda}^\top \boldsymbol{u}_i(\boldsymbol{\beta})\right)^{\frac{k}{k+1}} \right\}^{k+1},$$

which depends only on $(\boldsymbol{\lambda}, \boldsymbol{\beta})$. Plugging back into the outer problem (with the likelihood term independent of $\gamma$) gives the stated min-max form. $\qquad \square$

## APPENDIX D   PROPERTIES OF THE INNER DRO PENALTY

### D.1   SPECIAL CASES

**Lemma A1** (Special cases of $k \to 0$ and $k \to -1$ limits). *Let*

$$H_k(\boldsymbol{\lambda}, \boldsymbol{\beta}) = \frac{N_{\text{in}}}{k(k+1)} \left\{ \frac{1}{N_{\text{in}}} \sum_{i=1}^{N_{\text{in}}} \left(1 + \boldsymbol{\lambda}^\top \boldsymbol{u}_i(\boldsymbol{\beta})\right)^{k/(k+1)} \right\}^{k+1},$$

*with $t_i(\boldsymbol{\lambda}, \boldsymbol{\beta}) = 1 + \boldsymbol{\lambda}^\top \boldsymbol{u}_i(\boldsymbol{\beta}) > 0$. Then:*

    *1. As $k \to 0$,*

$$\lim_{k \to 0} \left( H_k(\boldsymbol{\lambda}, \boldsymbol{\beta}) - \tfrac{N_{\text{in}}}{k(k+1)} \right) = \sum_{i=1}^{N_{\text{in}}} \log \left(1 + \boldsymbol{\lambda}^\top \boldsymbol{u}_i(\boldsymbol{\beta})\right),$$

    *which coincides with the empirical likelihood (EL) dual objective.*

2. *As $k \to -1$,*

$$\lim_{k \to -1} \left( H_k(\boldsymbol{\lambda}, \boldsymbol{\beta}) - \tfrac{N_{\text{in}}}{k(k+1)} \right) = -N_{\text{in}} \log \left( \frac{1}{N_{\text{in}}} \sum_{i=1}^{N_{\text{in}}} \exp \left( \tfrac{1}{2} \boldsymbol{\theta}^\top \boldsymbol{u}_i(\boldsymbol{\beta}) \right) \right),$$

*where $\boldsymbol{\theta}$ is a reparameterization of $\boldsymbol{\lambda}$. This corresponds to the exponential tilting (ET/KL) dual form.*

*Proof.* For the EL case $k \to 0$ (hence $\alpha = \frac{k}{k+1} \to 0$), expand

$$t_i^{k/(k+1)} = \exp \left( \tfrac{k}{k+1} \log t_i \right) = 1 + \tfrac{k}{k+1} \log t_i + O(k^2),$$

where $t_i = 1 + \boldsymbol{\lambda}^\top \boldsymbol{u}_i(\beta)$. Averaging gives

$$\frac{1}{N_{\text{in}}} \sum_{i=1}^{N_{\text{in}}} t_i^{k/(k+1)} = 1 + \tfrac{k}{k+1} L(\boldsymbol{\lambda}, \boldsymbol{\beta}) + O(k^2), \quad L(\boldsymbol{\lambda}, \boldsymbol{\beta}) = \tfrac{1}{N_{\text{in}}} \sum_{i=1}^{N_{\text{in}}} \log t_i.$$

Write $A_k := \frac{1}{N_{\text{in}}} \sum_{i=1}^{N_{\text{in}}} t_i^{k/(k+1)}$, and $\delta_k := \tfrac{k}{k+1} L(\boldsymbol{\lambda}, \boldsymbol{\beta}) + O(k^2)$. Then

$$A_k^{k+1} = \exp \left( (k+1) \log(1 + \delta_k) \right).$$

Since $\log(1 + \delta_k) = \delta_k + O(\delta_k^2)$ with $\delta_k = O(k)$,

$$(k+1) \log(1 + \delta_k) = k L(\boldsymbol{\lambda}, \boldsymbol{\beta}) + O(k^2).$$

Exponentiating gives

$$A_k^{k+1} = \exp \left( k L(\boldsymbol{\lambda}, \boldsymbol{\beta}) + O(k^2) \right) = 1 + k L(\boldsymbol{\lambda}, \boldsymbol{\beta}) + O(k^2).$$

Substituting into $H_k$ gives

$$H_k(\lambda, \beta) = \frac{N_{\text{in}}}{k(k+1)} \left( 1 + k L(\boldsymbol{\lambda}, \boldsymbol{\beta}) + O(k^2) \right) = \frac{N_{\text{in}}}{k(k+1)} + \frac{N_{\text{in}}}{k+1} L(\boldsymbol{\lambda}, \boldsymbol{\beta}) + \frac{N_{\text{in}}}{k(k+1)} O(k^2)$$

where the last equality follows because $\frac{1}{k(k+1)} = O(1/k)$ as $k \to 0$ (indeed, $\frac{1}{k+1} = 1 + O(k)$), hence $\frac{1}{k(k+1)} O(k^2) = O(k)$.

and thus

$$\lim_{k \to 0} \left( H_k(\boldsymbol{\lambda}, \beta) - \tfrac{N_{\text{in}}}{k(k+1)} \right) = \sum_{i=1}^{N_{\text{in}}} \log \left( 1 + \boldsymbol{\lambda}^\top \boldsymbol{u}_i(\beta) \right).$$

For the ET case $k \to -1$, set $r = k + 1 \to 0$ so that $\alpha = k/(k+1) = 1 - 1/r$. Reparameterize $\boldsymbol{\lambda} = -\tfrac{r}{2} \boldsymbol{\vartheta}$ with $r = k+1 \to 0$. This scaling has two purposes: first, since $t_i(\boldsymbol{\lambda}, \beta) = 1 + \boldsymbol{\lambda}^\top \boldsymbol{u}_i(\beta)$, it guarantees $t_i(\boldsymbol{\lambda}, \beta) = 1 + O(r) > 0$ for sufficiently small $r$, thus preserving the positivity constraint; second, the factor $-\tfrac{1}{2}$ is chosen so that the expansion of $\alpha \log t_i$ with $\alpha = 1 - \tfrac{1}{r}$ yields the exponential tilt form $\exp \left( \tfrac{1}{2} \boldsymbol{\vartheta}^\top \boldsymbol{u}_i(\beta) \right)$ in the limit. A Taylor expansion shows

$$t_i^\alpha = \exp \left( \tfrac{1}{2} \boldsymbol{\vartheta}^\top \boldsymbol{u}_i(\beta) \right) (1 + O(r)).$$

Hence

$$\frac{1}{N_{\text{in}}} \sum_{i=1}^{N_{\text{in}}} t_i^\alpha = \frac{1}{N_{\text{in}}} \sum_{i=1}^{N_{\text{in}}} \exp \left( \tfrac{1}{2} \boldsymbol{\vartheta}^\top \boldsymbol{u}_i(\beta) \right) + O(r).$$

Set

$$M_k := \frac{1}{N_{\text{in}}} \sum_{i=1}^{N_{\text{in}}} t_i^\alpha = \frac{1}{N_{\text{in}}} \sum_{i=1}^{N_{\text{in}}} \exp \left( \tfrac{1}{2} \boldsymbol{\vartheta}^\top \boldsymbol{u}_i(\beta) \right) + O(r) =: M_{-1} + O(r),$$

with $M_{-1} > 0$ bounded away from 0 in a neighborhood. Then

$$(M_k)^r = \exp \left( r \log M_k \right) = \exp \left( r \left[ \log M_{-1} + O(r) \right] \right) = 1 + r \log M_{-1} + O(r^2),$$

i.e.

$$\left(\tfrac{1}{N_{\text{in}}} \sum t_i^\alpha\right)^r = 1 + r \log\left(\tfrac{1}{N_{\text{in}}} \sum \exp\left(\tfrac{1}{2}\boldsymbol{\vartheta}^\top \boldsymbol{u}_i(\beta)\right)\right) + O(r^2).$$

Substituting into $H_k$ gives

$$H_k(\boldsymbol{\lambda}, \beta) = \frac{N_{\text{in}}}{k(k+1)}\left[1 + r \log M_{-1} + O(r^2)\right] = \frac{N_{\text{in}}}{k(k+1)} + \frac{N_{\text{in}}}{k} \log M_{-1} + O(r),$$

where the last $O(r)$ term comes from $\frac{N_{\text{in}}}{k(k+1)}O(r^2) = \frac{N_{\text{in}}}{kr}O(r^2) = \frac{N_{\text{in}}}{k}O(r) = O(r)$ since $1/k = O(1)$ as $k \to -1$. $\qquad\square$

## D.2 Convexity of the inner optimization

**Lemma A2** (Inner dual equivalence and convexity). *For a fixed $\beta$, we define*

$$H_k(\boldsymbol{\lambda}, \boldsymbol{\beta}) = \frac{N_{\text{in}}}{k(k+1)}\left\{\frac{1}{N_{\text{in}}} \sum_{i=1}^{N_{\text{in}}}\left(1 + \boldsymbol{\lambda}^\top \boldsymbol{u}_i(\boldsymbol{\beta})\right)^{k/(k+1)}\right\}^{k+1},$$

*with the conventions that $k = 0$ (empirical likelihood, EL) and $k = -1$ (exponential tilting, ET) are taken as limits. Define the corresponding minimization form*

$$M_k(\boldsymbol{\lambda}, \boldsymbol{\beta}) = \begin{cases} -\sum_{i=1}^{N_{\text{in}}} \log\left(1 + \boldsymbol{\lambda}^\top \boldsymbol{u}_i(\boldsymbol{\beta})\right), & k = 0, \\[2mm] \sum_{i=1}^{N_{\text{in}}} \exp\left(\tfrac{1}{2}\boldsymbol{\lambda}^\top \boldsymbol{u}_i(\boldsymbol{\beta})\right), & k = -1, \\[2mm] \sum_{i=1}^{N_{\text{in}}}\left(1 + \boldsymbol{\lambda}^\top \boldsymbol{u}_i(\boldsymbol{\beta})\right)^{k/(k+1)}, & k < 0, k \neq -1, \\[2mm] -\sum_{i=1}^{N_{\text{in}}}\left(1 + \boldsymbol{\lambda}^\top \boldsymbol{u}_i(\boldsymbol{\beta})\right)^{k/(k+1)}, & k > 0. \end{cases}$$

*Then:*

(i) ***Monotone mapping and equivalence.*** *There exists a strictly decreasing scalar map $h_k(\cdot)$ such that*
$$H_k(\boldsymbol{\lambda}, \boldsymbol{\beta}) = h_k\left(M_k(\boldsymbol{\lambda}, \boldsymbol{\beta})\right),$$
*where explicitly*

$$h_k(m) = \begin{cases} -m, & k = 0, \\[2mm] -N_{\text{in}} \log\left(m/N_{\text{in}}\right) = -N_{\text{in}} \log m + \text{const}, & k = -1, \\[2mm] \dfrac{N_{\text{in}}^{-k}}{k(k+1)} m^{k+1}, & k < 0, k \neq -1, \\[3mm] \dfrac{N_{\text{in}}^{-k}}{k(k+1)} (-m)^{k+1}, & k > 0. \end{cases}$$

*Hence,*
$$\max_{\boldsymbol{\lambda}} H_k(\boldsymbol{\lambda}, \boldsymbol{\beta}) \iff \min_{\boldsymbol{\lambda}} M_k(\boldsymbol{\lambda}, \boldsymbol{\beta}).$$

(ii) ***Convexity of the inner problem.*** *For every $k \in \mathbb{R}$, the map $\boldsymbol{\lambda} \mapsto M_k(\boldsymbol{\lambda}, \boldsymbol{\beta})$ is convex on its domain, and thus the inner program*
$$\min_{\boldsymbol{\lambda}} M_k(\boldsymbol{\lambda}, \boldsymbol{\beta}) \quad (\text{equivalently, } \max_{\boldsymbol{\lambda}} H_k(\boldsymbol{\lambda}, \boldsymbol{\beta}))$$
*is a convex optimization problem in $\boldsymbol{\lambda}$.*

*Proof. (i) Substitute the definition of $M_k$ into $H_k$. For $k \neq -1, 0$,*

$$H_k(\boldsymbol{\lambda}, \boldsymbol{\beta}) = \frac{N_{\text{in}}}{k(k+1)}\left(\tfrac{1}{N_{\text{in}}}(\pm M_k)\right)^{k+1} = \frac{N_{\text{in}}^{-k}}{k(k+1)}(\pm M_k)^{k+1},$$

*where the $\pm$ sign matches the definition of $M_k$ above. For $k = 0$ and $k = -1$, the continuous limits give*

$$H_0(\boldsymbol{\lambda}, \boldsymbol{\beta}) = -M_0(\boldsymbol{\lambda}, \boldsymbol{\beta}), \qquad H_{-1}(\boldsymbol{\lambda}, \boldsymbol{\beta}) = -N_{\text{in}} \log\left(M_{-1}(\boldsymbol{\lambda}, \boldsymbol{\beta})/N_{\text{in}}\right).$$

*Thus in all cases $H_k = h_k(M_k)$, with derivatives*

$$h_k'(m) = \begin{cases} -1, & k = 0, \\[2mm] -\dfrac{N_{\text{in}}}{m}, & k = -1, \\[2mm] \dfrac{N_{\text{in}}^{-k}}{k} m^k, & k < 0, k \neq -1, \\[2mm] \dfrac{N_{\text{in}}^{-k}}{k} (-m)^k, & k > 0, \end{cases}$$

*which is strictly negative on the domain of $m$ in each case ($M_k > 0$ if $k < 0$, $M_k < 0$ if $k > 0$, $M_0 \in \mathbb{R}$, $M_{-1} > 0$). Thus, each $h_k$ is strictly decreasing, yielding the equivalence*

$$\max_{\boldsymbol{\lambda}} H_k(\boldsymbol{\lambda}, \boldsymbol{\beta}) \iff \min_{\boldsymbol{\lambda}} M_k(\boldsymbol{\lambda}, \boldsymbol{\beta}).$$

*(ii) For convexity, write $t_i(\boldsymbol{\lambda}) = 1 + \boldsymbol{\lambda}^\top \boldsymbol{u}_i(\boldsymbol{\beta})$, an affine function of $\boldsymbol{\lambda}$. If $k < 0$, then $\alpha = \frac{k}{k+1} > 1$ or $\alpha < 0$, so $t \mapsto t^\alpha$ is convex on $t > 0$ and $M_k$ is a sum of convex functions of affine maps. If $k > 0$, then $0 < \alpha < 1$, so $t^\alpha$ is concave, but the leading minus sign in $M_k$ flips concavity to convexity. For $k = 0$, $-\log t$ is convex on $t > 0$; for $k = -1$, $\exp(t/2)$ is convex on $\mathbb{R}$. Summation preserves convexity, so $M_k$ is convex in all cases.* $\qquad\square$

## APPENDIX E  PROOF OF THEOREM 2

**Assumption A1** (Regularity assumptions). *Let $\{(Y_i, \mathbf{X}_i, \mathbf{Z}_i)\}_{i=1}^{N_{\text{in}}}$ be i.i.d. observations from the true parameter $\boldsymbol{\beta}^\star \in \mathbb{R}^d$. Define the log-likelihood score $\boldsymbol{s}_i(\boldsymbol{\beta}) := \nabla_{\boldsymbol{\beta}} \log f_{\boldsymbol{\beta}}(Y_i \mid \mathbf{X}_i, \mathbf{Z}_i)$ and the auxiliary moment function $\boldsymbol{u}_i(\boldsymbol{\beta}) \in \mathbb{R}^q$. Assume:*

*(a) (Likelihood regularity) $\mathbb{E}\{s_i(\boldsymbol{\beta}^\star)\} = \mathbf{0}$, the Fisher information $\mathbf{J} := \mathbb{E}\{-\nabla_{\boldsymbol{\beta}\boldsymbol{\beta}}^2 \log f_{\boldsymbol{\beta}^\star}(Y_i \mid \mathbf{X}_i, \mathbf{Z}_i)\}$ is finite and positive definite, and standard MLE regularity conditions (dominated differentiability, interchange of differentiation and expectation, finite variance) hold.*

*(b) (Well-specified moments) $\mathbf{S} := \mathbb{E}\{\boldsymbol{u}_i(\boldsymbol{\beta}^\star)\boldsymbol{u}_i(\boldsymbol{\beta}^\star)^\top\}$ is finite and positive definite.*

*(c) (Jacobian and rank) $\boldsymbol{u}_i(\boldsymbol{\beta})$ is continuously differentiable near $\boldsymbol{\beta}^\star$ with $\mathbf{G} := \mathbb{E}\{\nabla_{\boldsymbol{\beta}} \boldsymbol{u}_i(\boldsymbol{\beta}^\star)\} \in \mathbb{R}^{q \times d}$ full column rank.*

*(d) (Positivity domain) There exists an open $\Lambda \subset \mathbb{R}^q$ containing $\mathbf{0}$ such that $t_i(\boldsymbol{\lambda}, \boldsymbol{\beta}) := 1 + \boldsymbol{\lambda}^\top \boldsymbol{u}_i(\boldsymbol{\beta}) > 0$ almost surely for all $(\boldsymbol{\lambda}, \boldsymbol{\beta}) \in \Lambda \times \mathcal{N}(\boldsymbol{\beta}^\star)$.*

*(e) (Moments) Finite $(2+\delta)$ moments exist for some $\delta > 0$, so uniform LLNs and CLTs apply to the empirical means $\bar{\boldsymbol{s}}_N(\boldsymbol{\beta}) := N_{\text{in}}^{-1} \sum_i \boldsymbol{s}_i(\boldsymbol{\beta})$, $\bar{\boldsymbol{u}}_N(\boldsymbol{\beta}) := N_{\text{in}}^{-1} \sum_i \boldsymbol{u}_i(\boldsymbol{\beta})$, and their Jacobians.*

Table A1 lists the notations for gradients, Hessians and cross Jacobian blocks that were used throughout the Appendix.

### E.1  INNER MAXIMIZER

Write the population inner objective as a composition

$$H_k(\boldsymbol{\lambda}, \boldsymbol{\beta}) = h_k\Big(M_k(\boldsymbol{\lambda}, \boldsymbol{\beta})\Big), \qquad M_k(\boldsymbol{\lambda}, \boldsymbol{\beta}) = \mathbb{E}\big[m_k\big(1 + \boldsymbol{\lambda}^\top \boldsymbol{u}(\boldsymbol{\beta})\big)\big], \quad t = \boldsymbol{\lambda}^\top \boldsymbol{u}(\boldsymbol{\beta})$$

where, for the Cressie-Read family (including the EL/ET limits),

$$m_k(t\boldsymbol{\lambda}) = \begin{cases} -\log[1 + t_i(\boldsymbol{\lambda})], & k = 0 \\ \exp[t_i(\boldsymbol{\lambda})/2], & k = -1, \\ [1 + t_i(\boldsymbol{\lambda})]^\alpha, & k < 0, k \neq -1, \\ -[1 + t_i(\boldsymbol{\lambda})]^\alpha, & k > 0. \end{cases} \qquad h_k(M) = \begin{cases} -M, & k = 0, \\ -\log M, & k = -1, \\ \dfrac{1}{k(k+1)} M^{k+1}, & k < 0, k \neq -1. \\ \dfrac{1}{k(k+1)} (-M)^{k+1}, & k > 0, \end{cases}$$

Table A1: Notations for gradients, Hessians, and cross blocks. Aggregate versions are defined as sums over $N_{\text{in}}$ observations. Per-observation versions are the per-sample contributions. Population limit denotes the expectation of the Per-observation.

| Block | Aggregate | Per-observation | Population limit |
|---|---|---|---|
| Log-likelihood score | $\mathcal{S}_n := \sum_{i=1}^{N_{\text{in}}} \boldsymbol{s}_i(\boldsymbol{\beta}^\star)$ | $\boldsymbol{s}_i(\boldsymbol{\beta}^\star)$ | $\mathbb{E}\{\boldsymbol{s}_i(\boldsymbol{\beta}^\star)\} = \boldsymbol{0}$ |
| Log-likelihood Hessian | $\mathcal{J}_n := -\sum_{i=1}^{N_{\text{in}}} \nabla^2_{\boldsymbol{\beta}\boldsymbol{\beta}} \ell_i(\boldsymbol{\beta}^\star)$ | $\mathbf{j}_i := -\nabla^2_{\boldsymbol{\beta}\boldsymbol{\beta}} \ell_i(\boldsymbol{\beta}^\star)$ | $\mathbf{J} := \mathbb{E}(\mathbf{j}_i)$ |
| Penalty gradient | $\mathcal{G}_n := \nabla_{\boldsymbol{\lambda}} H_k(\boldsymbol{0}, \boldsymbol{\beta}^\star)$ | $\mathbf{g}_i := \frac{1}{k+1} \boldsymbol{u}_i(\boldsymbol{\beta}^\star)$ | $\mathbb{E}[\mathbf{g}_i] = \boldsymbol{0}$ |
| Penalty Hessian | $\mathcal{H}_n := \nabla^2_{\boldsymbol{\lambda}\boldsymbol{\lambda}} H_k(\boldsymbol{0}, \boldsymbol{\beta}^\star)$ | $\mathbf{h}_i := -\frac{1}{(k+1)^2} \boldsymbol{u}_i(\boldsymbol{\beta}^\star) \boldsymbol{u}_i(\boldsymbol{\beta}^\star)^\top$ | $-\mathbb{E}[\mathbf{h}_i] = \frac{1}{(k+1)^2} \mathbf{S}$ |
| Cross Jacobian | $\mathcal{C}_n := \nabla^2_{\boldsymbol{\lambda}\boldsymbol{\beta}} H_k(\boldsymbol{0}, \boldsymbol{\beta}^\star)$ | $\mathbf{c}_i := \frac{1}{k+1} \nabla_{\boldsymbol{\beta}} \boldsymbol{u}_i(\boldsymbol{\beta}^\star)$ | $\mathbb{E}[\mathbf{c}_i] = \frac{1}{(k+1)} \mathbf{G}$ |
| Penalty covariance | $\boldsymbol{\Omega}_1^{\text{agg}} := \mathbb{E}[\mathcal{G}_n \mathcal{G}_n^\top]$ | $\boldsymbol{\Omega}_1 := \mathbb{E}[\mathbf{g}_i \mathbf{g}_i^\top]$ | $\boldsymbol{\Omega}_1 = \frac{1}{(k+1)^2} \mathbf{S}$ |
| Penalty curvature | $\boldsymbol{\Omega}_2^{\text{agg}} := -\mathbb{E}[\mathcal{H}_n]$ | $\boldsymbol{\Omega}_2 := -\mathbb{E}[\mathbf{h}_i]$ | $\boldsymbol{\Omega}_2 = \frac{1}{(k+1)^2} \mathbf{S}$ |

*Note.* Population limits are stated under correct model specification, i.e. $\mathbb{E}[\boldsymbol{u}_i(\boldsymbol{\beta}^\star)] = \boldsymbol{0}$. *Definitions.* $\ell_i(\boldsymbol{\beta}) := \log f_{\boldsymbol{\beta}}(Y_i \mid \mathbf{X}_i, \mathbf{Z}_i)$, $\mathbf{S} := \mathbb{E}[\boldsymbol{u}_i(\boldsymbol{\beta}^\star) \boldsymbol{u}_i(\boldsymbol{\beta}^\star)^\top]$, $\mathbf{G} := \mathbb{E}[\nabla_{\boldsymbol{\beta}} \boldsymbol{u}_i(\boldsymbol{\beta}^\star)]$. Here $H_k(\boldsymbol{\lambda}, \boldsymbol{\beta})$ denotes the inner objective for Cressie–Read index $k$, and $\nabla$. denotes gradients/Hessians evaluated at $(\boldsymbol{\lambda}, \boldsymbol{\beta}) = (\boldsymbol{0}, \boldsymbol{\beta}^\star)$.

Domain of the mapping $h_k$ is : $M > 0$ for $k < 0$ and $M < 0$ for $k > 0$. By the chain rule,

$$\nabla_{\boldsymbol{\lambda}} H_k(\boldsymbol{\lambda}, \boldsymbol{\beta}) = h_k'\big(M_k(\boldsymbol{\lambda}, \boldsymbol{\beta})\big) \cdot \mathbb{E}\Big[m_k'\big(1 + \boldsymbol{\lambda}^\top \boldsymbol{u}(\boldsymbol{\beta})\big)\,\boldsymbol{u}(\boldsymbol{\beta})\Big].$$

For every CR member (including EL/ET), $m_k'(t)$ and $h_k'(M)$ are finite. Hence, at $(\boldsymbol{\lambda}, \boldsymbol{\beta}) = (\mathbf{0}, \boldsymbol{\beta}^\star)$,

$$\nabla_{\boldsymbol{\lambda}} H_k(\mathbf{0}, \boldsymbol{\beta}^\star) = h_k'\big(M_k(\mathbf{0}, \boldsymbol{\beta}^\star)\big) \mathbb{E}\big[m_k'(1)\,\boldsymbol{u}(\boldsymbol{\beta}^\star)\big] = h_k'(\cdot)\,m_k'(0)\,\mathbb{E}[\boldsymbol{u}(\boldsymbol{\beta}^\star)] = \mathbf{0},$$

by the well-specified moment assumption $\mathbb{E}[\boldsymbol{u}(\boldsymbol{\beta}^\star)] = \mathbf{0}$. Equivalently, the population $M$-form first-order condition is

$$\nabla_{\boldsymbol{\lambda}} M_k(\boldsymbol{\lambda}, \boldsymbol{\beta}) = \mathbb{E}\Big[m_k'\big(1 + \boldsymbol{\lambda}^\top \boldsymbol{u}(\boldsymbol{\beta})\big)\,\boldsymbol{u}(\boldsymbol{\beta})\Big] = \mathbf{0},$$

which is satisfied at $(\boldsymbol{\lambda}, \boldsymbol{\beta}) = (\mathbf{0}, \boldsymbol{\beta}^\star)$. Since $M_k(\cdot, \boldsymbol{\beta})$ is convex in $\boldsymbol{\lambda}$ (and strictly convex under standard conditions), $\boldsymbol{\lambda}^\star(\boldsymbol{\beta}^\star) = \mathbf{0}$ is the unique population minimizer of $M_k$, and thus the unique maximizer of $H_k$.

### E.2 Lemmas for the Gradient, Hessian, Jacobian of $H$ under the population optimum

**Lemma A3** (Gradient and Hessian of $H_k(\boldsymbol{\lambda}, \boldsymbol{\beta})$ w.r.t. $\boldsymbol{\lambda}$ for $k \neq -1$ ). *Let $k \in \mathbb{R} \setminus \{-1\}$, and $\alpha = \frac{k}{k+1}$ and $t_i = 1 + \boldsymbol{\lambda}^\top \boldsymbol{u}_i(\boldsymbol{\beta}^\star)$. At the population optimum $(\boldsymbol{\beta}^\star, \boldsymbol{\lambda}^\star = \mathbf{0})$, define the aggregate gradient and Hessian*

$$\mathcal{G}_n := \nabla_{\boldsymbol{\lambda}} H_k(\mathbf{0}, \boldsymbol{\beta}^\star), \qquad \mathcal{H}_n := \nabla_{\boldsymbol{\lambda}\boldsymbol{\lambda}}^2 H_k(\mathbf{0}, \boldsymbol{\beta}^\star).$$

*Then*

$$\mathcal{G}_n = \frac{N_{\text{in}}}{k+1}\,\bar{\boldsymbol{u}}, \qquad \mathcal{H}_n = N_{\text{in}}\Big[\alpha^2 \bar{\boldsymbol{u}}\bar{\boldsymbol{u}}^\top - \frac{1}{(k+1)^2}\bar{\mathbf{S}}\Big],$$

*where $\bar{\boldsymbol{u}} := \frac{1}{N_{\text{in}}} \sum_{i=1}^{N_{\text{in}}} \boldsymbol{u}_i(\boldsymbol{\beta}^\star)$ and $\bar{\mathbf{S}} := \frac{1}{N_{\text{in}}} \sum_{i=1}^{N_{\text{in}}} \boldsymbol{u}_i(\boldsymbol{\beta}^\star)\boldsymbol{u}_i(\boldsymbol{\beta}^\star)^\top$.*

***Per-sample scaling.** Define the per-sample gradient and Hessian contributions*

$$\boldsymbol{g}_i := \frac{1}{k+1}\boldsymbol{u}_i(\boldsymbol{\beta}^\star), \qquad \mathbf{h}_i := -\frac{1}{(k+1)^2}\boldsymbol{u}_i(\boldsymbol{\beta}^\star)\boldsymbol{u}_i(\boldsymbol{\beta}^\star)^\top,$$

*so that $\mathcal{G}_n = \sum_{i=1}^{N_{\text{in}}} \boldsymbol{g}_i$, $\mathcal{H}_n = \sum_{i=1}^{N_{\text{in}}} \mathbf{h}_i$.*

*Then under correct specification, $\mathbb{E}[\boldsymbol{u}_i(\boldsymbol{\beta}^\star)] = \mathbf{0}$,*

$$\frac{1}{N_{\text{in}}}\mathcal{G}_n \xrightarrow{p} 0, \qquad -\frac{1}{N_{\text{in}}}\mathcal{H}_n \xrightarrow{p} \frac{1}{(k+1)^2}\mathbf{S}, \quad \mathbf{S} := \mathbb{E}[\boldsymbol{u}_i(\boldsymbol{\beta}^\star)\boldsymbol{u}_i(\boldsymbol{\beta}^\star)^\top].$$

***Penalty covariance and curvature.** At the per-sample level define*

$$\boldsymbol{\Omega}_1 := \mathbb{E}[\boldsymbol{g}_i \boldsymbol{g}_i^\top], \qquad \boldsymbol{\Omega}_2 := -\mathbb{E}[\mathbf{h}_i].$$

*Then*

$$\boldsymbol{\Omega}_1 = \frac{1}{(k+1)^2}\mathbf{S}, \qquad \boldsymbol{\Omega}_2 = \frac{1}{(k+1)^2}\mathbf{S} + O\left(\frac{1}{N_{\text{in}}}\right).$$

***Special case.** In the limit $k \to 0$ (empirical likelihood), these reduce to*

$$\text{EL } (k \to 0): \quad \boldsymbol{g}_i = \boldsymbol{u}_i(\boldsymbol{\beta}^\star), \qquad\qquad -\mathbf{h}_i = \boldsymbol{u}_i(\boldsymbol{\beta}^\star)\boldsymbol{u}_i(\boldsymbol{\beta}^\star)^\top.$$

*Proof of Lemma A3.* Let $\boldsymbol{\beta}^\star$ be the true parameter and define, for $\alpha = \frac{k}{k+1}$,

$$t_i(\boldsymbol{\lambda}) = 1 + \boldsymbol{\lambda}^\top \boldsymbol{u}_i(\boldsymbol{\beta}^\star), \qquad m_i(\boldsymbol{\lambda}) = t_i(\boldsymbol{\lambda})^\alpha, \qquad A(\boldsymbol{\lambda}) = \frac{1}{N_{\text{in}}} \sum_{i=1}^{N_{\text{in}}} m_i(\boldsymbol{\lambda}),$$

$$H_k(\boldsymbol{\lambda}) = \frac{N_{\text{in}}}{k(k+1)}\big(A(\boldsymbol{\lambda})\big)^{k+1}.$$

All derivatives are w.r.t. $\boldsymbol{\lambda}$. Since $t_i$ is affine,

$$\nabla_{\boldsymbol{\lambda}} m_i(\boldsymbol{\lambda}) = \alpha t_i(\boldsymbol{\lambda})^{\alpha-1}\boldsymbol{u}_i, \qquad \nabla^2_{\boldsymbol{\lambda}\boldsymbol{\lambda}} m_i(\boldsymbol{\lambda}) = \alpha(\alpha-1)t_i(\boldsymbol{\lambda})^{\alpha-2}\boldsymbol{u}_i\boldsymbol{u}_i^{\top},$$

hence

$$\nabla_{\boldsymbol{\lambda}} A(\boldsymbol{\lambda}) = \frac{1}{N_{\text{in}}}\sum_i \nabla_{\boldsymbol{\lambda}} m_i(\boldsymbol{\lambda}), \qquad \nabla^2_{\boldsymbol{\lambda}\boldsymbol{\lambda}} A(\boldsymbol{\lambda}) = \frac{1}{N_{\text{in}}}\sum_i \nabla^2_{\boldsymbol{\lambda}\boldsymbol{\lambda}} m_i(\boldsymbol{\lambda}).$$

By the chain rule,

$$\nabla_{\boldsymbol{\lambda}} H_k(\boldsymbol{\lambda}) = \frac{N_{\text{in}}}{k} A(\boldsymbol{\lambda})^k \nabla_{\boldsymbol{\lambda}} A(\boldsymbol{\lambda}),$$

$$\nabla^2_{\boldsymbol{\lambda}\boldsymbol{\lambda}} H_k(\boldsymbol{\lambda}) = \frac{N_{\text{in}}}{k}\left[kA(\boldsymbol{\lambda})^{k-1}\nabla_{\boldsymbol{\lambda}} A(\boldsymbol{\lambda})\nabla_{\boldsymbol{\lambda}} A(\boldsymbol{\lambda})^{\top} + A(\boldsymbol{\lambda})^k \nabla^2_{\boldsymbol{\lambda}\boldsymbol{\lambda}} A(\boldsymbol{\lambda})\right].$$

Evaluate at $\boldsymbol{\lambda} = \boldsymbol{0}$: since $t_i(\boldsymbol{0}) = 1$, we have $A(\boldsymbol{0}) = 1$ and

$$\nabla_{\boldsymbol{\lambda}} A(\boldsymbol{0}) = \alpha\bar{\boldsymbol{u}}, \qquad \nabla^2_{\boldsymbol{\lambda}\boldsymbol{\lambda}} A(\boldsymbol{0}) = \alpha(\alpha-1)\bar{\mathbf{S}},$$

with $\bar{\boldsymbol{u}} := N_{\text{in}}^{-1}\sum_i \boldsymbol{u}_i(\boldsymbol{\beta}^{\star})$ and $\bar{\mathbf{S}} := N_{\text{in}}^{-1}\sum_i \boldsymbol{u}_i(\boldsymbol{\beta}^{\star})\boldsymbol{u}_i(\boldsymbol{\beta}^{\star})^{\top}$. Define the *aggregate* penalty gradient and Hessian

$$\mathcal{G}_n := \nabla_{\boldsymbol{\lambda}} H_k(\boldsymbol{0}, \boldsymbol{\beta}^{\star}), \qquad \mathcal{H}_n := \nabla^2_{\boldsymbol{\lambda}\boldsymbol{\lambda}} H_k(\boldsymbol{0}, \boldsymbol{\beta}^{\star}).$$

Then

$$\mathcal{G}_n = \frac{N_{\text{in}}}{k}\alpha\bar{\boldsymbol{u}} = \frac{N_{\text{in}}}{k+1}\bar{\boldsymbol{u}},$$

$$\mathcal{H}_n = \frac{N_{\text{in}}}{k}\left[k\alpha^2\bar{\boldsymbol{u}}\bar{\boldsymbol{u}}^{\top} + \alpha(\alpha-1)\bar{\mathbf{S}}\right] = N_{\text{in}}\left[\alpha^2\bar{\boldsymbol{u}}\bar{\boldsymbol{u}}^{\top} - \frac{1}{(k+1)^2}\bar{\mathbf{S}}\right].$$

Under $\mathbb{E}[\boldsymbol{u}_i(\boldsymbol{\beta}^{\star})] = \boldsymbol{0}$,

$$\mathbb{E}[\bar{\boldsymbol{u}}\bar{\boldsymbol{u}}^{\top}] = \frac{1}{N_{\text{in}}}\mathbf{S}, \qquad \mathbb{E}[\bar{\mathbf{S}}] = \mathbf{S}, \qquad \mathbf{S} := \mathbb{E}[\boldsymbol{u}_i(\boldsymbol{\beta}^{\star})\boldsymbol{u}_i(\boldsymbol{\beta}^{\star})^{\top}],$$

so

$$\mathbb{E}[\mathcal{H}_n] = \frac{N_{\text{in}}}{k}\left[k\alpha^2 \cdot \frac{1}{N_{\text{in}}}\mathbf{S} + \alpha(\alpha-1)\mathbf{S}\right] = \alpha^2\mathbf{S} - \frac{N_{\text{in}}}{(k+1)^2}\mathbf{S} = -\frac{N_{\text{in}}}{(k+1)^2}\mathbf{S} + O(1).$$

Now introduce the average per-sample versions

$$\bar{\boldsymbol{g}}_n := \frac{1}{N_{\text{in}}}\mathcal{G}_n = \frac{1}{k+1}\bar{\boldsymbol{u}}, \qquad \bar{\mathbf{h}}_n := \frac{1}{N_{\text{in}}}\mathcal{H}_n = \alpha^2\bar{\boldsymbol{u}}\bar{\boldsymbol{u}}^{\top} - \frac{1}{(k+1)^2}\bar{\mathbf{S}}.$$

Then $\bar{\boldsymbol{g}}_n \xrightarrow{p} 0$ and $-\bar{\mathbf{h}}_n \xrightarrow{p} \frac{1}{(k+1)^2}\mathbf{S}$.

Finally, the per-observation penalty gradient and Hessian are

$$\boldsymbol{g}_i := \frac{1}{k+1}\boldsymbol{u}_i(\boldsymbol{\beta}^{\star}), \qquad \mathbf{h}_i := -\frac{1}{(k+1)^2}\boldsymbol{u}_i(\boldsymbol{\beta}^{\star})\boldsymbol{u}_i(\boldsymbol{\beta}^{\star})^{\top}.$$

Under correct specification, their population limits define the penalty covariance and curvature:

$$\boldsymbol{\Omega}_1 := \mathbb{E}[\boldsymbol{g}_i\boldsymbol{g}_i^{\top}] = \frac{1}{(k+1)^2}\mathbf{S}, \qquad \boldsymbol{\Omega}_2 := -\mathbb{E}[\mathbf{h}_i] = \frac{1}{(k+1)^2}\mathbf{S}.$$

$\square$

**Lemma A4** (Gradient and Hessian of $H_k(\boldsymbol{\lambda}, \boldsymbol{\beta})$, w.r.t. $\boldsymbol{\beta}$). *Let $k \in \mathbb{R} \setminus \{-1\}$, set $\alpha = \frac{k}{k+1}$, and define*

$$t_i(\boldsymbol{\lambda}, \boldsymbol{\beta}) = 1 + \boldsymbol{\lambda}^{\top}\boldsymbol{u}_i(\boldsymbol{\beta}), \qquad A(\boldsymbol{\lambda}, \boldsymbol{\beta}) = \frac{1}{N_{\text{in}}}\sum_{i=1}^{N_{\text{in}}} t_i(\boldsymbol{\lambda}, \boldsymbol{\beta})^{\alpha},$$

$$H_k(\boldsymbol{\lambda}, \boldsymbol{\beta}) = \frac{N_{\text{in}}}{k(k+1)}\left(A(\boldsymbol{\lambda}, \boldsymbol{\beta})\right)^{k+1}.$$

*Assume $\boldsymbol{u}_i(\boldsymbol{\beta}) \in \mathbb{R}^q$ is differentiable in $\boldsymbol{\beta} \in \mathbb{R}^d$. At the population optimum $(\boldsymbol{\beta}^\star, \boldsymbol{\lambda}^\star = \mathbf{0})$:*

**$\boldsymbol{\beta}$-block vanishes at $\boldsymbol{\lambda} = \mathbf{0}$.**

$$\nabla_{\boldsymbol{\beta}} H_k(\mathbf{0}, \boldsymbol{\beta}^\star) = \mathbf{0}, \qquad \nabla^2_{\boldsymbol{\beta}\boldsymbol{\beta}} H_k(\mathbf{0}, \boldsymbol{\beta}^\star) = \mathbf{0}.$$

**Cross blocks.** *Let $\overline{\mathbf{J}}_u := \frac{1}{N_{\mathrm{in}}} \sum_{i=1}^{N_{\mathrm{in}}} \nabla_{\boldsymbol{\beta}} \boldsymbol{u}_i(\boldsymbol{\beta}^\star) \in \mathbb{R}^{q \times d}$. Then*

$$\nabla^2_{\boldsymbol{\lambda}\boldsymbol{\beta}} H_k(\mathbf{0}, \boldsymbol{\beta}^\star) = \frac{N_{\mathrm{in}}}{k+1} \overline{\mathbf{J}}_u, \qquad \nabla^2_{\boldsymbol{\beta}\boldsymbol{\lambda}} H_k(\mathbf{0}, \boldsymbol{\beta}^\star) = \left( \nabla^2_{\boldsymbol{\lambda}\boldsymbol{\beta}} H_k(\mathbf{0}, \boldsymbol{\beta}^\star) \right)^\top.$$

**Limits of the per-observation cross jacobian** *Define the aggregate cross block $\mathcal{C}_n := \nabla^2_{\boldsymbol{\lambda}\boldsymbol{\beta}} H_k(\mathbf{0}, \boldsymbol{\beta}^\star)$ and its per-sample version $\mathbf{c}_n := \frac{1}{N_{\mathrm{in}}} \mathcal{C}_n$. Then*

$$\mathbf{c}_n = \frac{1}{k+1} \overline{\mathbf{J}}_u \xrightarrow{p} \frac{1}{k+1} \mathbf{G}, \qquad \mathbf{G} := \mathbb{E}\left[ \nabla_{\boldsymbol{\beta}} \boldsymbol{u}_i(\boldsymbol{\beta}^\star) \right] \in \mathbb{R}^{q \times d}.$$

*Equivalently, at the per-observation level,*

$$\mathbf{c}_i = \frac{1}{k+1} \nabla_{\boldsymbol{\beta}} \boldsymbol{u}_i(\boldsymbol{\beta}^\star), \qquad \mathbb{E}[\mathbf{c}_i] = \frac{1}{k+1} \mathbf{G}.$$

*All block dimensions are consistent: $\nabla^2_{\boldsymbol{\lambda}\boldsymbol{\beta}} H_k \in \mathbb{R}^{q \times d}$ and $\nabla^2_{\boldsymbol{\beta}\boldsymbol{\lambda}} H_k \in \mathbb{R}^{d \times q}$.*

*Proof of Lemma A4.* Write $A(\boldsymbol{\lambda}, \boldsymbol{\beta}) = N_{\mathrm{in}}^{-1} \sum_i t_i(\boldsymbol{\lambda}, \boldsymbol{\beta})^\alpha$ with $t_i = 1 + \boldsymbol{\lambda}^\top \boldsymbol{u}_i(\boldsymbol{\beta})$ and $\alpha = \frac{k}{k+1}$, where $\boldsymbol{\lambda} \in \mathbb{R}^q$, $\boldsymbol{\beta} \in \mathbb{R}^d$, and $\boldsymbol{u}_i(\boldsymbol{\beta}) \in \mathbb{R}^q$. Then

$$H_k(\boldsymbol{\lambda}, \boldsymbol{\beta}) = \frac{N_{\mathrm{in}}}{k(k+1)} A(\boldsymbol{\lambda}, \boldsymbol{\beta})^{k+1}.$$

Differentiate w.r.t. $\boldsymbol{\beta}$:

$$\nabla_{\boldsymbol{\beta}} H_k(\boldsymbol{\lambda}, \boldsymbol{\beta}) = \frac{N_{\mathrm{in}}}{k} A(\boldsymbol{\lambda}, \boldsymbol{\beta})^k \nabla_{\boldsymbol{\beta}} A(\boldsymbol{\lambda}, \boldsymbol{\beta}).$$

For each $i$, define the Jacobian and Hessian of $\boldsymbol{u}_i$ with respect to $\boldsymbol{\beta}$:

$$\mathbf{J}_i(\boldsymbol{\beta}) := \nabla_{\boldsymbol{\beta}} \boldsymbol{u}_i(\boldsymbol{\beta}) \in \mathbb{R}^{q \times d}, \qquad \mathbf{H}_{ij}(\boldsymbol{\beta}) := \nabla^2_{\boldsymbol{\beta}\boldsymbol{\beta}} u_{ij}(\boldsymbol{\beta}) \in \mathbb{R}^{d \times d},$$

for $j = 1, \ldots, q$.

Then

$$\nabla_{\boldsymbol{\beta}} t_i(\boldsymbol{\lambda}, \boldsymbol{\beta}) = \mathbf{J}_i(\boldsymbol{\beta})^\top \boldsymbol{\lambda} \in \mathbb{R}^d, \qquad \nabla^2_{\boldsymbol{\beta}\boldsymbol{\beta}} t_i(\boldsymbol{\lambda}, \boldsymbol{\beta}) = \sum_{j=1}^q \lambda_j \mathbf{H}_{ij}(\boldsymbol{\beta}) \in \mathbb{R}^{d \times d}.$$

By the chain rule with $\phi(x) = x^\alpha$, $\phi'(x) = \alpha x^{\alpha-1}$, $\phi''(x) = \alpha(\alpha-1)x^{\alpha-2}$,

$$\nabla_{\boldsymbol{\beta}} A(\boldsymbol{\lambda}, \boldsymbol{\beta}) = \frac{\alpha}{N_{\mathrm{in}}} \sum_{i=1}^{N_{\mathrm{in}}} t_i(\boldsymbol{\lambda}, \boldsymbol{\beta})^{\alpha-1} \mathbf{J}_i(\boldsymbol{\beta})^\top \boldsymbol{\lambda} \in \mathbb{R}^d, \tag{5}$$

$$\nabla^2_{\boldsymbol{\beta}\boldsymbol{\beta}} A(\boldsymbol{\lambda}, \boldsymbol{\beta}) = \frac{1}{N_{\mathrm{in}}} \sum_{i=1}^{N_{\mathrm{in}}} \left\{ \phi''(t_i) \nabla_{\boldsymbol{\beta}} t_i \nabla_{\boldsymbol{\beta}} t_i^\top + \phi'(t_i) \nabla^2_{\boldsymbol{\beta}\boldsymbol{\beta}} t_i \right\}$$

$$= \frac{\alpha}{N_{\mathrm{in}}} \sum_{i=1}^{N_{\mathrm{in}}} \left\{ (\alpha-1) t_i(\boldsymbol{\lambda}, \boldsymbol{\beta})^{\alpha-2} (\mathbf{J}_i(\boldsymbol{\beta})^\top \boldsymbol{\lambda})(\mathbf{J}_i(\boldsymbol{\beta})^\top \boldsymbol{\lambda})^\top + t_i(\boldsymbol{\lambda}, \boldsymbol{\beta})^{\alpha-1} \sum_{j=1}^q \lambda_j \mathbf{H}_{ij}(\boldsymbol{\beta}) \right\}.$$

At $\boldsymbol{\lambda} = \mathbf{0}$, we have $t_i(\mathbf{0}, \boldsymbol{\beta}) \equiv 1$, so

$$\nabla_{\boldsymbol{\beta}} A(\mathbf{0}, \boldsymbol{\beta}) = \mathbf{0}, \qquad \nabla^2_{\boldsymbol{\beta}\boldsymbol{\beta}} A(\mathbf{0}, \boldsymbol{\beta}) = \mathbf{0}.$$

Therefore,

$$\nabla_{\boldsymbol{\beta}} H_k(\mathbf{0}, \boldsymbol{\beta}) \equiv \mathbf{0}.$$

A second derivative gives

$$\nabla^2_{\boldsymbol{\beta\beta}} H_k(\boldsymbol{\lambda}, \boldsymbol{\beta}) = \frac{N_{\text{in}}}{k}\Big[ kA^{k-1}(\nabla_{\boldsymbol{\beta}} A)(\nabla_{\boldsymbol{\beta}} A)^\top + A^k \nabla^2_{\boldsymbol{\beta\beta}} A \Big],$$

so at $\boldsymbol{\lambda} = \mathbf{0}$ we also have $\nabla^2_{\boldsymbol{\beta\beta}} H_k(\mathbf{0}, \boldsymbol{\beta}) \equiv \mathbf{0}$.

**Remark:** The DRO objective $H_k(\boldsymbol{\lambda}, \boldsymbol{\beta})$ depends on $\boldsymbol{\beta}$ only through the moments $\boldsymbol{u}_i(\boldsymbol{\beta})$, which are multiplied by $\boldsymbol{\lambda}$. When $\boldsymbol{\lambda} = \mathbf{0}$, the moment conditions are effectively "switched off," so that $A(\boldsymbol{\lambda}, \boldsymbol{\beta}) \equiv 1$ regardless of $\boldsymbol{\beta}$. This implies that $H_k(\mathbf{0}, \boldsymbol{\beta})$ is constant in $\boldsymbol{\beta}$, with no gradient or curvature in the $\boldsymbol{\beta}$-block:

$$\nabla_{\boldsymbol{\beta}} H_k(\mathbf{0}, \boldsymbol{\beta}) = \mathbf{0}, \qquad \nabla^2_{\boldsymbol{\beta\beta}} H_k(\mathbf{0}, \boldsymbol{\beta}) = \mathbf{0}.$$

In other words, at the population optimum the $\boldsymbol{\beta}$-block carries no direct information; all information about $\boldsymbol{\beta}$ is mediated through the cross block $\nabla^2_{\boldsymbol{\lambda\beta}} H_k$, which involves the score functions $\boldsymbol{u}_i(\boldsymbol{\beta})$.

**Cross derivative $\nabla^2_{\boldsymbol{\lambda\beta}} A(\boldsymbol{\lambda}, \boldsymbol{\beta})$.** Recall the per-observation notations used previously

$$\mathbf{J}_i(\boldsymbol{\beta}) := \nabla_{\boldsymbol{\beta}} \boldsymbol{u}_i(\boldsymbol{\beta}) \in \mathbb{R}^{q \times d}, \qquad \mathbf{H}_{ij}(\boldsymbol{\beta}) := \nabla^2_{\boldsymbol{\beta\beta}} u_{ij}(\boldsymbol{\beta}) \in \mathbb{R}^{d \times d}.$$

Then

$$\nabla_{\boldsymbol{\lambda}} t_i(\boldsymbol{\lambda}, \boldsymbol{\beta}) = \boldsymbol{u}_i(\boldsymbol{\beta}), \quad \nabla_{\boldsymbol{\beta}} t_i(\boldsymbol{\lambda}, \boldsymbol{\beta}) = \mathbf{J}_i(\boldsymbol{\beta})^\top \boldsymbol{\lambda}, \quad \nabla^2_{\boldsymbol{\lambda\beta}} t_i(\boldsymbol{\lambda}, \boldsymbol{\beta}) = \mathbf{J}_i(\boldsymbol{\beta}).$$

With $\phi(x) = x^\alpha$ ($\phi'(x) = \alpha x^{\alpha-1}$, $\phi''(x) = \alpha(\alpha-1)x^{\alpha-2}$), and Eq. 5 the cross block of $A$ is

$$\nabla^2_{\boldsymbol{\lambda\beta}} A(\boldsymbol{\lambda}, \boldsymbol{\beta}) = \frac{1}{N_{\text{in}}} \sum_{i=1}^{N_{\text{in}}} \Big\{ \phi''(t_i) \boldsymbol{u}_i(\boldsymbol{\beta})\big(\mathbf{J}_i(\boldsymbol{\beta})^\top \boldsymbol{\lambda}\big)^\top + \phi'(t_i) \mathbf{J}_i(\boldsymbol{\beta}) \Big\}.$$

At $\boldsymbol{\lambda} = \mathbf{0}$, we have $t_i \equiv 1$ and hence

$$\nabla^2_{\boldsymbol{\lambda\beta}} A(\mathbf{0}, \boldsymbol{\beta}) = \frac{\alpha}{N_{\text{in}}} \sum_{i=1}^{N_{\text{in}}} \mathbf{J}_i(\boldsymbol{\beta}).$$

**Cross derivative $\nabla^2_{\boldsymbol{\lambda\beta}} H_k(\boldsymbol{\lambda}, \boldsymbol{\beta})$.** Since

$$H_k(\boldsymbol{\lambda}, \boldsymbol{\beta}) = \frac{N_{\text{in}}}{k(k+1)} \big(A(\boldsymbol{\lambda}, \boldsymbol{\beta})\big)^{k+1},$$

its gradient is

$$\nabla_{\boldsymbol{\lambda}} H_k(\boldsymbol{\lambda}, \boldsymbol{\beta}) = \frac{N_{\text{in}}}{k} A(\boldsymbol{\lambda}, \boldsymbol{\beta})^k \nabla_{\boldsymbol{\lambda}} A(\boldsymbol{\lambda}, \boldsymbol{\beta}).$$

Its cross derivative (a $q \times d$ matrix) is

$$\nabla^2_{\boldsymbol{\lambda\beta}} H_k(\boldsymbol{\lambda}, \boldsymbol{\beta}) = \frac{N_{\text{in}}}{k}\Big[ kA(\boldsymbol{\lambda}, \boldsymbol{\beta})^{k-1}\big(\nabla_{\boldsymbol{\lambda}} A(\boldsymbol{\lambda}, \boldsymbol{\beta})\big)\big(\nabla_{\boldsymbol{\beta}} A(\boldsymbol{\lambda}, \boldsymbol{\beta})\big)^\top + A(\boldsymbol{\lambda}, \boldsymbol{\beta})^k \nabla^2_{\boldsymbol{\lambda\beta}} A(\boldsymbol{\lambda}, \boldsymbol{\beta}) \Big]. \quad (6)$$

At $\boldsymbol{\lambda} = \mathbf{0}$, for the Cressie-Read inner average

$$A(\boldsymbol{\lambda}, \boldsymbol{\beta}) = \frac{1}{N_{\text{in}}} \sum_{i=1}^{N_{\text{in}}} \big(1 + \boldsymbol{\lambda}^\top \boldsymbol{u}_i(\boldsymbol{\beta})\big)^\alpha, \qquad \alpha = \tfrac{k}{k+1},$$

we have

$$A(\mathbf{0}, \boldsymbol{\beta}) = 1, \qquad \nabla_{\boldsymbol{\beta}} A(\mathbf{0}, \boldsymbol{\beta}) = \mathbf{0}, \qquad \nabla^2_{\boldsymbol{\lambda\beta}} A(\mathbf{0}, \boldsymbol{\beta}) = \frac{\alpha}{N_{\text{in}}} \sum_{i=1}^{N_{\text{in}}} \mathbf{J}_i(\boldsymbol{\beta}),$$

where $\mathbf{J}_i(\boldsymbol{\beta}) = \nabla_{\boldsymbol{\beta}} \boldsymbol{u}_i(\boldsymbol{\beta}) \in \mathbb{R}^{q \times d}$.

Thus, the first term in Eq. 6 vanishes, and

$$\nabla^2_{\boldsymbol{\lambda\beta}} H_k(\mathbf{0}, \boldsymbol{\beta}) = \frac{N_{\text{in}}}{k} \nabla^2_{\boldsymbol{\lambda\beta}} A(\mathbf{0}, \boldsymbol{\beta}) = \frac{N_{\text{in}}}{k} \cdot \frac{\alpha}{N_{\text{in}}} \sum_{i=1}^{N_{\text{in}}} \mathbf{J}_i(\boldsymbol{\beta}) = \frac{1}{k+1} \sum_{i=1}^{N_{\text{in}}} \mathbf{J}_i(\boldsymbol{\beta}) =: \mathcal{C}_n.$$

Per-sample scaling gives

$$\mathbf{c}_n := \frac{1}{N_{\text{in}}} \mathcal{C}_n = \frac{1}{k+1} \overline{\mathbf{J}}_u(\boldsymbol{\beta}), \qquad \overline{\mathbf{J}}_u(\boldsymbol{\beta}) := \frac{1}{N_{\text{in}}} \sum_{i=1}^{N_{\text{in}}} \mathbf{J}_i(\boldsymbol{\beta}).$$

At $\boldsymbol{\beta} = \boldsymbol{\beta}^\star$, a LLN gives

$$\mathbf{c}_n \xrightarrow{p} \frac{1}{k+1} \mathbf{G}, \qquad \mathbf{G} := \mathbb{E}\big[\nabla_{\boldsymbol{\beta}} \boldsymbol{u}_i(\boldsymbol{\beta}^\star)\big] \in \mathbb{R}^{q \times d}.$$

$\square$

**Lemma A5** (Exponential tilting (ET) blocks). *For the ET dual objective*

$$H_{-1}(\boldsymbol{\lambda}, \boldsymbol{\beta}) = -N_{\text{in}} \log \left\{ \frac{1}{N_{\text{in}}} \sum_{i=1}^{N_{\text{in}}} \exp\left(\tfrac{1}{2}\boldsymbol{\lambda}^\top \boldsymbol{u}_i(\boldsymbol{\beta})\right) \right\},$$

*let $\boldsymbol{u}_i(\boldsymbol{\beta}) \in \mathbb{R}^q$ and $\mathbf{J}_i(\boldsymbol{\beta}) := \nabla_{\boldsymbol{\beta}} \boldsymbol{u}_i(\boldsymbol{\beta}) \in \mathbb{R}^{q \times d}$. Define*

$$\bar{\boldsymbol{u}} := \tfrac{1}{N_{\text{in}}} \sum_i \boldsymbol{u}_i(\boldsymbol{\beta}), \quad \bar{\mathbf{S}} := \tfrac{1}{N_{\text{in}}} \sum_i \boldsymbol{u}_i(\boldsymbol{\beta})\boldsymbol{u}_i(\boldsymbol{\beta})^\top, \quad \overline{\mathbf{J}}_u := \tfrac{1}{N_{\text{in}}} \sum_i \mathbf{J}_i(\boldsymbol{\beta}).$$

*Then at $\boldsymbol{\lambda} = \mathbf{0}$,*

$$\nabla_{\boldsymbol{\lambda}} H_{-1}(\mathbf{0}, \boldsymbol{\beta}) = -\tfrac{N_{\text{in}}}{2} \bar{\boldsymbol{u}}, \qquad \nabla^2_{\boldsymbol{\lambda}\boldsymbol{\lambda}} H_{-1}(\mathbf{0}, \boldsymbol{\beta}) = -\tfrac{N_{\text{in}}}{4}\big(\bar{\mathbf{S}} - \bar{\boldsymbol{u}}\bar{\boldsymbol{u}}^\top\big),$$

$$\nabla^2_{\boldsymbol{\lambda}\boldsymbol{\beta}} H_{-1}(\mathbf{0}, \boldsymbol{\beta}) = -\tfrac{N_{\text{in}}}{2} \overline{\mathbf{J}}_u.$$

*Let*

$$\boldsymbol{g}_n := \tfrac{1}{N_{\text{in}}} \nabla_{\boldsymbol{\lambda}} H_{-1}, \quad \mathbf{h}_n := \tfrac{1}{N_{\text{in}}} \nabla^2_{\boldsymbol{\lambda}\boldsymbol{\lambda}} H_{-1}, \quad \mathbf{c}_n := \tfrac{1}{N_{\text{in}}} \nabla^2_{\boldsymbol{\lambda}\boldsymbol{\beta}} H_{-1}.$$

*Then at $\boldsymbol{\lambda} = \mathbf{0}$,*

$$\boldsymbol{g}_n = -\tfrac{1}{2}\bar{\boldsymbol{u}}, \qquad \mathbf{h}_n = -\tfrac{1}{4}(\bar{\mathbf{S}} - \bar{\boldsymbol{u}}\bar{\boldsymbol{u}}^\top), \qquad \mathbf{c}_n = -\tfrac{1}{2}\overline{\mathbf{J}}_u.$$

*Under correct specification $\mathbb{E}[\boldsymbol{u}_i(\boldsymbol{\beta}^\star)] = \mathbf{0}$,*

$$\boldsymbol{g}_n \xrightarrow{p} \mathbf{0}, \qquad -\mathbf{h}_n \xrightarrow{p} \tfrac{1}{4}\mathbf{S}, \qquad \mathbf{c}_n \xrightarrow{p} -\tfrac{1}{2}\mathbf{G},$$

*where $\mathbf{S} := \mathbb{E}[\boldsymbol{u}_i(\boldsymbol{\beta}^\star)\boldsymbol{u}_i(\boldsymbol{\beta}^\star)^\top]$ and $\mathbf{G} := \mathbb{E}[\nabla_{\boldsymbol{\beta}} \boldsymbol{u}_i(\boldsymbol{\beta}^\star)]$.*

*Proof of Lemma A5.* Let $\boldsymbol{\lambda} \in \mathbb{R}^q$ and $\boldsymbol{\beta} \in \mathbb{R}^d$. For each $i = 1, \ldots, N_{\text{in}}$, let $\boldsymbol{u}_i(\boldsymbol{\beta}) \in \mathbb{R}^q$ and

$$w_i(\boldsymbol{\lambda}, \boldsymbol{\beta}) := \exp\left(\tfrac{1}{2}\boldsymbol{\lambda}^\top \boldsymbol{u}_i(\boldsymbol{\beta})\right).$$

Define the sample mean

$$m(\boldsymbol{\lambda}, \boldsymbol{\beta}) := \frac{1}{N_{\text{in}}} \sum_{i=1}^{N_{\text{in}}} w_i(\boldsymbol{\lambda}, \boldsymbol{\beta}),$$

and the ET dual objective

$$H_{-1}(\boldsymbol{\lambda}, \boldsymbol{\beta}) := -N_{\text{in}} \log m(\boldsymbol{\lambda}, \boldsymbol{\beta}).$$

Let $D(\boldsymbol{\lambda}, \boldsymbol{\beta}) := \sum_{j=1}^{N_{\text{in}}} w_j(\boldsymbol{\lambda}, \boldsymbol{\beta}) = N_{\text{in}} m(\boldsymbol{\lambda}, \boldsymbol{\beta})$ and define softmax weights

$$p_i(\boldsymbol{\lambda}, \boldsymbol{\beta}) := \frac{w_i(\boldsymbol{\lambda}, \boldsymbol{\beta})}{D(\boldsymbol{\lambda}, \boldsymbol{\beta})}.$$

For any array $a_i$, write the $p$-weighted average $\mathbb{E}_p[a_i] := \sum_{i=1}^{N_{\text{in}}} p_i a_i$. Set

$$\boldsymbol{\mu}_u := \mathbb{E}[\boldsymbol{u}_i(\boldsymbol{\beta})] \in \mathbb{R}^q, \quad \boldsymbol{\Sigma}_u := \mathbb{E}[\boldsymbol{u}_i \boldsymbol{u}_i^\top] - \boldsymbol{\mu}_u \boldsymbol{\mu}_u^\top \in \mathbb{R}^{q \times q}, \quad \mathbf{J}_i(\boldsymbol{\beta}) := \nabla_{\boldsymbol{\beta}} \boldsymbol{u}_i(\boldsymbol{\beta}) \in \mathbb{R}^{q \times d}.$$

**(i) Gradient w.r.t. $\boldsymbol{\lambda}$.** By the chain rule,

$$\nabla_{\boldsymbol{\lambda}} H_{-1}(\boldsymbol{\lambda}, \boldsymbol{\beta}) = -N_{\text{in}} \frac{\nabla_{\boldsymbol{\lambda}} m}{m}.$$

Because

$$\nabla_{\boldsymbol{\lambda}} m = \frac{1}{N_{\text{in}}} \sum_{i=1}^{N_{\text{in}}} \nabla_{\boldsymbol{\lambda}} w_i = \frac{1}{N_{\text{in}}} \sum_{i=1}^{N_{\text{in}}} \left(\tfrac{1}{2}\boldsymbol{u}_i\right) w_i = \frac{D}{N_{\text{in}}} \cdot \frac{1}{2} \mathbb{E}_p[\boldsymbol{u}_i] = m \cdot \frac{1}{2}\boldsymbol{\mu}_u,$$

we get

$$\nabla_{\boldsymbol{\lambda}} H_{-1}(\boldsymbol{\lambda}, \boldsymbol{\beta}) = -\frac{N_{\text{in}}}{2}\boldsymbol{\mu}_u.$$

**(ii) Hessian w.r.t. $\boldsymbol{\lambda}$.** Differentiate the preceding display:

$$\nabla_{\boldsymbol{\lambda}\boldsymbol{\lambda}}^2 H_{-1}(\boldsymbol{\lambda}, \boldsymbol{\beta}) = -\frac{N_{\text{in}}}{2}\nabla_{\boldsymbol{\lambda}}\boldsymbol{\mu}_u.$$

Since $\boldsymbol{u}_i$ does not depend on $\boldsymbol{\lambda}$,

$$\nabla_{\boldsymbol{\lambda}}\boldsymbol{\mu}_u = \sum_i (\nabla_{\boldsymbol{\lambda}} p_i)\boldsymbol{u}_i^\top = \sum_i p_i (\nabla_{\boldsymbol{\lambda}} \log p_i)\boldsymbol{u}_i^\top.$$

But $\log p_i = \frac{1}{2}\boldsymbol{\lambda}^\top \boldsymbol{u}_i - \log D$, hence

$$\nabla_{\boldsymbol{\lambda}} \log p_i = \tfrac{1}{2}\boldsymbol{u}_i - \frac{1}{D}\sum_j w_j \tfrac{1}{2}\boldsymbol{u}_j = \tfrac{1}{2}\left(\boldsymbol{u}_i - \boldsymbol{\mu}_u\right).$$

Therefore

$$\nabla_{\boldsymbol{\lambda}}\boldsymbol{\mu}_u = \frac{1}{2}\left(\mathbb{E}_p[\boldsymbol{u}_i\boldsymbol{u}_i^\top] - \boldsymbol{\mu}_u\boldsymbol{\mu}_u^\top\right) = \frac{1}{2}\boldsymbol{\Sigma}_u,$$

and thus

$$\nabla_{\boldsymbol{\lambda}\boldsymbol{\lambda}}^2 H_{-1}(\boldsymbol{\lambda}, \boldsymbol{\beta}) = -\frac{N_{\text{in}}}{4}\boldsymbol{\Sigma}_u.$$

**(iii) Cross block $\nabla_{\boldsymbol{\lambda}\boldsymbol{\beta}}^2 H_{-1}$.** From (i),

$$\nabla_{\boldsymbol{\lambda}\boldsymbol{\beta}}^2 H_{-1}(\boldsymbol{\lambda}, \boldsymbol{\beta}) = -\frac{N_{\text{in}}}{2}\nabla_{\boldsymbol{\beta}}\boldsymbol{\mu}_u = -\frac{N_{\text{in}}}{2}\left(\sum_i (\nabla_{\boldsymbol{\beta}} p_i)\boldsymbol{u}_i^\top + \sum_i p_i \mathbf{J}_i\right).$$

Differentiate $\log p_i$ w.r.t. $\boldsymbol{\beta}$:

$$\nabla_{\boldsymbol{\beta}} \log p_i = \tfrac{1}{2}\mathbf{J}_i^\top \boldsymbol{\lambda} - \frac{1}{D}\sum_j w_j \tfrac{1}{2}\mathbf{J}_j^\top \boldsymbol{\lambda} = \tfrac{1}{2}\left(\mathbf{J}_i^\top \boldsymbol{\lambda} - \mathbb{E}_p[\mathbf{J}^\top \boldsymbol{\lambda}]\right).$$

Thus $\nabla_{\boldsymbol{\beta}} p_i = p_i \nabla_{\boldsymbol{\beta}} \log p_i$, and

$$\sum_i (\nabla_{\boldsymbol{\beta}} p_i)\boldsymbol{u}_i^\top = \frac{1}{2}\left(\mathbb{E}_p\left[\boldsymbol{u}_i(\mathbf{J}_i^\top \boldsymbol{\lambda})^\top\right]^\top - \boldsymbol{\mu}_u \mathbb{E}_p[\mathbf{J}^\top \boldsymbol{\lambda}]^\top\right).$$

Let $\boldsymbol{\mu}_{J\lambda} := \mathbb{E}_p[\mathbf{J}_i^\top \boldsymbol{\lambda}] \in \mathbb{R}^d$. Then

$$\nabla_{\boldsymbol{\lambda}\boldsymbol{\beta}}^2 H_{-1}(\boldsymbol{\lambda}, \boldsymbol{\beta}) = -\frac{N_{\text{in}}}{2}\mathbb{E}_p[\mathbf{J}_i] - \frac{N_{\text{in}}}{4}\left(\mathbb{E}_p[\boldsymbol{u}_i(\mathbf{J}_i^\top \boldsymbol{\lambda})^\top]^\top - \boldsymbol{\mu}_u\boldsymbol{\mu}_{J\lambda}^\top\right).$$

The last bracket is $\text{Cov}_p(\boldsymbol{u}_i, \mathbf{J}_i^\top \boldsymbol{\lambda})$.

**Specialization at $\boldsymbol{\lambda} = \mathbf{0}$.** When $\boldsymbol{\lambda} = \mathbf{0}$, $p_i = 1/N_{\text{in}}$, hence the weighted averages become simple averages:

$$\bar{\boldsymbol{u}} := \frac{1}{N_{\text{in}}}\sum_i \boldsymbol{u}_i, \quad \bar{\mathbf{S}} := \frac{1}{N_{\text{in}}}\sum_i \boldsymbol{u}_i\boldsymbol{u}_i^\top, \quad \overline{\mathbf{J}}_u := \frac{1}{N_{\text{in}}}\sum_i \mathbf{J}_i, \quad \text{and} \quad \boldsymbol{\mu}_{J\lambda} = \mathbf{0}.$$

Substituting into the general formulas yields the aggregate forms

$$\nabla_{\boldsymbol{\lambda}} H_{-1}(\mathbf{0}, \boldsymbol{\beta}) = -\frac{N_{\text{in}}}{2}\bar{\boldsymbol{u}}, \qquad \nabla_{\boldsymbol{\lambda}\boldsymbol{\lambda}}^2 H_{-1}(\mathbf{0}, \boldsymbol{\beta}) = -\frac{N_{\text{in}}}{4}\left(\bar{\mathbf{S}} - \bar{\boldsymbol{u}}\bar{\boldsymbol{u}}^\top\right),$$

$$\nabla_{\boldsymbol{\lambda}\boldsymbol{\beta}}^2 H_{-1}(\mathbf{0}, \boldsymbol{\beta}) = -\frac{N_{\text{in}}}{2}\overline{\mathbf{J}}_u.$$

These agree with the per-sample versions defined in the lemma after dividing by $N_{\text{in}}$. $\qquad\square$

### E.3 PROOF OF THEOREM 2

*Proof of Theorem 2.* Write the score $\boldsymbol{s}_i(\boldsymbol{\beta}) = \nabla_{\boldsymbol{\beta}} \log f_{\boldsymbol{\beta}}(Y_i \mid \mathbf{X}_i, \mathbf{Z}_i)$, the moment vector $\boldsymbol{u}_i(\boldsymbol{\beta}) \in \mathbb{R}^q$, $\mathbf{J} := \mathbb{E}[-\nabla_{\boldsymbol{\beta}\boldsymbol{\beta}}^2 \log f_{\boldsymbol{\beta}^\star}]$, $\mathbf{S} := \mathbb{E}[\boldsymbol{u}_i(\boldsymbol{\beta}^\star)\boldsymbol{u}_i(\boldsymbol{\beta}^\star)^\top]$, $\mathbf{G} := \mathbb{E}[\nabla_{\boldsymbol{\beta}} \boldsymbol{u}_i(\boldsymbol{\beta}^\star)]$. Let the Cressie-Read inner dual be

$$H_k(\boldsymbol{\lambda}, \boldsymbol{\beta}) = \frac{N_{\text{in}}}{k(k+1)} \left\{ \frac{1}{N_{\text{in}}} \sum_{i=1}^{N_{\text{in}}} \left(1 + \boldsymbol{\lambda}^\top \boldsymbol{u}_i(\boldsymbol{\beta})\right)^{k/(k+1)} \right\}^{k+1}.$$

Define the (outer) criterion

$$Q_{N_{\text{in}}}(\boldsymbol{\beta}) = -\sum_{i=1}^{N_{\text{in}}} \log f_{\boldsymbol{\beta}}(Y_i \mid \mathbf{X}_i, \mathbf{Z}_i) + \max_{\boldsymbol{\lambda} \in \Lambda} H_k(\boldsymbol{\lambda}, \boldsymbol{\beta}), \qquad \widehat{\boldsymbol{\beta}}_k \in \arg\min_{\boldsymbol{\beta}} Q_{N_{\text{in}}}(\boldsymbol{\beta}),$$

and let $\widehat{\boldsymbol{\lambda}}_k$ be a maximizer of $H_k(\cdot, \widehat{\boldsymbol{\beta}}_k)$. Assumptions (moved to Appendix C) ensure: (i) $t_i(\boldsymbol{\lambda}, \boldsymbol{\beta}) := 1 + \boldsymbol{\lambda}^\top \boldsymbol{u}_i(\boldsymbol{\beta}) > 0$ on $\Lambda \times \mathcal{N}(\boldsymbol{\beta}^\star)$; (ii) $\boldsymbol{u}_i$ is $C^1$ in $\boldsymbol{\beta}$; (iii) moments of order $2 + \delta$; (iv) inner program is concave in $\boldsymbol{\lambda}$ (Table 1 / Lemma A2); (v) identification at $(\boldsymbol{\beta}^\star, \boldsymbol{\lambda}^\star = \mathbf{0})$ under correct specification $\mathbb{E}[\boldsymbol{u}_i(\boldsymbol{\beta}^\star)] = \mathbf{0}$.

**KKT system and identification.** Define the stacked estimating map

$$\boldsymbol{\Psi}_{N_{\text{in}}}(\boldsymbol{\vartheta}) := \begin{pmatrix} \frac{1}{N_{\text{in}}} \sum_{i=1}^{N_{\text{in}}} \boldsymbol{s}_i(\boldsymbol{\beta}) - \frac{1}{N_{\text{in}}} \nabla_{\boldsymbol{\beta}} H_k(\boldsymbol{\lambda}, \boldsymbol{\beta}) \\ \frac{1}{N_{\text{in}}} \nabla_{\boldsymbol{\lambda}} H_k(\boldsymbol{\lambda}, \boldsymbol{\beta}) \end{pmatrix}, \qquad \boldsymbol{\vartheta} := \begin{pmatrix} \boldsymbol{\beta} \\ \boldsymbol{\lambda} \end{pmatrix}, \quad \widehat{\boldsymbol{\vartheta}}_k := \begin{pmatrix} \widehat{\boldsymbol{\beta}}_k \\ \widehat{\boldsymbol{\lambda}}_k \end{pmatrix}.$$

By construction, $\boldsymbol{\Psi}_{N_{\text{in}}}(\widehat{\boldsymbol{\vartheta}}_k) = \mathbf{0}$ (first line is the outer first-order condition (FOC); second line is the inner FOC). At the population optimum $\boldsymbol{\vartheta}^\star := (\boldsymbol{\beta}^{\star\top}, \mathbf{0}^\top)^\top$, correct specification implies

$$\mathbb{E}\left[\frac{1}{N_{\text{in}}} \sum_i \boldsymbol{s}_i(\boldsymbol{\beta}^\star)\right] = \mathbf{0}, \qquad \mathbb{E}\left[\frac{1}{N_{\text{in}}} \nabla_{\boldsymbol{\lambda}} H_k(\mathbf{0}, \boldsymbol{\beta}^\star)\right] = \mathbf{0},$$

hence $\mathbb{E}[\boldsymbol{\Psi}_{N_{\text{in}}}(\boldsymbol{\vartheta}^\star)] = \mathbf{0}$. Inner concavity (Lemma A2) yields uniqueness/continuity of $\boldsymbol{\lambda}^\star(\boldsymbol{\beta})$ in a neighborhood of $\boldsymbol{\beta}^\star$, so identification holds.

**Consistency.** By the uniform law of large numbers and continuity, $Q_{N_{\text{in}}}(\boldsymbol{\beta}) \to_p Q(\boldsymbol{\beta})$ uniformly on compact sets, where

$$Q(\boldsymbol{\beta}) = -\mathbb{E}[\log f_{\boldsymbol{\beta}}(Y \mid \mathbf{X}, \mathbf{Z})] + \max_{\boldsymbol{\lambda} \in \Lambda} \mathbb{E}[H_k(\boldsymbol{\lambda}, \boldsymbol{\beta})].$$

Under $\mathbb{E}[\boldsymbol{u}_i(\boldsymbol{\beta}^\star)] = \mathbf{0}$ and the inner FOC, $\boldsymbol{\lambda}^\star(\boldsymbol{\beta}^\star) = \mathbf{0}$ maximizes the population inner dual, and standard likelihood identification plus convexity of the inner problem ensure that $Q(\boldsymbol{\beta})$ is uniquely minimized at $\boldsymbol{\beta}^\star$. The argmin theorem thus gives $\widehat{\boldsymbol{\beta}}_k \to_p \boldsymbol{\beta}^\star$.

**Joint asymptotic normality of** $(\widehat{\boldsymbol{\beta}}_k, \widehat{\boldsymbol{\lambda}}_k)$**.** Apply a mean-value expansion of the stacked estimating equations at $\boldsymbol{\vartheta}^\star$:

$$\mathbf{0} = \boldsymbol{\Psi}_{N_{\text{in}}}(\widehat{\boldsymbol{\vartheta}}_k) = \boldsymbol{\Psi}_{N_{\text{in}}}(\boldsymbol{\vartheta}^\star) + \left[\nabla_{\boldsymbol{\vartheta}} \boldsymbol{\Psi}_{N_{\text{in}}}(\overline{\boldsymbol{\vartheta}})\right](\widehat{\boldsymbol{\vartheta}}_k - \boldsymbol{\vartheta}^\star),$$

for some $\overline{\boldsymbol{\vartheta}}$ on the line segment between $\widehat{\boldsymbol{\vartheta}}_k$ and $\boldsymbol{\vartheta}^\star$. Rearrange and scale:

$$\sqrt{N_{\text{in}}}(\widehat{\boldsymbol{\vartheta}}_k - \boldsymbol{\vartheta}^\star) = -\left[\nabla_{\boldsymbol{\vartheta}} \boldsymbol{\Psi}_{N_{\text{in}}}(\overline{\boldsymbol{\vartheta}})\right]^{-1} \sqrt{N_{\text{in}}} \boldsymbol{\Psi}_{N_{\text{in}}}(\boldsymbol{\vartheta}^\star).$$

By the CLT and regularity,

$$\sqrt{N_{\text{in}}} \boldsymbol{\Psi}_{N_{\text{in}}}(\boldsymbol{\vartheta}^\star) \to \mathcal{N}\left(\mathbf{0}, \begin{bmatrix} \mathbf{J} & \mathbf{0} \\ \mathbf{0} & \boldsymbol{\Omega}_1 \end{bmatrix}\right),$$

and by the LLN,

$$\nabla_{\boldsymbol{\vartheta}} \boldsymbol{\Psi}_{N_{\text{in}}}(\overline{\boldsymbol{\vartheta}}) \xrightarrow{p} -\begin{bmatrix} \mathbf{J} & -\mathbf{C}^\top \\ -\mathbf{C} & \boldsymbol{\Omega}_2 \end{bmatrix}.$$

Hence, by Slutsky,

$$\sqrt{N_{\text{in}}}\begin{pmatrix}\widehat{\boldsymbol{\beta}}_k - \boldsymbol{\beta}^\star \\ \widehat{\boldsymbol{\lambda}}_k\end{pmatrix} \to \mathcal{N}\left(\mathbf{0}, \begin{bmatrix}\mathbf{J} & -\mathbf{C}^\top \\ -\mathbf{C} & \boldsymbol{\Omega}_2\end{bmatrix}^{-1}\begin{bmatrix}\mathbf{J} & \mathbf{0} \\ \mathbf{0} & \boldsymbol{\Omega}_1\end{bmatrix}\begin{bmatrix}\mathbf{J} & -\mathbf{C}^\top \\ -\mathbf{C} & \boldsymbol{\Omega}_2\end{bmatrix}^{-1}\right).$$

**Profiled asymptotics for $\widehat{\boldsymbol{\beta}}_k$.** From the joint normal limit, profiling out $\boldsymbol{\lambda}$ (blockwise Schur complement) yields

$$\sqrt{N_{\text{in}}}(\widehat{\boldsymbol{\beta}}_k - \boldsymbol{\beta}^\star) \to \mathcal{N}(\mathbf{0}, \mathbf{A}^{-1}\mathbf{V}\mathbf{A}^{-1}),$$

$$\mathbf{A} := \mathbf{J} + \mathbf{C}^\top\boldsymbol{\Omega}_2^{-1}\mathbf{C}, \qquad \mathbf{V} := \mathbf{J} + \mathbf{C}^\top\boldsymbol{\Omega}_2^{-1}\boldsymbol{\Omega}_1\boldsymbol{\Omega}_2^{-1}\mathbf{C}.$$

**CR inner dual: $\boldsymbol{\Omega}_1 = \boldsymbol{\Omega}_2$ and $k$-invariance.** By Lemma A3 (gradient/Hessian of $H_k$ at $(\boldsymbol{\beta}^\star, \boldsymbol{\lambda} = \mathbf{0})$), under correct specification

$$\boldsymbol{\Omega}_1 := \frac{1}{N_{\text{in}}}\mathbb{E}\left[\{\nabla_{\boldsymbol{\lambda}}H_k\}\{\nabla_{\boldsymbol{\lambda}}H_k\}^\top\right] = \frac{1}{(k+1)^2}\mathbf{S},$$

$$\boldsymbol{\Omega}_2 := -\frac{1}{N_{\text{in}}}\mathbb{E}\left[\nabla^2_{\boldsymbol{\lambda}\boldsymbol{\lambda}}H_k\right] = \frac{1}{(k+1)^2}\mathbf{S}.$$

$$\mathbf{C} := \frac{1}{N_{\text{in}}}\mathbb{E}\left[\nabla^2_{\boldsymbol{\lambda}\boldsymbol{\beta}}H_k\right] = \frac{1}{(k+1)}\mathbf{G}.$$

Therefore $\boldsymbol{\Omega}_1 = \boldsymbol{\Omega}_2$, and the common factor $(k+1)^{-2}$ cancels in $\mathbf{A}$ and $\mathbf{V}$:

$$\mathbf{A} = \mathbf{J} + \mathbf{G}^\top\mathbf{S}^{-1}\mathbf{G}, \qquad \mathbf{V} = \mathbf{J} + \mathbf{G}^\top\mathbf{S}^{-1}\mathbf{G},$$

so

$$\sqrt{N_{\text{in}}}(\widehat{\boldsymbol{\beta}}_k - \boldsymbol{\beta}^\star) \to \mathcal{N}\left(\mathbf{0}, (\mathbf{J} + \mathbf{G}^\top\mathbf{S}^{-1}\mathbf{G})^{-1}\right).$$

$\square$

## APPENDIX F  PROOF OF THEOREM 3 UNDER DISTRIBUTIONAL SHIFT

**Lemma A6** (Influence expansion of the CR dual and additivity of first-order terms)**.** *Fix $(\boldsymbol{\lambda}, \boldsymbol{\beta})$ in the interior of the domain $\{(\boldsymbol{\lambda}, \boldsymbol{\beta}) : 1 + \boldsymbol{\lambda}^\top\boldsymbol{u}_{\boldsymbol{\beta}}(\mathbf{W}) > 0 \text{ a.s.}\}$. Let $P$ denote the population law of $\mathbf{W} = (Y, \mathbf{X}, \mathbf{Z})$ and $P_n$ the empirical measure. For the Cressie-Read dual,*

$$H_k(\boldsymbol{\lambda}, \boldsymbol{\beta}; P) := \frac{N_{\text{in}}}{k(k+1)}\left(A(P; \boldsymbol{\lambda}, \boldsymbol{\beta})\right)^{k+1}, \quad A(P; \boldsymbol{\lambda}, \boldsymbol{\beta}) := \mathbb{E}_P\left[(1 + \boldsymbol{\lambda}^\top\boldsymbol{u}_{\boldsymbol{\beta}}(\mathbf{W}))^\alpha\right], \quad \alpha := \frac{k}{k+1},$$

*assume $\boldsymbol{u}_{\boldsymbol{\beta}}$ is measurable, $P$-integrable to the needed orders, and that the maps $(\boldsymbol{\lambda}, \boldsymbol{\beta}) \mapsto \boldsymbol{u}_{\boldsymbol{\beta}}(\mathbf{w})$ are smooth. Then:*

(i) ***Gateaux derivative (influence function) of $H_k$.*** *Define $h_k(x) := \frac{N_{\text{in}}}{k(k+1)}x^{k+1}$ so that $H_k = h_k \circ A$. Then the influence function of $H_k$ at $P$ is*

$$\text{IF}_H(\mathbf{W}; \boldsymbol{\lambda}, \boldsymbol{\beta}; P) = \dot{H}_{k,P}(\mathbf{W}) = h'_k(A(P))\left(a(\mathbf{W}) - A(P)\right), \quad a(\mathbf{W}) := (1 + \boldsymbol{\lambda}^\top\boldsymbol{u}_{\boldsymbol{\beta}}(\mathbf{W}))^\alpha.$$

*Hence the linearization in the empirical process direction is*

$$H_k(\boldsymbol{\lambda}, \boldsymbol{\beta}; P_n) - H_k(\boldsymbol{\lambda}, \boldsymbol{\beta}; P) = \frac{1}{N_{\text{in}}}\sum_{i=1}^{N_{\text{in}}}\text{IF}_H(\mathbf{W}_i; \boldsymbol{\lambda}, \boldsymbol{\beta}; P) + o_p(N_{\text{in}}^{-1/2}).$$

(ii) ***Influence functions of the gradients.*** *Recall that subscripts denote partial derivatives with respect to components of $\boldsymbol{\vartheta} = (\boldsymbol{\beta}^\top, \boldsymbol{\lambda}^\top)^\top$. Write*

$$\boldsymbol{A}_{\boldsymbol{\vartheta}} := \nabla_{\boldsymbol{\vartheta}}A \in \mathbb{R}^{d+q}, \qquad \mathbf{A}_{\boldsymbol{\vartheta}\boldsymbol{\vartheta}} := \nabla^2_{\boldsymbol{\vartheta}\boldsymbol{\vartheta}}A \in \mathbb{R}^{(d+q)\times(d+q)}.$$

*By the chain rule,*

$$\nabla_{\boldsymbol{\vartheta}}H_k = h'_k(A)\boldsymbol{A}_{\boldsymbol{\vartheta}}, \qquad \nabla^2_{\boldsymbol{\vartheta}\boldsymbol{\vartheta}}H_k = h''_k(A)\boldsymbol{A}_{\boldsymbol{\vartheta}}\boldsymbol{A}_{\boldsymbol{\vartheta}}^\top + h'_k(A)\mathbf{A}_{\boldsymbol{\vartheta}\boldsymbol{\vartheta}}.$$

*Moreover, the influence functions of the gradients are*

$$\mathrm{IF}_{\nabla_{\vartheta} H_k}(\mathbf{W}; \boldsymbol{\lambda}, \boldsymbol{\beta}; P) = h_k''(A)(a(\mathbf{W}) - A)\boldsymbol{A}_{\vartheta} + h_k'(A)\big(\boldsymbol{A}_{\vartheta}(\mathbf{W}) - \boldsymbol{A}_{\vartheta}\big),$$

*where $\boldsymbol{A}_{\vartheta}(\mathbf{W})$ is obtained by differentiating $a(\mathbf{W})$ pointwise in $\vartheta$, and $\boldsymbol{A}_{\vartheta} = \mathbb{E}_P[\boldsymbol{A}_{\vartheta}(\mathbf{W})]$. In particular, the* centered *linear terms* $\mathrm{IF}_{\nabla_{\vartheta} H_k}(\mathbf{W}) - \mathbb{E}_P[\mathrm{IF}_{\nabla_{\vartheta} H_k}(\mathbf{W})]$ *are additive in* $\mathbf{W}$.

*(iii)* **Additive "meat" for the stacked score.** *Consider the stacked KKT map*

$$\boldsymbol{\Psi}(P, \vartheta) := \begin{pmatrix} \mathbb{E}_P[\boldsymbol{s}(\boldsymbol{\beta})] - \nabla_{\boldsymbol{\beta}} H_k(\boldsymbol{\lambda}, \boldsymbol{\beta}; P)/N_{\mathrm{in}} \\ \nabla_{\boldsymbol{\lambda}} H_k(\boldsymbol{\lambda}, \boldsymbol{\beta}; P)/N_{\mathrm{in}} \end{pmatrix}, \qquad \vartheta := (\boldsymbol{\beta}^{\top}, \boldsymbol{\lambda}^{\top})^{\top}.$$

*Although $H_k$ itself is not additively separable, its empirical fluctuations are linearized by the influence functions in (i)-(ii). Hence*

$$\sqrt{N_{\mathrm{in}}}\big(\boldsymbol{\Psi}(P_n, \vartheta) - \boldsymbol{\Psi}(P, \vartheta)\big) = \frac{1}{\sqrt{N_{\mathrm{in}}}} \sum_{i=1}^{N_{\mathrm{in}}} \underbrace{\begin{pmatrix} \boldsymbol{s}_i(\boldsymbol{\beta}) - \mathbb{E}_P[\boldsymbol{s}(\boldsymbol{\beta})] - \frac{1}{N_{\mathrm{in}}}\mathrm{IF}_{\nabla_{\boldsymbol{\beta}} H_k}(\mathbf{W}_i) \\ \frac{1}{N_{\mathrm{in}}}\mathrm{IF}_{\nabla_{\boldsymbol{\lambda}} H_k}(\mathbf{W}_i) \end{pmatrix}}_{:= \boldsymbol{\varphi}(\mathbf{W}_i; \vartheta, P)} + o_p(1).$$

*Thus, the asymptotic "meat" of the sandwich covariance is*

$$\boldsymbol{\mathcal{B}}(\vartheta) := \mathrm{Var}_P\big(\boldsymbol{\varphi}(\mathbf{W}; \vartheta, P)\big),$$

*which is a variance of per-sample contributions.*

*(iv)* **CR specialization near $\boldsymbol{\lambda} = \mathbf{0}$.** *At $\boldsymbol{\lambda} = \mathbf{0}$ we have $A(P) = 1$ and $h_k'(1) = N_{\mathrm{in}}/k$. Writing $\boldsymbol{g}(\mathbf{W}) := \boldsymbol{u}_{\boldsymbol{\beta}}(\mathbf{W})$ and recalling $\alpha = \frac{k}{k+1}$,*

$$\mathrm{IF}_{\nabla_{\boldsymbol{\lambda}} H_k}(\mathbf{W}) = h_k'(1)\big(\boldsymbol{A}_{\boldsymbol{\lambda}}(\mathbf{W}) - \boldsymbol{A}_{\boldsymbol{\lambda}}\big) = \frac{N_{\mathrm{in}}}{k}\alpha\big(\boldsymbol{g}(\mathbf{W}) - \mathbb{E}_P[\boldsymbol{g}(\mathbf{W})]\big) \propto \boldsymbol{g}(\mathbf{W}) - \mathbb{E}_P[\boldsymbol{g}(\mathbf{W})].$$

*Consequently, the leading blocks of $\boldsymbol{\mathcal{B}}^{\dagger}$ reduce (up to known multiplicative constants that cancel in the small-shift CR expansions) to the familiar*

$$\mathbf{K}^{\dagger} = \mathrm{Var}\big(\boldsymbol{s}_i(\boldsymbol{\beta}^{\dagger})\big), \qquad \boldsymbol{\Omega}_1^{\dagger} = \mathrm{Var}\big(\boldsymbol{g}_i\big), \qquad \mathbf{K}_{sg}^{\dagger} = \mathrm{Cov}\big(\boldsymbol{s}_i(\boldsymbol{\beta}^{\dagger}), \boldsymbol{g}_i\big),$$

*with $\boldsymbol{g}_i := \boldsymbol{u}_i(\boldsymbol{\beta}^{\dagger})$.*

*Proof sketch.* (i) View $H_k = h_k \circ A$ with $A(P) = \int a(\mathbf{W})dP$. The Gateaux derivative in the direction $\delta_{\mathbf{w}} - P$ is $\dot{A}_P(\mathbf{W}) = a(\mathbf{W}) - A(P)$. Apply the chain rule to get $\dot{H}_{k,P}(\mathbf{W}) = h_k'(A)\dot{A}_P(\mathbf{W})$.

(ii) Differentiate $H_k$ w.r.t. parameters via the chain rule; then take the Gateaux derivative in $P$ again. The term $\boldsymbol{A}_{\vartheta}(\mathbf{W}) - \boldsymbol{A}_{\vartheta}$ comes from linearizing $\int \boldsymbol{A}_{\vartheta}(\mathbf{W})dP$ and the factor $h_k''(A)(a(\mathbf{W}) - A)\boldsymbol{A}_{\vartheta}$ comes from differentiating $h_k'(A)$.

(iii) Stack the score for $\boldsymbol{\beta}$ with the dual score for $\boldsymbol{\lambda}$ and linearize $\boldsymbol{\Psi}(P_n, \vartheta) - \boldsymbol{\Psi}(P, \vartheta)$ using (i)-(ii). This yields a sum of i.i.d. mean-zero terms, which defines the additive influence vector $\boldsymbol{\varphi}(\mathbf{W}_i; \vartheta, P)$.

(iv) Substitute $\boldsymbol{\lambda} = \mathbf{0}$ and simplify $a(\mathbf{W}) = (1 + \boldsymbol{\lambda}^{\top}\boldsymbol{g}(\mathbf{W}))^{\alpha}$ and its derivatives. The leading constants cancel in the CR small-shift profiles, giving the standard block forms used in the main theorem. $\square$

**Theorem A6** (Asymptotic normality of DRO estimator under shift: full version). *Assume the regularity conditions in Appendix D. Let $(\widehat{\boldsymbol{\beta}}_k, \widehat{\boldsymbol{\lambda}}_k)$ solve the sample KKT system of the DRO objective with Cressie-Read index $k$. Define the stacked estimating map*

$$\boldsymbol{\Psi}_{N_{\mathrm{in}}}(\vartheta) := \begin{pmatrix} \frac{1}{N_{\mathrm{in}}} \sum_{i=1}^{N_{\mathrm{in}}} \boldsymbol{s}_i(\boldsymbol{\beta}) - \frac{1}{N_{\mathrm{in}}}\nabla_{\boldsymbol{\beta}} H_k(\boldsymbol{\lambda}, \boldsymbol{\beta}) \\ \frac{1}{N_{\mathrm{in}}}\nabla_{\boldsymbol{\lambda}} H_k(\boldsymbol{\lambda}, \boldsymbol{\beta}) \end{pmatrix}, \qquad \vartheta := \begin{pmatrix} \boldsymbol{\beta} \\ \boldsymbol{\lambda} \end{pmatrix},$$

where $\boldsymbol{s}_i(\boldsymbol{\beta}) = \nabla_{\boldsymbol{\beta}}\ell_i(\boldsymbol{\beta})$ and $\ell_i(\boldsymbol{\beta}) = \log f_{\boldsymbol{\beta}}(Y_i \mid \mathbf{X}_i, \mathbf{Z}_i)$. Let $\widehat{\boldsymbol{\vartheta}}_k$ be the solution to $\boldsymbol{\Psi}_{N_{\mathrm{in}}}(\widehat{\boldsymbol{\vartheta}}_k) = \mathbf{0}$, and let $\boldsymbol{\vartheta}^\dagger = (\boldsymbol{\beta}^{\dagger\top}, \boldsymbol{\lambda}^{\dagger\top})^\top$ be the unique population solution satisfying $\mathbb{E}[\boldsymbol{\Psi}_{N_{\mathrm{in}}}(\boldsymbol{\vartheta}^\dagger)] = \mathbf{0}$.

**Bread and meat.** Define the bread *and* meat *as*

$$\boldsymbol{\mathcal{A}}^\dagger := \mathbb{E}\big[\nabla_{\boldsymbol{\vartheta}} \boldsymbol{\Psi}_{N_{\mathrm{in}}}(\boldsymbol{\vartheta}^\dagger)\big], \qquad \boldsymbol{\mathcal{B}}^\dagger := \mathrm{Var}\big(\boldsymbol{\Phi}_i(\boldsymbol{\vartheta}^\dagger)\big),$$

*where the per-sample contribution* $\boldsymbol{\Phi}_i$ *is*

$$\boldsymbol{\Phi}_i(\boldsymbol{\vartheta}^\dagger) := \begin{pmatrix} \boldsymbol{s}_i(\boldsymbol{\beta}^\dagger) + \boldsymbol{H}_{\boldsymbol{\beta}}^{(i)}(\boldsymbol{\lambda}^\dagger, \boldsymbol{\beta}^\dagger) \\ \boldsymbol{H}_{\boldsymbol{\lambda}}^{(i)}(\boldsymbol{\lambda}^\dagger, \boldsymbol{\beta}^\dagger) \end{pmatrix},$$

*and* (definition) *the terms* $\boldsymbol{H}_{\boldsymbol{\beta}}^{(i)}, \boldsymbol{H}_{\boldsymbol{\lambda}}^{(i)}$ *are the* scaled influence contributions *of* $H_k$ *at* $\boldsymbol{\vartheta}^\dagger$*, namely*

$$\boldsymbol{H}_{\boldsymbol{\beta}}^{(i)} := -\frac{1}{N_{\mathrm{in}}}\mathrm{IF}_{\nabla_{\boldsymbol{\beta}} H_k}(\mathbf{W}_i; \boldsymbol{\vartheta}^\dagger; P), \qquad \boldsymbol{H}_{\boldsymbol{\lambda}}^{(i)} := \frac{1}{N_{\mathrm{in}}}\mathrm{IF}_{\nabla_{\boldsymbol{\lambda}} H_k}(\mathbf{W}_i; \boldsymbol{\vartheta}^\dagger; P),$$

*so that* $\frac{1}{\sqrt{N_{\mathrm{in}}}}\sum_i \boldsymbol{\Phi}_i$ *is precisely the linearization of* $\sqrt{N_{\mathrm{in}}}\big(\boldsymbol{\Psi}(P_n, \boldsymbol{\vartheta}^\dagger) - \boldsymbol{\Psi}(P, \boldsymbol{\vartheta}^\dagger)\big)$ *(see Lemma A6).*

**Block form.** *Then* $\boldsymbol{\mathcal{A}}^\dagger$ *and* $\boldsymbol{\mathcal{B}}^\dagger$ *admit the block decompositions*

$$\boldsymbol{\mathcal{A}}^\dagger = \begin{bmatrix} \mathbf{J}^\dagger + \mathbf{C}_{\boldsymbol{\beta}\boldsymbol{\beta}}^\dagger & -(\mathbf{C}^\dagger)^\top \\ -\mathbf{C}^\dagger & \boldsymbol{\Omega}_2^\dagger \end{bmatrix}, \qquad \boldsymbol{\mathcal{B}}^\dagger = \begin{bmatrix} \mathbf{K}^\dagger & \mathbf{K}_{sg}^\dagger \\ (\mathbf{K}_{sg}^\dagger)^\top & \boldsymbol{\Omega}_1^\dagger \end{bmatrix},$$

*with*

$$\mathbf{J}^\dagger := \mathbb{E}\big[-\nabla_{\boldsymbol{\beta}\boldsymbol{\beta}}^2 \ell_i(\boldsymbol{\beta}^\dagger)\big],$$

$$\mathbf{C}_{\boldsymbol{\beta}\boldsymbol{\beta}}^\dagger := -\frac{1}{N_{\mathrm{in}}}\mathbb{E}\big[\nabla_{\boldsymbol{\beta}\boldsymbol{\beta}}^2 H_k(\boldsymbol{\lambda}^\dagger, \boldsymbol{\beta}^\dagger)\big], \qquad \mathbf{C}^\dagger := \frac{1}{N_{\mathrm{in}}}\mathbb{E}\big[\nabla_{\boldsymbol{\lambda}\boldsymbol{\beta}}^2 H_k(\boldsymbol{\lambda}^\dagger, \boldsymbol{\beta}^\dagger)\big],$$

$$\boldsymbol{\Omega}_2^\dagger := -\frac{1}{N_{\mathrm{in}}}\mathbb{E}\big[\nabla_{\boldsymbol{\lambda}\boldsymbol{\lambda}}^2 H_k(\boldsymbol{\lambda}^\dagger, \boldsymbol{\beta}^\dagger)\big],$$

$$\mathbf{K}^\dagger := \mathrm{Var}\big(\boldsymbol{s}_i(\boldsymbol{\beta}^\dagger) + \boldsymbol{H}_{\boldsymbol{\beta}}^{(i)}(\boldsymbol{\lambda}^\dagger, \boldsymbol{\beta}^\dagger)\big), \quad \mathbf{K}_{sg}^\dagger := \mathrm{Cov}\big(\boldsymbol{s}_i(\boldsymbol{\beta}^\dagger) + \boldsymbol{H}_{\boldsymbol{\beta}}^{(i)}(\boldsymbol{\lambda}^\dagger, \boldsymbol{\beta}^\dagger), \boldsymbol{H}_{\boldsymbol{\lambda}}^{(i)}(\boldsymbol{\lambda}^\dagger, \boldsymbol{\beta}^\dagger)\big),$$

$$\boldsymbol{\Omega}_1^\dagger := \mathrm{Var}\big(\boldsymbol{H}_{\boldsymbol{\lambda}}^{(i)}(\boldsymbol{\lambda}^\dagger, \boldsymbol{\beta}^\dagger)\big).$$

(Remarks.) *(i) The signs and* $1/N_{\mathrm{in}}$ *factors match* $\boldsymbol{\Psi}_{N_{\mathrm{in}}}$*: the minus sign on* $\nabla_{\boldsymbol{\beta}} H_k$ *in* $\boldsymbol{\Psi}_{N_{\mathrm{in}}}$ *is absorbed by the definition of* $\boldsymbol{H}_{\boldsymbol{\beta}}^{(i)}$ *above. (ii)* $H_k$ *is the aggregate dual penalty from Eq. 2; its second derivatives scale like* $N_{\mathrm{in}}$*, and the explicit* $1/N_{\mathrm{in}}$ *factors above ensure each block of* $\boldsymbol{\mathcal{A}}^\dagger$ *is* $O(1)$.

**Limit distribution.** *Then*

$$\sqrt{N_{\mathrm{in}}}\big(\widehat{\boldsymbol{\vartheta}}_k - \boldsymbol{\vartheta}^\dagger\big) \xrightarrow{d} \mathcal{N}\big(\mathbf{0}, (\boldsymbol{\mathcal{A}}^\dagger)^{-1}\boldsymbol{\mathcal{B}}^\dagger(\boldsymbol{\mathcal{A}}^\dagger)^{-\top}\big),$$

*where* $(\boldsymbol{\mathcal{A}}^\dagger)^{-\top} := \big((\boldsymbol{\mathcal{A}}^\dagger)^{-1}\big)^\top$. *In particular, the marginal limit for* $\widehat{\boldsymbol{\beta}}_k$ *is*

$$\sqrt{N_{\mathrm{in}}}(\widehat{\boldsymbol{\beta}}_k - \boldsymbol{\beta}^\dagger) \xrightarrow{d} \mathcal{N}\big(\mathbf{0}, \mathbf{V}_{\beta}^\dagger\big),$$

*with the profile sandwich variance*

$$\mathbf{V}_{\beta}^\dagger = (\mathbf{A}_{\beta}^\dagger)^{-1}\Big(\mathbf{K}^\dagger + (\mathbf{C}^\dagger)^\top(\boldsymbol{\Omega}_2^\dagger)^{-1}\boldsymbol{\Omega}_1^\dagger(\boldsymbol{\Omega}_2^\dagger)^{-1}\mathbf{C}^\dagger + (\mathbf{C}^\dagger)^\top(\boldsymbol{\Omega}_2^\dagger)^{-1}\mathbf{K}_{sg}^\dagger + (\mathbf{K}_{sg}^\dagger)^\top(\boldsymbol{\Omega}_2^\dagger)^{-1}\mathbf{C}^\dagger\Big)(\mathbf{A}_{\beta}^\dagger)^{-1},$$

*where*

$$\mathbf{A}_{\beta}^\dagger := \mathbf{J}^\dagger + \mathbf{C}_{\boldsymbol{\beta}\boldsymbol{\beta}}^\dagger + (\mathbf{C}^\dagger)^\top(\boldsymbol{\Omega}_2^\dagger)^{-1}\mathbf{C}^\dagger.$$

*Proof of Theorem A6.* Let $P$ denote the population law of $\mathbf{W} = (Y, \mathbf{X}, \mathbf{Z})$ and $P_n := N_{\mathrm{in}}^{-1}\sum_{i=1}^{N_{\mathrm{in}}}\delta_{\mathbf{W}_i}$ the empirical measure. Define the stacked KKT map (as a functional of $(P, \boldsymbol{\vartheta})$)

$$\boldsymbol{\Psi}(P, \boldsymbol{\vartheta}) := \begin{pmatrix} \mathbb{E}_P\big[\boldsymbol{s}_i(\boldsymbol{\beta})\big] - \frac{1}{N_{\mathrm{in}}}\nabla_{\boldsymbol{\beta}} H_k(\boldsymbol{\lambda}, \boldsymbol{\beta}; P) \\ \frac{1}{N_{\mathrm{in}}}\nabla_{\boldsymbol{\lambda}} H_k(\boldsymbol{\lambda}, \boldsymbol{\beta}; P) \end{pmatrix}, \qquad \boldsymbol{\Psi}_n(\boldsymbol{\vartheta}) := \boldsymbol{\Psi}(P_n, \boldsymbol{\vartheta}),$$

where $H_k(\boldsymbol{\lambda}, \boldsymbol{\beta}; P)$ is the CR dual objective with expectation under $P$ replacing the empirical average. For the CR family, $H_k$ is a smooth composition of linear functionals of $P$ (e.g., $A(\boldsymbol{\lambda}, \boldsymbol{\beta}; P) = \int (1 + \boldsymbol{\lambda}^\top \boldsymbol{u}_{\boldsymbol{\beta}})^\alpha dP$ with $\alpha = \frac{k}{k+1}$) and smooth maps (power, log), so $\boldsymbol{\Psi}(\cdot, \cdot)$ is Hadamard differentiable jointly in $(P, \boldsymbol{\vartheta})$ in a neighborhood of $(P, \boldsymbol{\vartheta}^\dagger)$.

**Consistency.** By assumption, the population KKT system $\boldsymbol{\Psi}(P, \boldsymbol{\vartheta}) = \boldsymbol{0}$ has a unique solution $\boldsymbol{\vartheta}^\dagger = (\boldsymbol{\beta}^{\dagger\top}, \boldsymbol{\lambda}^{\dagger\top})^\top$. A uniform law of large numbers for $\{\boldsymbol{\Psi}(P, \boldsymbol{\vartheta}) : \boldsymbol{\vartheta} \in \Theta\}$ (together with identification; see Appendix D) implies $\widehat{\boldsymbol{\vartheta}}_k \to_p \boldsymbol{\vartheta}^\dagger$.

**Joint linearization.** A first-order Fréchet expansion of $\boldsymbol{\Psi}$ in both arguments around $(P, \boldsymbol{\vartheta}^\dagger)$ gives

$$\boldsymbol{0} = \boldsymbol{\Psi}(P_n, \widehat{\boldsymbol{\vartheta}}_k) = \underbrace{\boldsymbol{\Psi}(P, \boldsymbol{\vartheta}^\dagger)}_{=\boldsymbol{0}} + \underbrace{\dot{\boldsymbol{\Psi}}_{\boldsymbol{\vartheta}}(P, \boldsymbol{\vartheta}^\dagger)}_{=:\boldsymbol{\mathcal{A}}^\dagger}(\widehat{\boldsymbol{\vartheta}}_k - \boldsymbol{\vartheta}^\dagger) + \underbrace{\dot{\boldsymbol{\Psi}}_P(P, \boldsymbol{\vartheta}^\dagger)[P_n - P]}_{=:\Delta_n} + r_n,$$

where $\dot{\boldsymbol{\Psi}}_{\boldsymbol{\vartheta}}$ is the Jacobian w.r.t. $\boldsymbol{\vartheta}$ and $\dot{\boldsymbol{\Psi}}_P(P, \boldsymbol{\vartheta}^\dagger)[\cdot]$ is the Gâteaux derivative in the direction of a signed measure. Hadamard differentiability plus $\widehat{\boldsymbol{\vartheta}}_k \to_p \boldsymbol{\vartheta}^\dagger$ imply $r_n = o_p(N_{\text{in}}^{-1/2})$. Differentiating $\boldsymbol{\Psi}$ at $\boldsymbol{\vartheta}^\dagger$ yields

$$\boldsymbol{\mathcal{A}}^\dagger = \nabla_{\boldsymbol{\vartheta}} \boldsymbol{\Psi}(P, \boldsymbol{\vartheta})\big|_{\boldsymbol{\vartheta}=\boldsymbol{\vartheta}^\dagger} = \begin{bmatrix} \mathbf{J}^\dagger + \mathbf{C}_{\boldsymbol{\beta\beta}}^\dagger & -(\mathbf{C}^\dagger)^\top \\ -\mathbf{C}^\dagger & \boldsymbol{\Omega}_2^\dagger \end{bmatrix},$$

with the blocks defined exactly as in the theorem (note the explicit $1/N_{\text{in}}$ factors inherited from $\boldsymbol{\Psi}$).

**Influence representation.** By the functional delta method (e.g., Van der Vaart (1998, Thm. 20.8)),

$$\sqrt{N_{\text{in}}} \Delta_n = \frac{1}{\sqrt{N_{\text{in}}}} \sum_{i=1}^{N_{\text{in}}} \boldsymbol{\varphi}_i + o_p(1),$$

for a mean-zero influence vector $\boldsymbol{\varphi}_i$ obtained by applying Lemma A6 componentwise to $\boldsymbol{\Psi}$. Writing the observation-level contributions to the $H_k$ gradients as

$$\boldsymbol{H}_{\boldsymbol{\beta}}^{(i)}(\boldsymbol{\lambda}^\dagger, \boldsymbol{\beta}^\dagger) := -\frac{1}{N_{\text{in}}} \text{IF}_{\nabla_{\boldsymbol{\beta}} H_k}(\mathbf{W}_i; \boldsymbol{\vartheta}^\dagger; P), \qquad \boldsymbol{H}_{\boldsymbol{\lambda}}^{(i)}(\boldsymbol{\lambda}^\dagger, \boldsymbol{\beta}^\dagger) := \frac{1}{N_{\text{in}}} \text{IF}_{\nabla_{\boldsymbol{\lambda}} H_k}(\mathbf{W}_i; \boldsymbol{\vartheta}^\dagger; P),$$

we can take

$$\boldsymbol{\varphi}_i = \begin{pmatrix} \boldsymbol{s}_i(\boldsymbol{\beta}^\dagger) + \boldsymbol{H}_{\boldsymbol{\beta}}^{(i)}(\boldsymbol{\lambda}^\dagger, \boldsymbol{\beta}^\dagger) \\ \boldsymbol{H}_{\boldsymbol{\lambda}}^{(i)}(\boldsymbol{\lambda}^\dagger, \boldsymbol{\beta}^\dagger) \end{pmatrix} - \mathbb{E}\left[ \begin{pmatrix} \boldsymbol{s}_i(\boldsymbol{\beta}^\dagger) + \boldsymbol{H}_{\boldsymbol{\beta}}^{(i)}(\boldsymbol{\lambda}^\dagger, \boldsymbol{\beta}^\dagger) \\ \boldsymbol{H}_{\boldsymbol{\lambda}}^{(i)}(\boldsymbol{\lambda}^\dagger, \boldsymbol{\beta}^\dagger) \end{pmatrix} \right].$$

Its covariance $\boldsymbol{\mathcal{B}}^\dagger := \text{Var}(\boldsymbol{\varphi}_i)$ has the block form stated in the theorem:

$$\boldsymbol{\mathcal{B}}^\dagger = \begin{bmatrix} \mathbf{K}^\dagger & \mathbf{K}_{sg}^\dagger \\ (\mathbf{K}_{sg}^\dagger)^\top & \boldsymbol{\Omega}_1^\dagger \end{bmatrix}.$$

**Joint asymptotic normality.** Rearranging the linearization,

$$\sqrt{N_{\text{in}}}(\widehat{\boldsymbol{\vartheta}}_k - \boldsymbol{\vartheta}^\dagger) = -(\boldsymbol{\mathcal{A}}^\dagger)^{-1} \sqrt{N_{\text{in}}} \Delta_n + o_p(1) \xrightarrow{d} \mathcal{N}(\boldsymbol{0}, \boldsymbol{\mathcal{A}}^{\dagger-1} \boldsymbol{\mathcal{B}}^\dagger \boldsymbol{\mathcal{A}}^{\dagger-\top}).$$

**Profiling to obtain $\mathbf{V}_{\boldsymbol{\beta}}^\dagger$.** Write $\boldsymbol{\mathcal{A}}^\dagger = \begin{bmatrix} \mathbf{A} & \mathbf{B} \\ \mathbf{B}^\top & \mathbf{D} \end{bmatrix}$ with $\mathbf{A} := \mathbf{J}^\dagger + \mathbf{C}_{\boldsymbol{\beta\beta}}^\dagger$, $\mathbf{B} := -\mathbf{C}^\dagger$, $\mathbf{D} := \boldsymbol{\Omega}_2^\dagger$, and $\boldsymbol{\mathcal{B}}^\dagger = \begin{bmatrix} \mathbf{K}^\dagger & \mathbf{K}_{sg}^\dagger \\ (\mathbf{K}_{sg}^\dagger)^\top & \boldsymbol{\Omega}_1^\dagger \end{bmatrix}$. The Schur complement of $\mathbf{D}$ in $\boldsymbol{\mathcal{A}}^\dagger$ is

$$\mathbf{A}_{\boldsymbol{\beta}}^\dagger := \mathbf{A} - \mathbf{B}\mathbf{D}^{-1}\mathbf{B}^\top = \mathbf{J}^\dagger + \mathbf{C}_{\boldsymbol{\beta\beta}}^\dagger + (\mathbf{C}^\dagger)^\top (\boldsymbol{\Omega}_2^\dagger)^{-1} \mathbf{C}^\dagger.$$

Standard block-matrix algebra gives the profiled covariance of the $\boldsymbol{\beta}$-component:

$$\mathbf{V}_{\boldsymbol{\beta}}^\dagger = (\mathbf{A}_{\boldsymbol{\beta}}^\dagger)^{-1} \Big( \mathbf{K}^\dagger + (\mathbf{C}^\dagger)^\top (\boldsymbol{\Omega}_2^\dagger)^{-1} \boldsymbol{\Omega}_1^\dagger (\boldsymbol{\Omega}_2^\dagger)^{-1} \mathbf{C}^\dagger + (\mathbf{C}^\dagger)^\top (\boldsymbol{\Omega}_2^\dagger)^{-1} \mathbf{K}_{sg}^\dagger + (\mathbf{K}_{sg}^\dagger)^\top (\boldsymbol{\Omega}_2^\dagger)^{-1} \mathbf{C}^\dagger \Big) (\mathbf{A}_{\boldsymbol{\beta}}^\dagger)^{-1}.$$

**Comment on non-additivity.** Although $\nabla_{\boldsymbol{\beta}} H_k$ and $\nabla_{\boldsymbol{\lambda}} H_k$ are not simple sums over $i$, they are smooth functionals of empirical means (such as $A(P_n)$, $\log A(P_n)$, powers), hence admit linear influence representations in $(P_n - P)$. This is exactly what enters the preceding expansion via Lemma A6, so the standard Z-estimation machinery applies.

All steps are standard for Z-estimators with smooth (Hadamard differentiable) scores; see, e.g., Newey and McFadden (1994, Sec. 7) and Van der Vaart (1998, Ch. 5, 20). □

## APPENDIX G  LOCAL EXPANSION UNDER DISTRIBUTIONAL SHIFT

**Lemma A7** (First-order bias expansion under shift). *Let the assumptions of Theorem A6 hold and expand the KKT system locally around $\boldsymbol{\vartheta}^\star = (\boldsymbol{\beta}^\star, \mathbf{0})$. Define the shift $\boldsymbol{\delta} := \mathbb{E}[\boldsymbol{u}_i(\boldsymbol{\beta}^\star)] \in \mathbb{R}^q$ and*

$$\mathbf{J} := \mathbb{E}\big[ - \nabla_{\boldsymbol{\beta}\boldsymbol{\beta}}^2 \ell_i(\boldsymbol{\beta}^\star)\big], \quad \mathbf{G} := \mathbb{E}\big[\nabla_{\boldsymbol{\beta}} \boldsymbol{u}_i(\boldsymbol{\beta}^\star)\big], \quad \mathbf{S} := \mathbb{E}\big[\boldsymbol{u}_i(\boldsymbol{\beta}^\star)\boldsymbol{u}_i(\boldsymbol{\beta}^\star)^\top\big].$$

*Assume $\mathbf{S}$ is positive definite and $\mathbf{A}_\beta := \mathbf{J} - \mathbf{G}^\top \mathbf{S}^{-1} \mathbf{G}$ is nonsingular. Then, for the Cressie-Read dual with $\|\widehat{\boldsymbol{\lambda}}_k\| = o_p(1)$ (small shift),*

$$\widehat{\boldsymbol{\beta}}_k - \boldsymbol{\beta}^\star = \big(\mathbf{J} - \mathbf{G}^\top \mathbf{S}^{-1} \mathbf{G}\big)^{-1} \mathbf{G}^\top \mathbf{S}^{-1} \boldsymbol{\delta} + O_p\big(N_{\mathrm{in}}^{-1/2}\big) + o(\|\boldsymbol{\delta}\|).$$

*In particular, the leading (deterministic) bias is $k$-free; dependence on $k$ enters only through higher-order terms when the shift is not infinitesimal.*

*Proof of Lemma A7.* Let $P$ be the population law and $P_n$ the empirical measure. Work with the scaled KKT map

$$\boldsymbol{\Psi}_{N_{\mathrm{in}}}(\boldsymbol{\vartheta}) := \begin{pmatrix} \dfrac{1}{N_{\mathrm{in}}} \sum_{i=1}^{N_{\mathrm{in}}} \boldsymbol{s}_i(\boldsymbol{\beta}) - \dfrac{1}{N_{\mathrm{in}}} \nabla_{\boldsymbol{\beta}} H_k(\boldsymbol{\lambda}, \boldsymbol{\beta}; P_n) \\ \dfrac{1}{N_{\mathrm{in}}} \nabla_{\boldsymbol{\lambda}} H_k(\boldsymbol{\lambda}, \boldsymbol{\beta}; P_n) \end{pmatrix}, \qquad \boldsymbol{\vartheta} = \begin{pmatrix} \boldsymbol{\beta} \\ \boldsymbol{\lambda} \end{pmatrix}.$$

At the population level and at $\boldsymbol{\vartheta}^\star = (\boldsymbol{\beta}^\star, \mathbf{0})$, its population counterpart is

$$\boldsymbol{\Psi}(P, \boldsymbol{\vartheta}^\star) := \begin{pmatrix} \mathbb{E}_P[\boldsymbol{s}_i(\boldsymbol{\beta}^\star)] - \frac{1}{N_{\mathrm{in}}} \nabla_{\boldsymbol{\beta}} H_k(\mathbf{0}, \boldsymbol{\beta}^\star; P) \\ \frac{1}{N_{\mathrm{in}}} \nabla_{\boldsymbol{\lambda}} H_k(\mathbf{0}, \boldsymbol{\beta}^\star; P) \end{pmatrix}.$$

For the Cressie-Read (CR) family ($k \neq 0, -1$),

$$\frac{1}{N_{\mathrm{in}}} \nabla_{\boldsymbol{\lambda}} H_k(\mathbf{0}, \boldsymbol{\beta}^\star; P) = \frac{1}{k+1} \mathbb{E}_P[\boldsymbol{u}_i(\boldsymbol{\beta}^\star)] = \frac{1}{k+1} \boldsymbol{\delta}, \qquad \frac{1}{N_{\mathrm{in}}} \nabla_{\boldsymbol{\beta}} H_k(\mathbf{0}, \boldsymbol{\beta}^\star; P) = \mathbf{0},$$

so $\boldsymbol{\Psi}(P, \boldsymbol{\vartheta}^\star) = (\mathbf{0}, c_k \boldsymbol{\delta})^\top$ with $c_k = \frac{1}{k+1}$.

Now take a first-order expansion of $\boldsymbol{\Psi}(P, \cdot)$ around $\boldsymbol{\vartheta}^\star$:

$$\mathbf{0} = \boldsymbol{\Psi}(P, \widehat{\boldsymbol{\vartheta}}_k) = \boldsymbol{\Psi}(P, \boldsymbol{\vartheta}^\star) + \boldsymbol{\mathcal{A}}^\star \begin{pmatrix} \widehat{\boldsymbol{\beta}}_k - \boldsymbol{\beta}^\star \\ \widehat{\boldsymbol{\lambda}}_k - \mathbf{0} \end{pmatrix} + \boldsymbol{R}_1, \qquad \boldsymbol{R}_1 = o\big(\|\widehat{\boldsymbol{\vartheta}}_k - \boldsymbol{\vartheta}^\star\|\big),$$

where the Jacobian at $(\boldsymbol{\beta}^\star, \mathbf{0})$ is

$$\boldsymbol{\mathcal{A}}^\star := \mathbb{E}\big[\nabla_{\boldsymbol{\vartheta}} \boldsymbol{\Psi}_{N_{\mathrm{in}}}(\boldsymbol{\vartheta}^\star)\big] = \begin{bmatrix} \mathbf{J} & -(\mathbf{C}^\star)^\top \\ -\mathbf{C}^\star & \boldsymbol{\Omega}_2^\star \end{bmatrix}, \quad \mathbf{J} := \mathbb{E}\big[ - \nabla_{\boldsymbol{\beta}\boldsymbol{\beta}}^2 \ell_i(\boldsymbol{\beta}^\star)\big].$$

For CR at $\boldsymbol{\lambda} = \mathbf{0}$ (under the same scaling by $1/N_{\mathrm{in}}$ used in $\boldsymbol{\Psi}_{N_{\mathrm{in}}}$),

$$\mathbf{C}^\star = \frac{1}{k+1} \mathbf{G}, \qquad \boldsymbol{\Omega}_2^\star = \frac{1}{(k+1)^2} \mathbf{S},$$

$$\mathbf{G} := \mathbb{E}[\nabla_{\boldsymbol{\beta}} \boldsymbol{u}_i(\boldsymbol{\beta}^\star)], \qquad \mathbf{S} := \mathbb{E}[\boldsymbol{u}_i(\boldsymbol{\beta}^\star)\boldsymbol{u}_i(\boldsymbol{\beta}^\star)^\top].$$

Ignoring sampling noise for the bias calculation and solving the linear system $\boldsymbol{\mathcal{A}}^\star \big( \widehat{\boldsymbol{\beta}}_k - \boldsymbol{\beta}^\star, \ \widehat{\boldsymbol{\lambda}}_k \big)^\top = -(\mathbf{0}, \ c_k \boldsymbol{\delta})^\top$ by Schur complement yields

$$\widehat{\boldsymbol{\beta}}_k - \boldsymbol{\beta}^\star = \big( \mathbf{J} - (\mathbf{C}^\star)^\top (\boldsymbol{\Omega}_2^\star)^{-1} \mathbf{C}^\star \big)^{-1} (\mathbf{C}^\star)^\top (\boldsymbol{\Omega}_2^\star)^{-1} (c_k \boldsymbol{\delta}) \ + \ \text{(higher order)}.$$

Now use

$$(\mathbf{C}^\star)^\top (\boldsymbol{\Omega}_2^\star)^{-1} = \left( \tfrac{1}{k+1} \mathbf{G}^\top \right) \big( (k+1)^2 \mathbf{S}^{-1} \big) = (k+1) \mathbf{G}^\top \mathbf{S}^{-1},$$

so that the factor $c_k = \frac{1}{k+1}$ cancels $(k+1)$ exactly:

$$(\mathbf{C}^\star)^\top (\boldsymbol{\Omega}_2^\star)^{-1} (c_k \boldsymbol{\delta}) = \mathbf{G}^\top \mathbf{S}^{-1} \boldsymbol{\delta}.$$

Also,

$$(\mathbf{C}^\star)^\top (\boldsymbol{\Omega}_2^\star)^{-1} \mathbf{C}^\star = \mathbf{G}^\top \mathbf{S}^{-1} \mathbf{G}.$$

Hence the deterministic first-order term simplifies to

$$\widehat{\boldsymbol{\beta}}_k - \boldsymbol{\beta}^\star = \big( \mathbf{J} - \mathbf{G}^\top \mathbf{S}^{-1} \mathbf{G} \big)^{-1} \mathbf{G}^\top \mathbf{S}^{-1} \boldsymbol{\delta} \ + \ \text{(higher order)}.$$

Finally, (i) the empirical-process remainder contributes $O_p(N_{\text{in}}^{-1/2})$ to $\widehat{\boldsymbol{\beta}}_k - \boldsymbol{\beta}^\star$ by Theorem A6; (ii) the Taylor remainder is $o(\|\boldsymbol{\delta}\|)$ by smoothness (Hadamard differentiability in $P$ and $C^2$ in $(\boldsymbol{\lambda}, \boldsymbol{\beta})$) near $(\mathbf{0}, \boldsymbol{\beta}^\star)$. Collecting terms gives

$$\widehat{\boldsymbol{\beta}}_k - \boldsymbol{\beta}^\star = \big( \mathbf{J} - \mathbf{G}^\top \mathbf{S}^{-1} \mathbf{G} \big)^{-1} \mathbf{G}^\top \mathbf{S}^{-1} \boldsymbol{\delta} + O_p(N_{\text{in}}^{-1/2}) + o(\|\boldsymbol{\delta}\|),$$

as claimed. $\qquad \square$

**Interpretation.** The bias expansion (Lemma A7) shows that misspecification enters through the shift term $\boldsymbol{\delta}$, with bias proportional to $\big( \mathbf{J} - \mathbf{G}^\top \mathbf{S}^{-1} \mathbf{G} \big)^{-1} \mathbf{G}^\top \mathbf{S}^{-1} \boldsymbol{\delta}$. Hence the estimator is stable under mild shifts ($\|\boldsymbol{\delta}\|$ small) but deteriorates when the shift is large.

## APPENDIX H  INNER/OUTER DERIVATIVES AND IMPLICIT DIFFERENTIATION (GENERAL $k$)

Let $N := N_{\text{in}}$. For a fixed $\boldsymbol{\beta}$, define

$$t_i(\boldsymbol{\lambda}, \boldsymbol{\beta}) := \boldsymbol{\lambda}^\top \boldsymbol{u}_i(\boldsymbol{\beta}), \qquad m_i := \begin{cases} -\log(1 + t_i), & k = 0 \\ \exp\left( \frac{t_i}{2} \right), & k = -1 \\ (1 + t_i)^{\frac{k}{k+1}}, & k < 0, \ k \neq -1 \\ -(1 + t_i)^{\frac{k}{k+1}}, & k > 0, \end{cases} \qquad M := \sum_{i=1}^N m_i.$$

The dual objective can be written as $H_k(\boldsymbol{\lambda}, \boldsymbol{\beta}) = h_k(M)$ with

$$h_k(m) = \begin{cases} -m, & k = 0 \\ -N \log(m/N), & k = -1 \\ \dfrac{N^{-k}}{k(k+1)} m^{k+1}, & k < 0, \ k \neq -1 \\ \dfrac{N^{-k}}{k(k+1)} (-m)^{k+1}, & k > 0. \end{cases}$$

Hence

$$h_k'(m) = \begin{cases} -1, & k = 0 \\ -\dfrac{N}{m}, & k = -1 \\ \dfrac{N^{-k}}{k} m^k, & k < 0, \ k \neq -1 \\ -\dfrac{N^{-k}}{k} (-m)^k, & k > 0 \end{cases} \qquad h_k''(m) = \begin{cases} 0, & k = 0 \\ \dfrac{N}{m^2}, & k = -1 \\ N^{-k} m^{k-1}, & k < 0, \ k \neq -1 \\ N^{-k} (-m)^{k-1}, & k > 0. \end{cases}$$

**Lemma A8** (Derivatives for the inner and outer problems). *For each $i$, let $\mathbf{J}_i(\boldsymbol{\beta}) := \nabla_{\boldsymbol{\beta}} \boldsymbol{u}_i(\boldsymbol{\beta}) \in \mathbb{R}^{q \times d}$ and $\mathbf{H}_i(\boldsymbol{\beta}) := \nabla^2_{\boldsymbol{\beta}\boldsymbol{\beta}} \boldsymbol{u}_i(\boldsymbol{\beta}) \in \mathbb{R}^{q \times d \times d}$.*

*(A) Inner derivatives (w.r.t. $\boldsymbol{\lambda}$).*

$$M_{\boldsymbol{\lambda}} = \sum_{i=1}^{N} m_i'(t_i) \boldsymbol{u}_i, \qquad M_{\boldsymbol{\lambda}\boldsymbol{\lambda}} = \sum_{i=1}^{N} m_i''(t_i) \boldsymbol{u}_i \boldsymbol{u}_i^{\top}.$$

*Here*

$$m_i'(t) = \begin{cases} -\dfrac{1}{1+t}, & k = 0 \\[2mm] \dfrac{1}{2} e^{t/2}, & k = -1 \\[2mm] \dfrac{k}{k+1}(1+t)^{-1/(k+1)}, & k < 0,\ k \neq -1 \\[2mm] -\dfrac{k}{k+1}(1+t)^{-1/(k+1)}, & k > 0 \end{cases}$$

$$m_i''(t) = \begin{cases} \dfrac{1}{(1+t)^2}, & k = 0 \\[2mm] \dfrac{1}{4} e^{t/2}, & k = -1 \\[2mm] -\dfrac{k}{(k+1)^2}(1+t)^{-(k+2)/(k+1)}, & k < 0,\ k \neq -1 \\[2mm] \dfrac{k}{(k+1)^2}(1+t)^{-(k+2)/(k+1)}, & k > 0. \end{cases}$$

*(B) Cross derivatives (w.r.t. $(\boldsymbol{\lambda}, \boldsymbol{\beta})$). For each coordinate $\beta^{[\ell]}$ (row index) and $\lambda^{[k]}$ (column index),*

$$[M_{\boldsymbol{\lambda},\boldsymbol{\beta}}]_{k\ell} = \sum_{i=1}^{N} \left( m_i'(t_i)[\mathbf{J}_i]_{k\ell} + m_i''(t_i)(\mathbf{J}_i^{\top}\boldsymbol{\lambda})_{\ell} u_i^{[k]} \right).$$

*Equivalently, in matrix form*

$$M_{\boldsymbol{\lambda},\boldsymbol{\beta}} = \sum_{i=1}^{N} \left( m_i'(t_i)\mathbf{J}_i + m_i''(t_i)\boldsymbol{u}_i(\mathbf{J}_i^{\top}\boldsymbol{\lambda})^{\top} \right) \in \mathbb{R}^{q \times d}.$$

*(C) Implicit differentiation of the inner solution. Let $\widehat{\boldsymbol{\lambda}}(\boldsymbol{\beta})$ solve the inner FOC $M_{\boldsymbol{\lambda}}(\widehat{\boldsymbol{\lambda}}(\boldsymbol{\beta}), \boldsymbol{\beta}) = \mathbf{0}$. If $M_{\boldsymbol{\lambda}\boldsymbol{\lambda}}$ is nonsingular at $(\widehat{\boldsymbol{\lambda}}, \boldsymbol{\beta})$, then*

$$\frac{d\widehat{\boldsymbol{\lambda}}}{d\boldsymbol{\beta}} = -\left( M_{\boldsymbol{\lambda}\boldsymbol{\lambda}} \right)^{-1} M_{\boldsymbol{\lambda},\boldsymbol{\beta}}.$$

*(D) Total derivatives of $M(\widehat{\boldsymbol{\lambda}}(\boldsymbol{\beta}), \boldsymbol{\beta})$ w.r.t. $\boldsymbol{\beta}$. Using $M_{\boldsymbol{\lambda}} = 0$ at the inner optimum:*

$$M_{\boldsymbol{\beta}} = \frac{dM}{d\boldsymbol{\beta}} = \sum_{i=1}^{N} m_i'(t_i)\mathbf{J}_i^{\top}\boldsymbol{\lambda} \equiv \mathbf{A}\boldsymbol{\lambda}, \quad \mathbf{A} := \sum_{i=1}^{N} m_i'(t_i)\mathbf{J}_i^{\top} \in \mathbb{R}^{d \times q}.$$

*The exact Hessian splits into a "direct" term and a Schur-complement term:*

$$M_{\boldsymbol{\beta}\boldsymbol{\beta}} = \underbrace{\sum_{i=1}^{N} \left( m_i'(t_i)\mathcal{H}_i^{\top}\boldsymbol{\lambda} + m_i''(t_i)(\mathbf{J}_i^{\top}\boldsymbol{\lambda})(\mathbf{J}_i^{\top}\boldsymbol{\lambda})^{\top} \right)}_{M_{\boldsymbol{\beta}\boldsymbol{\beta}}^{\mathrm{dir}}} - \underbrace{M_{\boldsymbol{\lambda},\boldsymbol{\beta}}^{\top} M_{\boldsymbol{\lambda}\boldsymbol{\lambda}}^{-1} M_{\boldsymbol{\lambda},\boldsymbol{\beta}}}_{\textit{Schur term}},$$

*where $\mathcal{H}_i$ stacks the slices of $\mathbf{H}_i$ so that $\mathcal{H}_i^\top \boldsymbol{\lambda} \in \mathbb{R}^d$ has entries $\sum_k \lambda^{[k]} \partial^2 u_i^{[k]} / \partial\beta^{[\ell]} \partial\beta^{[q]}$.*

***(E) Total derivatives of the composed dual*** $H_k(\widehat{\boldsymbol{\lambda}}(\boldsymbol{\beta}), \boldsymbol{\beta}) = h_k(M)$. *By the chain rule,*

$$H_{\boldsymbol{\beta}} = h_k'(M) M_{\boldsymbol{\beta}}, \qquad H_{\boldsymbol{\beta}\boldsymbol{\beta}} = h_k''(M) M_{\boldsymbol{\beta}} M_{\boldsymbol{\beta}}^\top + h_k'(M) M_{\boldsymbol{\beta}\boldsymbol{\beta}}.$$

***(F) Near-$\boldsymbol{\lambda} = 0$ simplification.*** *If $\|\widehat{\boldsymbol{\lambda}}\|$ is small (e.g., under mild shift), then $m_i''(t_i) = m_i''(0) + o(1)$ and the quadratic/second-derivative terms in $M_{\boldsymbol{\beta}\boldsymbol{\beta}}^{\mathrm{dir}}$ are $o(1)$, giving the useful approximation*

$$M_{\boldsymbol{\beta}\boldsymbol{\beta}} \approx -M_{\boldsymbol{\lambda},\boldsymbol{\beta}}^\top M_{\boldsymbol{\lambda}\boldsymbol{\lambda}}^{-1} M_{\boldsymbol{\lambda},\boldsymbol{\beta}}, \qquad H_{\boldsymbol{\beta}\boldsymbol{\beta}} \approx h_k''(M) M_{\boldsymbol{\beta}} M_{\boldsymbol{\beta}}^\top - h_k'(M) M_{\boldsymbol{\lambda},\boldsymbol{\beta}}^\top M_{\boldsymbol{\lambda}\boldsymbol{\lambda}}^{-1} M_{\boldsymbol{\lambda},\boldsymbol{\beta}}.$$

*Proof (sketch).* (A)-(B) follow by the chain rule with $t_i = \boldsymbol{\lambda}^\top \boldsymbol{u}_i(\boldsymbol{\beta})$. (C) applies the implicit function theorem to the inner FOC $M_{\boldsymbol{\lambda}}(\widehat{\boldsymbol{\lambda}}(\boldsymbol{\beta}), \boldsymbol{\beta}) = \mathbf{0}$. (D) uses the total derivative

$$\frac{dM}{d\boldsymbol{\beta}} = M_{\boldsymbol{\beta}} + M_{\boldsymbol{\lambda}} \frac{d\widehat{\boldsymbol{\lambda}}}{d\boldsymbol{\beta}},$$

and the FOC $M_{\boldsymbol{\lambda}} = 0$ at the inner optimum; differentiating once more and substituting the implicit-diff formula in (C) yields the decomposition into the "direct" term and the Schur complement term for $M_{\boldsymbol{\beta}\boldsymbol{\beta}}$. (E) is a direct application of the chain rule to $H_k = h_k \circ M$. (F) drops terms that are $O(\|\widehat{\boldsymbol{\lambda}}\|)$ (or quadratic in $\widehat{\boldsymbol{\lambda}}$), which is justified under mild shift where the dual multipliers concentrate near $\mathbf{0}$.

$\square$

# APPENDIX I    EMPIRICAL BAYES DERIVATION OF EB-DRO

## I.1    EMPIRICAL BAYES DERIVATION OF EB-DRO

**Hierarchical model.** We model $\boldsymbol{\beta}$ hierarchically as

$$\boldsymbol{\beta} \mid \widehat{\boldsymbol{\beta}}_{\mathrm{DRO},k} \sim \mathcal{N}(\widehat{\boldsymbol{\beta}}_{\mathrm{DRO},k}, \mathbf{A}), \qquad \widehat{\boldsymbol{\beta}}_I \mid \boldsymbol{\beta} \sim \mathcal{N}(\boldsymbol{\beta}, \boldsymbol{\Sigma}),$$

where $\widehat{\boldsymbol{\beta}}_{\mathrm{DRO},k}$ is the DRO estimator, $\widehat{\boldsymbol{\beta}}_I$ is the internal-only GLM MLE, $\mathbf{A}$ quantifies prior uncertainty around the DRO solution, and $\boldsymbol{\Sigma}$ is the covariance of the internal estimator.

**Posterior mean.** By Gaussian conjugacy,

$$\mathbb{E}\left[\boldsymbol{\beta} \mid \widehat{\boldsymbol{\beta}}_{\mathrm{DRO},k}, \widehat{\boldsymbol{\beta}}_I\right] = \left(\mathbf{A}^{-1} + \boldsymbol{\Sigma}^{-1}\right)^{-1} \left(\mathbf{A}^{-1}\widehat{\boldsymbol{\beta}}_{\mathrm{DRO},k} + \boldsymbol{\Sigma}^{-1}\widehat{\boldsymbol{\beta}}_I\right),$$

yielding the EB estimator in Section 2.2. Equivalently, using matrix identities,

$$\widehat{\boldsymbol{\beta}}_{\mathrm{EB},k} = (\mathbf{I} - \mathbf{W})\widehat{\boldsymbol{\beta}}_{\mathrm{DRO},k} + \mathbf{W}\widehat{\boldsymbol{\beta}}_I, \qquad \mathbf{W} = \mathbf{A}(\mathbf{A} + \boldsymbol{\Sigma})^{-1},$$

which shows EB-DRO as a weighted average of the DRO and naive estimators.

**Empirical Bayes estimates.** In practice, we estimate the unknown quantities as follows:

- $\boldsymbol{\Sigma}$ is replaced by $\widehat{\boldsymbol{\Sigma}}_I$, the covariance estimator of $\widehat{\boldsymbol{\beta}}_I$ from the internal GLM.
- $\mathbf{A}$ is taken as
$$\widehat{\mathbf{A}} = \left(\widehat{\boldsymbol{\beta}}_I - \widehat{\boldsymbol{\beta}}_{\mathrm{DRO},k}\right)\left(\widehat{\boldsymbol{\beta}}_I - \widehat{\boldsymbol{\beta}}_{\mathrm{DRO},k}\right)^\top,$$
a rank-one matrix encoding the discrepancy between the two estimators.

Substituting $(\widehat{\mathbf{A}}, \widehat{\boldsymbol{\Sigma}}_I)$ into the posterior mean yields the implementable EB estimator $\widehat{\boldsymbol{\beta}}_{\mathrm{EB},k}$. Since $\widehat{\mathbf{A}}$ is rank-one, we interpret $\widehat{\mathbf{A}}^{-1}$ in the pseudoinverse sense (or equivalently as the $\varepsilon \downarrow 0$ limit of $\widehat{\mathbf{A}} + \varepsilon\mathbf{I}$), while $\widehat{\mathbf{A}} + \widehat{\boldsymbol{\Sigma}}_I$ is positive definite and invertible by construction.

### I.2 PROOF OF PROPOSITION 1

In this section, we show that the matrix-valued weight $\widehat{\mathbf{W}}$ from the definition of the EB-DRO estimator can be simplified to a scalar multiplication by $\alpha$.

Let $\mathbf{Z} := \widehat{\boldsymbol{\beta}}_I - \widehat{\boldsymbol{\beta}}_{\mathrm{DRO},k}$, $\widehat{\mathbf{A}} := \mathbf{Z}\mathbf{Z}^\top$, and $\widehat{\boldsymbol{\Sigma}}_I \succ 0$ denote the covariance estimator of $\widehat{\boldsymbol{\beta}}_I$. Recall the EB-DRO estimator in matrix-average form

$$\widehat{\boldsymbol{\beta}}_{\mathrm{EB},k} = (\mathbf{I} - \widehat{\mathbf{W}})\widehat{\boldsymbol{\beta}}_{\mathrm{DRO},k} + \widehat{\mathbf{W}}\widehat{\boldsymbol{\beta}}_I, \qquad \widehat{\mathbf{W}} := \widehat{\mathbf{A}}(\widehat{\mathbf{A}} + \widehat{\boldsymbol{\Sigma}}_I)^{-1}.$$

(Equivalently, this follows from Gaussian conjugacy together with $(\mathbf{A}^{-1} + \boldsymbol{\Sigma}^{-1})^{-1}\mathbf{A}^{-1} = \boldsymbol{\Sigma}(\mathbf{A} + \boldsymbol{\Sigma})^{-1}$ and $(\mathbf{A}^{-1} + \boldsymbol{\Sigma}^{-1})^{-1}\boldsymbol{\Sigma}^{-1} = \mathbf{A}(\mathbf{A} + \boldsymbol{\Sigma})^{-1}$.)

Set $s := \mathbf{Z}^\top \widehat{\boldsymbol{\Sigma}}_I^{-1} \mathbf{Z} \geq 0$. By the Sherman-Morrison-Woodbury identity for a rank-one update,

$$(\widehat{\boldsymbol{\Sigma}}_I + \mathbf{Z}\mathbf{Z}^\top)^{-1} = \widehat{\boldsymbol{\Sigma}}_I^{-1} - \widehat{\boldsymbol{\Sigma}}_I^{-1}\mathbf{Z}(1 + s)^{-1}\mathbf{Z}^\top \widehat{\boldsymbol{\Sigma}}_I^{-1}.$$

Multiplying both sides by $\mathbf{Z}$ gives

$$(\widehat{\boldsymbol{\Sigma}}_I + \mathbf{Z}\mathbf{Z}^\top)^{-1}\mathbf{Z} = \widehat{\boldsymbol{\Sigma}}_I^{-1}\mathbf{Z}\Big(1 - \frac{s}{1+s}\Big) = \frac{1}{1+s}\widehat{\boldsymbol{\Sigma}}_I^{-1}\mathbf{Z}. \tag{7}$$

Using $\widehat{\mathbf{W}} = \mathbf{Z}\mathbf{Z}^\top(\widehat{\boldsymbol{\Sigma}}_I + \mathbf{Z}\mathbf{Z}^\top)^{-1}$ and Eq. 7,

$$\widehat{\mathbf{W}}\mathbf{Z} = \mathbf{Z}\mathbf{Z}^\top \frac{1}{1+s}\widehat{\boldsymbol{\Sigma}}_I^{-1}\mathbf{Z} = \frac{s}{1+s}\mathbf{Z}.$$

Define

$$\alpha := \frac{s}{1+s} = \frac{\mathbf{Z}^\top \widehat{\boldsymbol{\Sigma}}_I^{-1}\mathbf{Z}}{1 + \mathbf{Z}^\top \widehat{\boldsymbol{\Sigma}}_I^{-1}\mathbf{Z}} \in [0, 1).$$

Since $\widehat{\boldsymbol{\beta}}_{\mathrm{EB},k} = \widehat{\boldsymbol{\beta}}_{\mathrm{DRO},k} + \widehat{\mathbf{W}}\mathbf{Z}$, we obtain

$$\widehat{\boldsymbol{\beta}}_{\mathrm{EB},k} = \widehat{\boldsymbol{\beta}}_{\mathrm{DRO},k} + \alpha\mathbf{Z} = (1 - \alpha)\widehat{\boldsymbol{\beta}}_{\mathrm{DRO},k} + \alpha\widehat{\boldsymbol{\beta}}_I,$$

which is Eq. 3. Moreover, $\alpha = 0$ iff $\mathbf{Z} = \mathbf{0}$ (the two estimators coincide), and $\alpha \uparrow 1$ as the Mahalanobis discrepancy $s$ grows, proving the stated range $\alpha \in (0, 1)$ for $\mathbf{Z} \neq \mathbf{0}$. $\qquad\square$

### I.3 PROOF OF THEOREM 4

*Proof of Theorem 4.* Write

$$\widehat{\boldsymbol{\beta}}(w) = (1 - w)\widehat{\boldsymbol{\beta}}_{\mathrm{DRO},k} + w\widehat{\boldsymbol{\beta}}_I, \qquad R(w) = \mathbb{E}\big\|\widehat{\boldsymbol{\beta}}(w) - \boldsymbol{\beta}^\star\big\|_2^2.$$

Let $\boldsymbol{b}_k = \mathbb{E}[\widehat{\boldsymbol{\beta}}_{\mathrm{DRO},k}] - \boldsymbol{\beta}^\star$, $\mathbf{V}_{\mathrm{DRO},k} = \mathrm{Var}(\widehat{\boldsymbol{\beta}}_{\mathrm{DRO},k})$, $\boldsymbol{\Sigma}_I = \mathrm{Var}(\widehat{\boldsymbol{\beta}}_I)$, and $\mathbf{C}_k = \mathrm{Cov}(\widehat{\boldsymbol{\beta}}_{\mathrm{DRO},k}, \widehat{\boldsymbol{\beta}}_I)$. Then

$$\mathbb{E}\big[\widehat{\boldsymbol{\beta}}(w) - \boldsymbol{\beta}^\star\big] = (1 - w)\boldsymbol{b}_k, \quad \mathrm{Var}\big(\widehat{\boldsymbol{\beta}}(w)\big) = (1 - w)^2\mathbf{V}_{\mathrm{DRO},k} + w^2\boldsymbol{\Sigma}_I + 2w(1 - w)\mathbf{C}_k.$$

Hence

$$R(w) = \|(1 - w)\boldsymbol{b}_k\|_2^2 + \mathrm{tr}\Big[(1 - w)^2\mathbf{V}_{\mathrm{DRO},k} + w^2\boldsymbol{\Sigma}_I + 2w(1 - w)\mathbf{C}_k\Big]$$

$$= (1 - w)^2 a + w^2 c + 2w(1 - w)m,$$

with $a = \|\boldsymbol{b}_k\|_2^2 + \mathrm{tr}(\mathbf{V}_{\mathrm{DRO},k})$, $c = \mathrm{tr}(\boldsymbol{\Sigma}_I)$, $m = \mathrm{tr}(\mathbf{C}_k)$. Expanding gives the quadratic

$$R(w) = a + (-2a + 2m)w + (a + c - 2m)w^2.$$

Let $\kappa := a + c - 2m = \|\boldsymbol{b}_k\|_2^2 + \mathrm{tr}\,\mathrm{Var}(\widehat{\boldsymbol{\beta}}_{\mathrm{DRO},k} - \widehat{\boldsymbol{\beta}}_I) \geq 0$. The unconstrained minimizer is $w_{\mathbb{R}}^\star = (a - m)/\kappa$ (if $\kappa = 0$, $R(w)$ is constant and any $w$ is minimizer). Completing the square yields the **exact identity**

$$R(w) = R(w_{\mathbb{R}}^\star) + \kappa(w - w_{\mathbb{R}}^\star)^2, \qquad (\kappa \geq 0). \tag{$*$}$$

Projecting onto $[0, 1]$ gives the oracle weight $w^\star = \Pi_{[0,1]}(w_{\mathbb{R}}^\star)$, which minimizes $R(w)$ over $[0, 1]$ by convexity. Therefore
$$R(w^\star) \leq \min\{R(0), R(1)\},$$
establishing the dominance claim. Finally, using $R(w_{\mathbb{R}}^\star) \leq R(w^\star) \leq \min\{R(0), R(1)\}$ in $(*)$ gives the **regret bound**
$$R(w) = R(w_{\mathbb{R}}^\star) + \kappa(w - w_{\mathbb{R}}^\star)^2 \leq \min\{R(0), R(1)\} + \kappa(w - w_{\mathbb{R}}^\star)^2.$$

If $w^\star \in (0, 1)$ (no projection), then $w^\star = w_{\mathbb{R}}^\star$ and the regret identity simplifies to $R(w) = R(w^\star) + \kappa(w - w^\star)^2$. $\qquad\square$

## APPENDIX J   PROOF OF THEOREM 5

**Setup and notation.** Stack the $K$ EB-DRO estimators as
$$\widehat{\mathbf{B}} = \begin{bmatrix} \widehat{\boldsymbol{\beta}}_{\text{EB},1} \\ \vdots \\ \widehat{\boldsymbol{\beta}}_{\text{EB},K} \end{bmatrix} \in \mathbb{R}^{Kd}, \qquad \mathbb{E}[\widehat{\mathbf{B}}] = (\mathbf{1}_K \otimes \mathbf{I}_d)(\boldsymbol{\beta}^\star + \boldsymbol{b}).$$

Let $\mathbf{V} = \text{Cov}(\widehat{\mathbf{B}})$ with $d \times d$ blocks $\boldsymbol{\Omega}_{k\ell} = \text{Cov}(\widehat{\boldsymbol{\beta}}_{\text{EB},k}, \widehat{\boldsymbol{\beta}}_{\text{EB},\ell})$, and write the block-diagonal of marginal covariances as
$$\mathbf{D} = \text{diag}(\boldsymbol{\Omega}_{11}, \ldots, \boldsymbol{\Omega}_{KK}).$$

Let $\mathbf{H} = \mathbf{1}_K \otimes \mathbf{I}_p \in \mathbb{R}^{Kd \times d}$, so that $\mathbb{E}[\widehat{\mathbf{B}}] = \mathbf{H}(\boldsymbol{\beta}^\star + \boldsymbol{b})$. The *oracle* GLS estimator of $\boldsymbol{\beta}^\star + \boldsymbol{b}$ from $\widehat{\mathbf{B}}$ is
$$\widetilde{\boldsymbol{\beta}}_{\text{Ens}} = (\mathbf{H}^\top \mathbf{V}^{-1} \mathbf{H})^{-1} \mathbf{H}^\top \mathbf{V}^{-1} \widehat{\mathbf{B}}, \qquad \text{Cov}(\widetilde{\boldsymbol{\beta}}_{\text{Ens}}) = \mathbf{G}(\mathbf{V}) := (\mathbf{H}^\top \mathbf{V}^{-1} \mathbf{H})^{-1}.$$

The *precision-weighted* (PW) ensemble replaces $\mathbf{V}$ by the block-diagonal precision:
$$\widehat{\boldsymbol{\beta}}_{\text{PW}} = (\mathbf{H}^\top \widehat{\mathbf{D}}^{-1} \mathbf{H})^{-1} \mathbf{H}^\top \widehat{\mathbf{D}}^{-1} \widehat{\mathbf{B}}, \qquad \widehat{\mathbf{D}} = \text{diag}(\widehat{\boldsymbol{\Omega}}_1, \ldots, \widehat{\boldsymbol{\Omega}}_K).$$

We assume $\widehat{\boldsymbol{\Omega}}_k \to_p \boldsymbol{\Omega}_{kk}$ for all $k$.

Throughout, mean-squared error (MSE) under squared loss decomposes as
$$\mathbb{E}\|\widehat{\theta} - \boldsymbol{\beta}^\star\|_2^2 = \underbrace{\|\mathbb{E}[\widehat{\theta}] - \boldsymbol{\beta}^\star\|_2^2}_{\text{(squared bias)}} + \underbrace{\text{tr}\{\text{Cov}(\widehat{\theta})\}}_{\text{(variance)}}.$$

Because all signals share the same mean $\boldsymbol{\beta}^\star + \boldsymbol{b}$, any linear unbiased combination has squared bias $\|\boldsymbol{b}\|_2^2$. Hence MSE comparisons reduce to comparing the *trace* of their covariance matrices.

**Step 1: GLS dominates any component (Loewner order).** Consider the class of linear unbiased estimators $\widehat{\boldsymbol{\beta}} = \mathbf{A}^\top \widehat{\mathbf{B}}$ with $\mathbf{A} \in \mathbb{R}^{Kd \times d}$ satisfying the unbiasedness constraint $\mathbf{A}^\top \mathbf{H} = \mathbf{I}_p$. The GLS choice $\mathbf{A}_\star = \mathbf{V}^{-1} \mathbf{H}(\mathbf{H}^\top \mathbf{V}^{-1} \mathbf{H})^{-1}$ minimizes $\text{Cov}(\mathbf{A}^\top \widehat{\mathbf{B}}) = \mathbf{A}^\top \mathbf{V} \mathbf{A}$ in the Loewner order; hence for any feasible $\mathbf{A}$,
$$\mathbf{G}(\mathbf{V}) = \text{Cov}(\widetilde{\boldsymbol{\beta}}_{\text{Ens}}) \preceq \mathbf{A}^\top \mathbf{V} \mathbf{A}.$$

Take the special feasible choice that picks a single component: for fixed $j$, set $\mathbf{A}_j = \mathbf{e}_j \otimes \mathbf{I}_p$ (so $\mathbf{A}_j^\top \mathbf{H} = \mathbf{I}_p$). Then $\mathbf{A}_j^\top \mathbf{V} \mathbf{A}_j = \boldsymbol{\Omega}_{jj}$ and thus
$$\mathbf{G}(\mathbf{V}) \preceq \boldsymbol{\Omega}_{jj} \qquad \forall j \in \{1, \ldots, K\}. \tag{8}$$

Taking traces yields the componentwise variance dominance:
$$\text{tr}\{\mathbf{G}(\mathbf{V})\} \leq \min_j \text{tr}(\boldsymbol{\Omega}_{jj}).$$

**Step 2: Comparing PW to GLS via a block-diagonal perturbation.** Write $\mathbf{V} = \mathbf{D}^{1/2}(\mathbf{I} + \mathbf{E})\mathbf{D}^{1/2}$ with
$$\mathbf{E} := \mathbf{D}^{-1/2}(\mathbf{V} - \mathbf{D})\mathbf{D}^{-1/2},$$

so $\mathbf{E}$ contains only the (scaled) off-diagonal blocks. Assume $\|\mathbf{E}\|_{\mathrm{op}} < 1$, so $(\mathbf{I} + \mathbf{E})^{-1} = \sum_{r \geq 0}(-\mathbf{E})^r$. Then

$$\mathbf{V}^{-1} = \mathbf{D}^{-1/2}(\mathbf{I} + \mathbf{E})^{-1}\mathbf{D}^{-1/2} = \mathbf{D}^{-1} - \mathbf{D}^{-1/2}\mathbf{E}\mathbf{D}^{-1/2} + \mathbf{R},$$

with remainder $\mathbf{R} = \mathbf{D}^{-1/2}\sum_{r \geq 2}(-\mathbf{E})^r\mathbf{D}^{-1/2}$. Define

$$\mathbf{G}(\mathbf{M}) := (\mathbf{H}^\top \mathbf{M}^{-1}\mathbf{H})^{-1} \qquad (\mathbf{M} \succ 0).$$

Then

$$\mathbf{G}(\mathbf{V})^{-1} - \mathbf{G}(\mathbf{D})^{-1} = \mathbf{H}^\top(\mathbf{V}^{-1} - \mathbf{D}^{-1})\mathbf{H} = \mathbf{H}^\top \mathbf{D}^{-1/2}\Big(-\mathbf{E} + \sum_{r \geq 2}(-\mathbf{E})^r\Big)\mathbf{D}^{-1/2}\mathbf{H}.$$

Using the identity $\mathbf{A}^{-1} - \mathbf{B}^{-1} = \mathbf{A}^{-1}(\mathbf{B} - \mathbf{A})\mathbf{B}^{-1}$ and the positive definiteness of $\mathbf{G}(\cdot)$, one obtains the Loewner bound

$$\mathbf{G}(\mathbf{D}) - \mathbf{G}(\mathbf{V}) \succeq \mathbf{G}(\mathbf{D})\,\mathbf{H}^\top \mathbf{D}^{-1/2}\Big(\mathbf{E} - \sum_{r \geq 2}(-\mathbf{E})^r\Big)\mathbf{D}^{-1/2}\mathbf{H}\,\mathbf{G}(\mathbf{D}) \succeq \mathbf{0}. \qquad (9)$$

Taking traces and the norm bound $\sum_{r \geq 1}\|\mathbf{E}\|_{\mathrm{op}}^r = \|\mathbf{E}\|_{\mathrm{op}}/(1 - \|\mathbf{E}\|_{\mathrm{op}})$,

$$\mathrm{tr}\{\mathbf{G}(\mathbf{D})\} \leq \mathrm{tr}\{\mathbf{G}(\mathbf{V})\} + \frac{1}{1 - \|\mathbf{E}\|_{\mathrm{op}}}\,\mathrm{tr}\Big[\mathbf{G}(\mathbf{D})\,\mathbf{H}^\top \mathbf{D}^{-1}(\mathbf{V} - \mathbf{D})\mathbf{D}^{-1}\mathbf{H}\,\mathbf{G}(\mathbf{D})\Big]$$

$$=: \mathrm{tr}\{\mathbf{G}(\mathbf{V})\} + \Delta_{\mathrm{cross}}. \qquad (10)$$

Thus the variance of the block-diagonal precision estimator (with true $\mathbf{D}$) is within $\Delta_{\mathrm{cross}}$ of GLS. By Slutsky and continuity of the map $(\cdot) \mapsto (\mathbf{H}^\top(\cdot)^{-1}\mathbf{H})^{-1}$ on $\mathbb{S}_{++}$, replacing $\mathbf{D}$ by $\widehat{\mathbf{D}} \to_p \mathbf{D}$ yields the same relation with an $o(1)$ term.

**Step 3: Assemble the bound.** By Step 1 and Eq. 10,

$$\mathrm{tr}\{\mathrm{Cov}(\widehat{\boldsymbol{\beta}}_{\mathrm{PW}})\} \leq \mathrm{tr}\{\mathbf{G}(\mathbf{D})\} + o(1) \leq \mathrm{tr}\{\mathbf{G}(\mathbf{V})\} + \Delta_{\mathrm{cross}} + o(1) \leq \min_j \mathrm{tr}(\boldsymbol{\Omega}_{jj}) + \Delta_{\mathrm{cross}} + o(1).$$

Adding the common squared bias $\|\boldsymbol{b}\|_2^2$ to both sides yields

$$\mathbb{E}\|\widehat{\boldsymbol{\beta}}_{\mathrm{PW}} - \boldsymbol{\beta}^\star\|_2^2 \leq \min_{j \in \mathcal{K}} \mathbb{E}\|\widehat{\boldsymbol{\beta}}_{\mathrm{EB},j} - \boldsymbol{\beta}^\star\|_2^2 + \Delta_{\mathrm{cross}} + o(1),$$

which is Theorem 5.

**Explicit form of the penalty.** From Eq. 10, one convenient expression is

$$\Delta_{\mathrm{cross}} \leq \frac{1}{1 - \|\mathbf{E}\|_{\mathrm{op}}}\,\mathrm{tr}\Big[(\mathbf{H}^\top \mathbf{D}^{-1}\mathbf{H})^{-1}\mathbf{H}^\top \mathbf{D}^{-1}(\mathbf{V} - \mathbf{D})\mathbf{D}^{-1}\mathbf{H}(\mathbf{H}^\top \mathbf{D}^{-1}\mathbf{H})^{-1}\Big],$$

with $\mathbf{E} = \mathbf{D}^{-1/2}(\mathbf{V} - \mathbf{D})\mathbf{D}^{-1/2}$. This depends only on the off-diagonal cross-covariances $\{\boldsymbol{\Omega}_{k\ell}\}_{k \neq \ell}$. $\qquad\square$

