# OpenReview forum: "Adaptive robust integration of internal data with external summaries under distributional shift"
_ICLR.cc/2026/Conference — Submitted to ICLR 2026_

### Official Review · Reviewer_a6WT · 2025-10-18

**Soundness:** 2
**Presentation:** 3
**Contribution:** 2
**Rating:** 2
**Confidence:** 2

**Summary:**

The paper tackles a common but messy scenario: you have a small internal dataset with rich covariates $(X,Z)$ and a huge external dataset that exposes only summary statistics for a reduced model on $X$. Populations differ, so naïve pooling is brittle. The authors propose a distributionally robust optimization (DRO) framework that (i) fuses internal individual‑level likelihood with moment constraints derived from the external summaries, using Cressie–Read divergences to reweight the internal sample, (ii) supplies an Empirical‑Bayes stabilization (EB‑DRO) that adaptively shrinks back toward the internal‑only MLE when shifts are large, and (iii) builds a precision‑weighted ensemble over multiple Cressie–Read indices to smooth out the k‑choice sensitivity. Theory covers duality, asymptotics with and without shift, and an oracle‑risk result for the EB shrinkage; experiments are simulation‑based across three shift families.

**Strengths:**

- Clean unification via Cressie–Read duality. The derivation in Theorem 1/Corollary 1 is tidy and subsumes EL, ET, GMM, χ² as special cases (Table 1).

**Weaknesses:**

- All evidence is synthetic. No real‑world case study (biobank ↔ internal study, EHR, registries). Given the paper’s “privacy‑preserving external summaries” pitch, a real deployment would materially increase credibility.
- k‑selection is under‑specified. The Cressie–Read index controls finite‑sample robustness, and Fig. 1 shows strong sensitivity. Beyond “ensemble it,” there’s little guidance for choosing
$k$ or adapting it to a measured shift.
- The model assumptions are likely to be flawed (see the first question below).
- There is a mismatch between the setup and the motivated application.

**Questions:**

- The GLM model $Y\mid (X,Z)$ does not necessarily imply that $Y\mid X$ is also GLM. The authors should provide more discussion or examples for this.
- For the motivating application, UK biobank, one usually has access to only summarized data ($Y^{\top}X$ and $X^{\top}X$), to protect patient information. This is different from the setup in the current paper.
- 'abcdefghijrefer' on abstract
- (dispersion parameter) For GLM, there is also a dispersion parameter that determines the likelihood. The authors should note this.

---

### Official Review · Reviewer_MkZp · 2025-10-30

**Soundness:** 2
**Presentation:** 3
**Contribution:** 3
**Rating:** 4
**Confidence:** 2

**Summary:**

The paper addresses the problem of integrating summary statistics (i.e., MLE estimates) from a large-scale study ($N > 10^5$) with covariates $X$ into a small-scale internal study with covariates $X, Z$. Covariates $X$ are measured across both studies, while covariates $Z$ are only observed in the internal study. Assuming that $\beta$ and $\theta$ indicate the internal and external model parameters, respectively, the goal is to robustly estimate $\beta$ by leveraging the precision of $\theta$.

The two studies may present a distributional shift, e.g., due to sampling bias or differences in the data collection protocols. The authors claim that shifts stem from changes in the distribution of individual covariates or the conditional distribution $Y|X$, or both. To tackle these shifts, they propose a DRO formulation claimed to be robust to both.

Their DRO formulation combines an MLE estimator for $Y|X,Z$ with per-sample re-weighting via Cressie-Read divergences to match moment constraints between the current parameters for the internal estimators and the MLE parameters from the external large-scale study.

The authors simplify their objective to a min-max optimization problem over $\beta$ and $\lambda$. The use of a Cressie-Read divergence measure allows them to recover several estimators by varying $k$ to achieve desired bias-variance-robustness trade-offs. They optimize the problem by iteratively maximizing $\lambda$ and updating $\beta$ for up to $T$ steps or until $\beta$ converges.

The authors also show that DRO estimators are robust to mild shifts, but their robustness deteriorates as shifts increase (and can even perform worse than using only the individual-level data). To tackle this, they propose an Empirical-Bayes stabilized DRO estimator, EB-DRO, which, in extreme cases, recovers either DRO or a naive estimator, so that EB-DRO is never worse than either.

To further stabilize performance across shifts, the authors aggregate several EB-DRO estimators in an ensemble.

They evaluate the three estimators (DRO, EB-DRO, EB-DRO Ensemble) using a synthetic internal dataset of 500 samples with five binary covariates and the MLE parameters of the model fitted on a synthetic external dataset of $10^5$ samples with three binary covariates (shared with the internal dataset), sampled from the same distribution. The authors simulate three shifts in the data:

1. Shifting the intercept of the data-generating coefficients.
2. Perturbing the first covariate by altering the data-generating coefficients for upper tail outcomes $\rightarrow$ sub-group specific shift.
3. Rotate the data-generating coefficients, changing the direction of the signal while preserving magnitude.

They evaluate estimators across 19 values of $k$, showing that the resulting estimators exhibit varying performance under moderate and large shifts. In particular, the results show that the DRO estimator is only robust to small shifts, yielding worse performance than the internal-only estimator under moderate or large shifts, while the EB-DRO estimator is significantly more robust, with significant improvements under no or moderate shift and matching or performing slightly worse than the internal-only estimator under large shifts, but the latter issue can be solved using the EB-DRO ensemble, which performs equally to the internal-only baseline in its worst-case.

**Strengths:**

- The paper is well-written, motivated, and tackles an interesting and practical problem.
- The proposed estimators are theoretically motivated, and the relevant proofs are provided.
- The claims of the paper are well-reflected in the empirical experiments.

**Weaknesses:**

- The evaluation is limited to a setup with a limited number of covariates, as noted by the authors. It would have been very interesting to see how their approach scales as the number of covariates (or the delta in the number of covariates between studies) grows.
- The estimators are only evaluated on a single artificial setup. It would have been interesting to see performance on real-world data or in a setup where the covariates are sampled from a distribution other than Bernoulli.

Other minor details:

- There are two typos in lines 11 and 12 of the abstract.

**Questions:**

1. Could you provide some experimental results with a larger number of covariates and/or results on real-world data?
2. How would you recommend picking $k$ in practice?

---

### Official Review · Reviewer_i3od · 2025-11-01

**Soundness:** 2
**Presentation:** 1
**Contribution:** 2
**Rating:** 2
**Confidence:** 4

**Summary:**

The paper studies integration of internal individual-level data with external summary statistics under distribution shift, formulating a Cressie-Read based distributional robust (DRO) estimator. It proposes an empirical Bayes (EB) shrinkage that convexly combines the DRO estimator with the internal-only GLM estimator via a data-adaptive weight. They also propose an ensemble across divergence choices for stability. Theoretical results that the estimator targets a pseudo-true parameter (bias need not vanish), which motivates the EB and ensemble. However, the exposition is very hard to follow (key assumptions/lemmas and regularity conditions only in the appendix). Empirically, evidence is simulation-only with no real data or benchmark evaluation, so practical value and robustness claims are hard to tell.

**Strengths:**

1. Rigorous setup connecting DRO with Cressie-Read family recovers moment-baesd estimators in the limit.

2. EB shrinkage and precision-weighted ensemble make sense and should stabilize performance across divergence choices.

**Weaknesses:**

1. Strange errors in the 2nd sentence in the abstract suggests LLM-generated the text: "Large datasets such as biobanksabcdefghijrefer to as external dataabcdefghijoffer substantial sample sizes but often lack in-depth information due to cost constraints."

2. Poor readability makes the paper extremely hard to follow. Further, the important assumptions, lemmas, and regularity conditions are not included in the main text and make readability even more difficult.

3. The influence-function based development does not yield double robustness, even though it seems like a feasible thing to combine outcome-regression and density ratio weighting to establish this. Further, focus on GLMs can limit robustness and it is not clear how to extend to nonparametric ML models or high-D models.

4. Dominance bound depends on covariance terms that are not estimated or bounded.

5. No real-data applications or benchmark datasets make this a severe limitation

6. Relative to empirical likelihood and GMM type approaches for summary-based integration, the contribution feels incremental. Also, there are many papers by now that consider DRO-based methods for multiple data sources, esp in the causal inference context. None of these papers are cited.

**Questions:**

1. Can the data-driven estimator for mixing weight explicitly minimize MSE, or MSE with a penalty for large bias, with guarantees on no negative transfer?

2. Under what structural assumptions, e.g., invariance, moment restrictions, does the pseduo-true target align with a relevant estimand, and how would practitioners diagnose when it does not?

3. What breaks algorithmically and theoretically if the predictive model is not a GLM?

4. How do you estimate cross-covariances among base estimators to control the penalty in the dominance bound, and how sensitive are results to the estimation?

5. What real world examples of internal individual-level data with external summary statistics motivate this study? Running the method and showing its performance compared to existing methods on real world data is really needed.

---

### Official Review · Reviewer_Cf4W · 2025-11-01

**Soundness:** 3
**Presentation:** 3
**Contribution:** 3
**Rating:** 6
**Confidence:** 4

**Summary:**

This paper proposes a unified Distributionally Robust Optimization (DRO) framework for integrating internal individual-level data with external summary-level information when the two sources exhibit distributional shift. The authors extend classical empirical-likelihood and exponential-tilting estimators by introducing a Cressie–Read divergence family that regularizes the reweighting of internal samples to align with external summaries. They then develop two extensions: (i) an Empirical Bayes DRO (EB-DRO) that adaptively shrinks between the internal-only and DRO estimators to stabilize performance under strong shift, and (ii) an ensemble EB-DRO that aggregates multiple divergence families to further enhance robustness. Theoretical results establish asymptotic normality, efficiency improvement under no shift, and bounded bias under mild to moderate shift, providing a rigorous foundation for robust data integration.

**Strengths:**

1. Comprehensive theoretical development.
The paper offers a mathematically rigorous treatment of the proposed estimators, with clear asymptotic analyses and bias–variance trade-off results. The connection to empirical likelihood, exponential tilting, and GMM through the generalized Cressie–Read family is elegant and intellectually unifying.
2. Methodological depth and breadth.
The authors present three layers of estimators (DRO, EB-DRO, ensemble EB-DRO), each motivated by a well-defined robustness consideration. This layered structure demonstrates strong conceptual completeness and awareness of practical limitations.
3. Solid technical correctness.
The proofs, although dense, appear sound. The asymptotic theory and pseudo-true bias expansion are technically competent and extend existing DRO literature in a non-trivial way.

**Weaknesses:**

1. Exposition and clarity.
The paper’s presentation is dense and difficult to follow. Key definitions (e.g., moment constraints, divergence functions, the role of u(X,Z;β,θ^\star) are introduced abruptly, and intuitive explanations for equations are scarce. On the other hand, many concepts appear early in the article but show their definition later. For a theory-heavy paper, pedagogical clarity is essential but lacking here.
2. Limited scope of “distributional shift.”
Despite its title, all simulated shifts are implemented by perturbing regression coefficients—i.e., conditional shifts. The work does not cover covariate shifts, which substantially limits the claimed generality of “robust under shift.”
3. Theory–experiment disconnect.
The simulations confirm qualitative robustness but do not validate the key theoretical claims (e.g., linear bias growth with δ or asymptotic variance reduction). Empirical verification of theoretical quantities would significantly strengthen credibility.
4. Lack of discussion on computational complexity and limitations.
The nested Newton optimization and the stability of the EB shrinkage parameter α are not analyzed. Moreover, the method’s sensitivity to noisy external summaries is unaddressed. I think this part is important since this work doesn’t provide open-source code at least in the present.

**Questions:**

1. The authors should explicitly define what kinds of distributional shifts the theory covers (conditional, covariate, or both). Currently, the framework is “agnostic” in claim but specific in practice.
2. A separate subsection or figure explaining the intuition behind the DRO–EB–ensemble hierarchy would greatly help readers navigate the theoretical landscape.
3. Provide diagnostic plots or numerical checks verifying asymptotic predictions—e.g., empirical bias versus theoretical δ, or variance convergence rates
4. Complexity analysis of the nested iteration and guidelines for tuning divergence index k or shrinkage weight α would make the approach more reproducible.
5. Finally, I have a question on the big picture: the MSE results of the experiment are always not so bad in different settings, so how can we know we are benefiting from the leverage of the external data?

---

### Meta-Review · Area_Chair_MVN3 · 2025-12-30

**Summary:**

The reviewers' common concerns are 1) lack of justification of key assumptions; 2) lack of more realistic experiments; 3) lack of presentation clarity; 4) lack of more detailed analysis of the solution, including sensitivity to hyperparameters.

Personally, I found the DRO idea for integrating internal data with external data summaries an interesting one. I encourage resubmission of the paper in the future after fixing the above issues.

**Reviewer Concerns:**

All of the concerns are outstanding due to no rebuttal.

**Reviewer Scores:**

Scores would not be changed due to no rebuttal.

---

### Decision · Program_Chairs · 2026-01-26

Reject